# Endothelial RNF20 suppresses endothelial-to-mesenchymal transition and safeguards physiological angiocrine signaling to prevent congenital heart disease

Yanliang Dou[1,14], Nalan Tetik-Elsherbiny[1,14], Rui Gao[1,2,14], Yonggang Ren[1,2], Yu-wen Chen[1,3], Moritz Merbecks[4], Aadhyaa Setya[1], Olga Lityagina[1], Yinuo Wang[1], Evgeny Chichelnitskiy[2], Aya Abouissa[5], Chi-Chung Wu[6], Guillermo Barreto[7], Michael Potente[8,9,10,11], Thomas Wieland[11,12], Roxana Ola[13], Philippe Grieshaber[3], Tsvetomir Loukanov[3], Matthias Gorenflo[4], Joerg Heineke[5,10,11], Julio Cordero[1,11] ✉ & Gergana Dobreva[1,2,10,11] ✉

Heart morphogenesis and function rely on intricate communication among distinct cardiac cell types. How their co-development and crosstalk are coordinated is largely unexplored. Our study unveils key functions of the histone H2B ubiquitin (H2Bub1) ligase RNF20 in second heart field development and cardiac endothelial cells. We demonstrate that RNF20 promotes *Nrg1* expression through a RNF20-H2Bub1-dependent mechanism and restrains TGF-β signaling by influencing RNA polymerase II pause release at TGF-β target genes in endothelial cells. While heightened TGF-β signaling following RNF20 loss results endothelial-to-mesenchymal transition (EndMT), both impaired Nrg1 signaling and elevated TGF-β activity contribute to abnormal cardiomyocyte proliferation and contractility. Importantly, *RNF20* expression is significantly reduced in cardiac endothelial cells from congenital heart disease patients showing a positive correlation with oxygen saturation and a negative correlation with key components and downstream effectors of TGF-β signaling. In summary, our work identifies a crucial role for RNF20 in safeguarding endothelial identity and physiological angiocrine signaling, thereby ensuring proper heart development and function.

Congenital heart disease (CHD) originates during the early stages of pregnancy, as a result of abnormalities in the intricate development of the heart[1,2]. Despite advances in understanding the genetic basis of CHD, it is increasingly evident that an intricate interplay between genetics and epigenetics plays a pivotal role in its pathogenesis, underscoring the importance of unraveling the molecular players at the interface of these processes[3–6].

Heart development relies on the precise specification, expansion, migration, and differentiation of cardiovascular progenitor cells into various cardiac cell types[7,8]. These distinct cardiac cell types engage in constant crosstalk, which is essential for cardiac morphogenesis, postnatal cardiac growth, contractile performance, and rhythmicity[9–11]. During gastrulation, cardiac mesoderm is pre-patterned into distinct precursors[7,12]. The first heart field (FHF) forms the primitive heart tube

and left ventricle, while the second heart field (SHF) supplies progenitors to the outflow tract, right ventricle, atria, and sinoatrial node, with its anterior and posterior regions contributing to arterial and venous structures, respectively[7,13–15]. Perturbations of SHF development are responsible for the most prevalent congenital heart defects, underscoring its clinical importance.

SHF precursors, are multipotent and give rise to distinct lineages in the heart, including myocardial and endocardial endothelial cells (ECs)[16–18]. The endocardium has recently gained attention due to its remarkable cellular plasticity and important contributions, as well as its signaling function during heart development, disease, and regeneration[19]. On one hand, it forms a vascular endothelial layer contiguous with the rest of the vasculature, providing nutrients and oxygen to the heart. On the other hand, by receiving and releasing cues that drive trabeculation, valve and septum formation, and myocardial compaction, the endocardium exerts an instructive influence known as angiocrine signaling[9–11]. For instance, Notch activity in endocardial ECs promotes proliferation and differentiation of adjacent myocardial cells, as well as ventricular trabeculation, by inducing the expression of *Bmp10*, *Efnb2*, and *Nrg1*[20]. Nrg1, expressed by endocardial ECs, activates the tyrosine kinase receptor ErbB4 and its co-receptor ErbB2, expressed on adjacent cardiomyocytes, resulting in cardiomyocyte proliferation, survival and cardiac trabeculation[11,20,21]. An important anatomical signaling hub between the endocardium and myocardium is located in the developing endocardial cushions of the outflow tract (OFT) and the atrioventricular canal (AVC), which later form the cardiac valves[22,23]. Endocardial ECs lining these cushions undergo a process referred to as endothelial-to-mesenchymal transition (EndMT), transforming into mesenchymal cells that move into the cushions to form the future valve interstitial cells. Autocrine and paracrine signaling molecules have been shown to drive EndMT, including TGF-β, Wnt/β-catenin, VEGF and Notch[23–26]. TGF-β and BMP signaling play a pivotal role in cardiac cushion formation, as well as the regulation of EC identity and proliferation[27,28]. Upon ligand binding, TGF-β or BMP receptors phosphorylate specific Smads (Smad2/3 for TGF-β; Smad1/5/8 for BMP), which translocate to the nucleus with Smad4 to regulate gene expression[29]. In the developing heart, myocardial BMP2 induces endocardial *Tgfb2* expression, triggering EndMT and mesenchymal invasion of the cushions[23]. Tgfb2-driven EndMT depends on Snail and β-catenin, while Notch induces Slug to repress VE-cadherin[30]. VEGF signaling also plays a critical role in valve development, where its levels must be precisely regulated. Insufficient VEGF leads to reduced endothelial cell proliferation and hypoplastic cushions due to impaired EndMT, while excessive VEGF inhibits EndMT, underscoring the importance of a tightly controlled physiological VEGF range for proper cardiac cushion formation[24,31]. Thus, proper cardiogenesis depends on complex signaling pathways and cell-cell interactions; however, the mechanisms that coordinate signal-dependent transcription and inter-pathway crosstalk to control cardiac cell function remain largely unclear.

One key control mechanism of inducible transcription is the regulation of RNA Polymerase II (Pol II) pausing and release near the promoter region[32–34]. Numerous factors have been identified that play a role in the regulation of Pol II pausing and elongation, including RNF20, the major E3 ubiquitin ligase responsible for mono-ubiquitination of histone H2B at lysine 120 (H2BK120ub, H2Bub1)[35–37]. RNF20 plays a critical role in regulating Pol II activity and functions as a central signaling hub that influences the transcriptional dynamics[37–40]. On the one hand, RNF20-mediated H2Bub1 promotes transcriptional elongation by cooperating with the FACT complex to destabilize nucleosomes[37]. Conversely, RNF20 impedes the recruitment of TFIIS, a factor necessary for the release of Pol II into productive elongation at tumor-promoting genes, thereby repressing a pro-oncogenic transcriptional program[40]. Similarly, in endothelial cells, RNF20 plays a pivotal role in regulating VEGF-Notch signaling circuits during vessel

growth through its dual role in regulating Pol II activity[41]. This function is tightly coordinated, as signaling pathways modulate RNF20 activity, which in turn shapes Pol II pausing and elongation[41,42]. The crucial role of RNF20 in proper heart development became evident through exome sequencing data, which identified a de novo mutation in RNF20 associated with CHD in human patients[6]. Further research in animal models revealed that RNF20-mediated H2Bub1 is essential for heart development[43] and drives cardiomyocyte maturation and polarization in the early postnatal period[44,45]. Despite these intriguing findings, the role of RNF20 in the different cardiac cell types for ensuring proper cardiogenesis and CHD remains largely unexplored.

In this study, we found a significant reduction in *RNF20* levels in cardiac endothelial cells from patients with Tetralogy of Fallot (TOF), which positively correlates with oxygen saturation and inversely correlates with the expression of key components and downstream effectors of TGF-β signaling. Functionally, RNF20 plays a central role in cardiac ECs for proper cardiac morphogenesis and function by inhibiting EndMT and aberrant angiocrine signaling that induce CM cell cycle withdrawal and arrhythmic beating behavior. Mechanistically, RNF20 exerts these effects by maintaining the critical balance between TGF-β and growth factor signaling during heart development, through regulating Pol II pausing and elongation, respectively.

## Results

### RNF20 levels are decreased in TOF patients and positively correlate with SpO2 levels

TOF is the most common type of cyanotic CHD and the most frequent complex CHD encountered in adulthood. It is characterized by a combination of anatomical defects, including ventricular septal defect (VSD), overriding of the aorta, variable obstruction of the right ventricular outflow tract (RVOT), and right ventricular (RV) hypertrophy[46]. These structural abnormalities lead to a pathophysiological state characterized by chronic hypoxia and right ventricular pressure overload, which together drive profound alterations in cardiac structure, function, and cellular signaling. Bulk RNA-sequencing data from Tetralogy of Fallot (TOF) patients revealed reduced *RNF20* expression compared to control donors (Supplementary Fig. 1a[47]). Further interrogation of single-cell RNA-sequencing data[48] demonstrated significant downregulation of *RNF20* in endothelial cells, mural cells, and cardiac fibroblasts in TOF patients relative to controls (Fig. 1a, Supplementary Fig. 1b–e). Gene Ontology (GO) analysis of the EC population exhibiting major *RNF20* loss showed enrichment for genes involved in cell division, regulation of small GTPase-mediated signal transduction, and actin filament-based processes (Fig. 1b). Notably, this EC population was expanded in TOF patients, in line with a recent study suggesting an ERG- and TEAD1-mediated EndMT program activated in TOF patients[49].

To further investigate the impact of chronic hypoxia and right ventricular pressure overload on *RNF20* expression, we performed RNA sequencing on right ventricular outflow tract (RVOT) specimens from TOF patients and collected clinical parameters such as peripheral oxygen saturation (SpO$_2$) at admission and echocardiography reports (Supplementary Table 1). Correlation analysis revealed a significant positive correlation between *RNF20* levels and SpO$_2$, but no correlation with patient age or right ventricular pressure (Fig. 1c, d), suggesting that *RNF20* expression may be directly regulated by oxygen levels. Consistent with this, experimental exposure of cells to hypoxic conditions resulted in a significant reduction in *RNF20* mRNA levels, accompanied by altered expression of hypoxia-responsive genes (Fig. 1e).

### *Rnf20* is essential for SHF development

A de novo mutation in RNF20 was identified in a patient with congenital heart disease[6], and *RNF20* expression is significantly downregulated in ECs of Tetralogy of Fallot (TOF) patients as well as under hypoxic

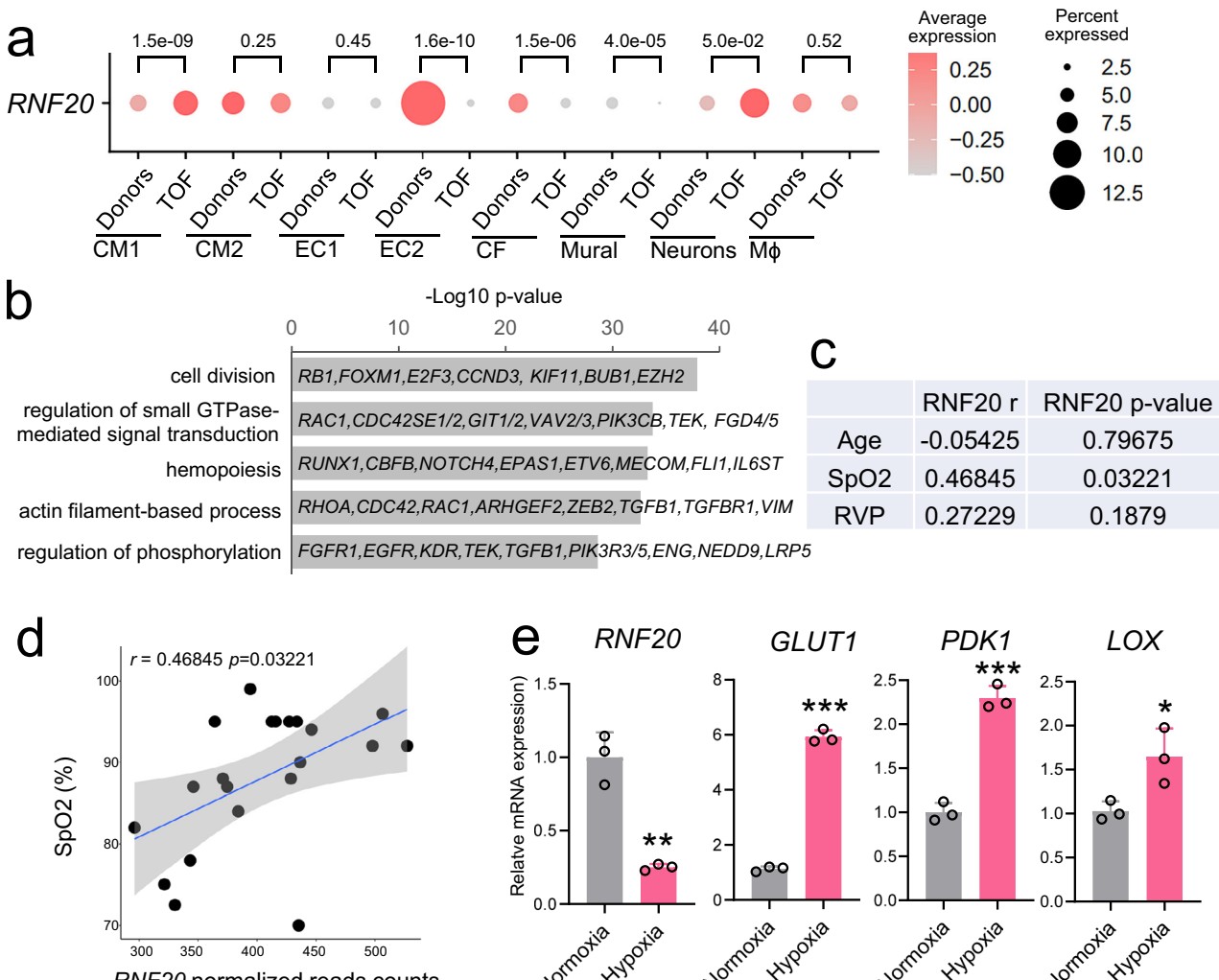

**Fig. 1 | *RNF20* levels are reduced in TOF patients and positively correlate with SpO₂ levels. a** Dot plot showing the expression of *RNF20* in different cell populations separated by Donor and Tetralogy of Fallot (TOF) patients from (GSE203274). Size of the dots reflects the percentage of cells expressing *RNF20* and color represents the average expression level in the different cell types from donor and TOF patients. P-values were determined using the two-sided Wilcoxon test provided by FindMarkers. **b** Representative genes and enriched GO terms in marker genes for EC2. *P*-values were calculated using a hypergeometric test as performed by Metascape (v.3.5). **c** Spearman correlation between *RNF20* expression and clinical traits in TOF patients (*n* = 25). **d** Dot plot showing the correlation between SpO₂ levels and *RNF20* gene expression in samples from TOF patients (*n* = 25). Each black dot represents an independent biological sample. The blue line corresponds to the linear regression fit, while the surrounding gray area denotes the 95% confidence interval, indicating the range within which the true regression line is likely to lie with 95% certainty. Spearman correlation was used to assess correlations between gene expression levels and clinical data in (**c, d**), and *p*-values were estimated using a two-tailed test based on a t-distribution approximation. **e** Relative mRNA expression of *RNF20*, *GLUT1*, *PDK1* and *LOX* in HUVECs cultured in normoxia (*n* = 3) and hypoxia (1% oxygen level, 24 h, *n* = 3), determined by qPCR. Statistical analysis between two groups in (**e**) was performed using an unpaired two-tailed Student's *t* test. Data are shown as means ± SEM. *$p < 0.05$, **$p < 0.01$, ***$p < 0.001$.

conditions. These observations prompted us to investigate the function of Rnf20 in cardiac progenitors and ECs. Single-cell RNA sequencing (scRNA-seq) data from E7.75 mouse embryos[50] revealed significantly higher expression of *Rnf20* in second heart field (SHF) progenitors (Isl1[pos]) compared to first heart field (FHF) precursors (Nkx2.5[pos] Isl1[neg]) (Fig. 2a), suggesting an important role for Rnf20 in SHF development. Perturbation of SHF deployment has previously been linked to TOF pathogenesis[51]. Conditional ablation of *Rnf20* using Nkx2.5-Cre has demonstrated its critical role in heart development[43]. However, since *Nkx2.5* is expressed in both FHF and SHF lineages[52], we employed a more specific approach to study the role of *Rnf20* in SHF progenitors by using the Isl1-Cre driver line[53]. Isl1 is a well-established marker of second heart field (SHF) progenitors in the mouse. While it is initially expressed in both anterior and posterior SHF domains, its expression becomes progressively restricted to anterior SHF derivatives, particularly those contributing to the outflow tract and right ventricle[13,14,53,54]. Quantitative PCR

(qPCR) analysis of dissected right (RV) and left ventricles (LV) revealed a significant reduction of *Rnf20* expression in the RV of *Isl1^{cre/+}Rnf20^{fl/fl}* hearts, while levels in the LV were unchanged (Fig. 2b).

*Rnf20*-deficient mice did not survive beyond E14.5 and displayed severe cardiac and vascular abnormalities, including a lack of septation of the cardiac OFT, and a smaller RV (Fig. 2c–e, Supplementary Fig. 2a, b). Analysis of image sequences recorded on E10.5 control and *Isl1^{cre/+}Rnf20^{fl/fl}* hearts using MYOCYTER[55] (Fig. 2f), revealed abnormal contractility with decreased amplitude (Fig. 2g–i); one out of twelve (1/12) control hearts showed arrhythmic beating behavior, while five out of six (5/6) *Isl1^{cre/+}Rnf20^{fl/fl}* hearts exhibited arrhythmic contractility (Supplementary Fig. 2c, Supplementary Table 2). Detailed histological analysis revealed hypoplastic ventricular wall, abnormal trabeculation and disorganized endocardium (Fig. 2j–l).

We next tested whether the decreased cardiomyocyte number could be due to decreased CM proliferation or increased apoptosis.

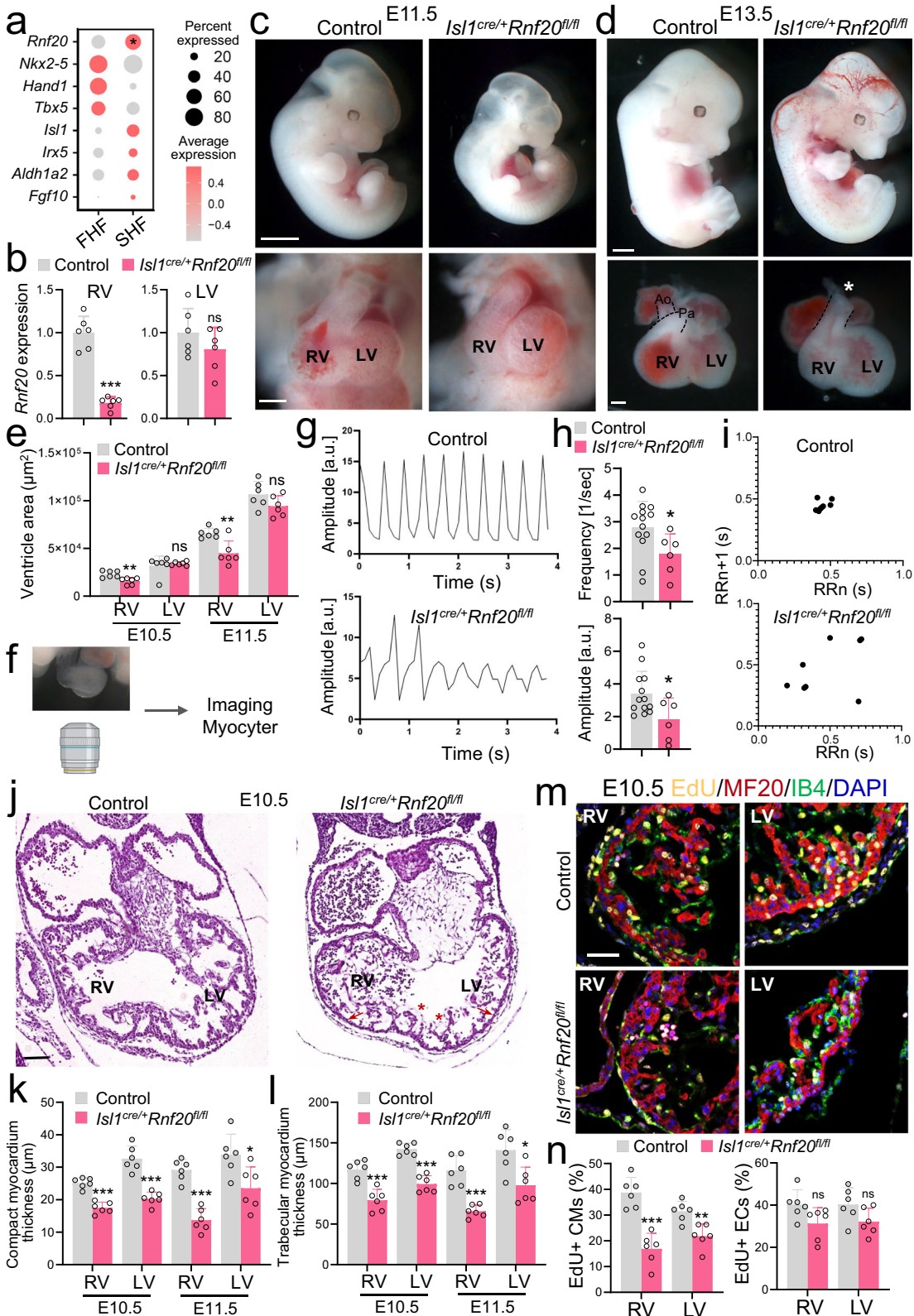

Terminal deoxynucleotidyl transferase dUTP Nick-End Labeling (TUNEL) revealed no apoptosis in both control and *Isl1^{cre/+}Rnf20^{fl/fl}* hearts at E10.5, but significant TUNEL-positive cells in *Isl1^{cre/+}Rnf20^{fl/fl}* hearts at E11.5 (Supplementary Fig. 2d, e). As there were no apoptotic cells detected at E10.5, we examined cell proliferation at this stage. Co-immunostaining for the proliferation marker EdU, together with iso-lectin B4 (IB4) as an endothelial marker and myosin heavy chain

(MF20) as a cardiomyocyte marker, revealed significantly decreased cardiomyocyte proliferation in the right and left ventricles of E10.5 *Rnf20*-deficient hearts, while there was not significant effect on endocardial EC proliferation (Fig. 2m, n). Surprisingly, immunostaining for H2Bub1, a histone modification catalyzed by Rnf20, showed a loss of H2Bub1 in both CMs and ECs of the RV and the LV (Supplementary Fig. 2f, g). H2Bub1 levels are dynamically regulated by the coordinated

**Fig. 2 | Defects in SHF development upon *Rnf20* ablation in Isl1+ cardiovascular progenitors. a** Dot plot of *Rnf20* expression in FHF and SHF cells in E7.75 embryos[50]. FHF population were defined as cells sorted by Nkx2-5 with negligible expression of Isl1. SHF cells were sorted by Isl1. *P* = 0.021 calculated using the Wilcoxon test from FindMarkers. **b** Relative *Rnf20* mRNA expression in dissected RV and LV of E10.5 control (*n* = 6) and *Isl1^{cre/+}Rnf20^{fl/fl}* hearts (*n* = 6). Gross appearance of control and *Isl1^{cre/+}Rnf20^{fl/fl}* embryos at E11.5 (**c**, top panel) and E13.5 (**d**, top panel). Frontal views of E11.5 (**c**, bottom panel) and E13.5 (**d**, bottom panel) control and *Isl1^{cre/+}Rnf20^{fl/fl}* hearts. Scale bars: **c** and **d** (top panels), 1 mm; **c** and **d** (bottom panels), 200 μm. Representative images showing pulmonary artery (Pa) and aorta (Ao) in wild-type embryos, and a single outflow vessel in the knockout (indicated by asterisk). **e** RV and LV size of control and *Isl1^{cre/+}Rnf20^{fl/fl}* hearts (*n* = 6 for E10.5, *n* = 6 for E11.5 per group). **f, g** Analysis of image sequences recorded on E10.5 control and *Isl1^{cre/+}Rnf20^{fl/fl}* hearts using MYOCYTER[55] **h** Frequency and amplitude of image

sequences recorded on E10.5 control (*n* = 13) and Rnf20 *Isl1^{cre/+}Rnf20^{fl/fl}* (*n* = 6) hearts using MYOCYTER[55]. **i** Poincaré plot of image sequences recorded on E10.5 control and *Isl1^{cre/+}Rnf20^{fl/fl}* hearts. **j** Histological analysis of E10.5 control and *Isl1^{cre/+}Rnf20^{fl/fl}* hearts. Red asterisks highlight abnormal endocardial cell morphology. Red arrows indicate reduced myocardial thickness. Scale bars, 200 μm. **k** RV and LV wall thickness of control and *Isl1^{cre/+}Rnf20^{-/fl}* E10.5 or E11.5 hearts (*n* = 6 hearts for all groups). **l** RV and LV trabecular myocardium thickness of control and *Isl1^{cre/+}Rnf20^{-/fl}* E10.5 or E11.5 hearts (*n* = 6 hearts for all groups). Immunostaining with EdU, anti-MF20 (CMs), anti-IB4 (ECs) and DAPI (nucleus) (**m**) and quantification of EdU-labeled CMs (EdU + /MF20+ cells) and EdU-labeled ECs (EdU + /IB4+ cells) in RV and LV of E10.5 control and *Isl1^{cre/+}Rnf20^{fl/fl}* embryos (*n* = 6) (**n**). Scale bars in (**m**), 50 μm. Statistical analysis between two groups in (**b, e, h, i, k, l, n**) was performed using an unpaired two-tailed Student's *t* test. Data are shown as mean ± SEM. *\*p* < 0.05, *\*\*p* < 0.01, *\*\*\*p* < 0.001.

actions of E3 ubiquitin ligases, such as RNF20/RNF40 and deubiquitinases. Thus, the observed decrease in H2Bub1 in the LV may result from secondary changes in other components of this regulatory machinery. Indeed, qPCR analysis of microdissected RV and LV tissue revealed increased *Rnf40* expression in the RV, likely reflecting a compensatory response, and decreased expression in the LV, which may contribute to the reduced H2Bub1 levels (Supplementary Fig. 2h).

To elucidate Rnf20's downstream effectors governing cardiogenesis, we performed single-cell RNA-Seq of E10.5 control and *Isl1^{cre/+}Rnf20^{fl/fl}* hearts (Fig. 3a, b). Similar to previous studies, we detected several distinct cardiac cell types with transcriptional signatures of cardiomyocytes (CM), ECs and endocardial ECs (EC/EndoC), endocardial valve cells (EndoV), mesenchymal cells (MES), epicardial cells (EpiC) and others[56–58] (Fig. 3b, Supplementary Fig. 3a, b, Supplementary Data 1). We observed significant reduction in CM population 1 (CM1) and a corresponding increase in CM2. In the epicardial compartment, mesothelial and proliferative cells (EpiC1) were decreased, while epicardial cells undergoing epithelial-to-mesenchymal transition (EpiC2) were increased. Moreover, the number of mesenchymal cells, including dorsal outflow tract mesenchyme (dOFTm), OFT mesenchyme (OFT-MES), and cardiac mesenchymal cells (cMES), was significantly elevated in *Isl1^{cre/+}Rnf20^{fl/fl}* hearts (Fig. 3c).

Interestingly, *Rnf20* was highly expressed in EpiC, EC/EndoC, OFT and cardiac mesenchymal cells, while its expression was notably lower in CMs (Supplementary Fig. 3c). This pattern is consistent with immunostaining data showing strong Rnf20 protein signals in epicardial and endocardial cells, and a graded expression in cardiomyocytes[43]. Similarly, single-cell transcriptomic analysis of the Isl1 lineage[50] revealed that *Rnf20* expression was significantly higher in Isl1-derived endocardial cells than in Isl1-derived cardiomyocytes in the developing E8.75 heart (Supplementary Fig. 3d, e). Further *Rnf20* levels were significantly lower in EC/EndoC cells of *Isl1^{cre/+}Rnf20^{fl/fl}* hearts (Fig. 3d), suggesting a critical role for Rnf20 in ECs during early heart development.

GO analysis revealed that genes downregulated in *Rnf20*-deficient EC/EndoC cells were linked to heart morphogenesis and regulation of force contraction, but also enriched in GO terms connected to developmental cell growth and positive regulation of cell cycle, further supporting the important function of Rnf20 in heterocellular crosstalk within the heart (Fig. 3e). Genes upregulated in EndoV cells were linked to extracellular matrix organization (ECM), regulation of cell shape and integrin mediated signaling, whereas downregulated genes were similarly involved in heart morphogenesis and regulation of force contraction. Both CM1 and CM2, which showed either no change or a significant decrease in *Rnf20* expression in *Isl1^{cre/+}Rnf20^{fl/fl}* hearts, respectively, exhibited downregulation of genes associated with cell-cell communication pathways involved in cardiac conduction (Fig. 3d, e, Supplementary Data 2).

We next re-clustered the CM and EC populations to investigate alterations in *Isl1^{cre/+}Rnf20^{fl/fl}* hearts in greater detail (Fig. 3f, Supplementary Fig. 4a, b, Supplementary Data 1). We observed significantly lower numbers of proliferative ventricular CMs, in line with the reduced CM proliferation observed in our immunohistochemistry analysis, and increase in ventricular CMs (vCM1 and vCM3), exhibiting signs of metabolic maturation and structural differentiation. Interestingly, the number of right atrial/sinoatrial node (RA/SAN) cardiomyocytes was significantly reduced, suggesting a critical role for Rnf20 in SAN development, whereas the cardiomyocyte numbers in the left atrium (LA), left ventricle (LV), and AVC remained largely unchanged (Fig. 3g).

Re-clustering of ECs identified four distinct clusters (Fig. 3h, Supplementary Fig. 4c, d, Supplementary Data 1). Endothelial cluster 2 (EC2) exhibited high expression of endocardial markers such as *Npr3* and *Nfatc1*, along with general endothelial markers including *Pecam1* and *Kdr*, while showing minimal expression of mesenchymal markers such as *Postn* and *Tagln* (Fig. 3i, Supplementary Fig. 4d, e). Differential expression analysis followed by GO analysis revealed that Rnf20 deficiency in this cluster results in decreased levels of genes linked to positive regulation of cell cycle and regulation of developmental growth, including factors with key function during cardiogenesis such as *Igf1/2*, *Nrg1*, *Nrp1* and *Slit2*. In contrast, upregulated genes were involved in glycolysis and p53-mediated signaling (Fig. 3j, Supplementary Fig. 4e). EC3 showed initial activation of EndMT-associated markers, such as *Zeb2*, which progressively increased in EC1 and was most pronounced in EC4. GO analysis of differentially expressed genes in EC3 upon *Rnf20* loss revealed an upregulation of genes associated with cell migration, while EC4 exhibited increased expression of genes involved in the TGF-β response and downregulation of cell cycle-related genes following Rnf20 deficiency (Fig. 3i, j). Taken together, our results demonstrate that Rnf20 plays a key role in SHF development and suggest an important function in endocardial ECs in regulating EndMT and CM proliferation.

## Rnf20 plays a key role in cardiac ECs for heart development and function

To further examine RNF20 function in ECs, we inactivated *Rnf20* using a *Tie2-Cre* deleter, which recombines in all endothelial lineages, including angioblasts and hemangioblasts[59]. *Tie2-Cre*-mediated *Rnf20* deletion resulted in early embryonic lethality around E10.5 (Supplementary Fig. 5a). *Tie2^{Cre}Rnf20^{fl/fl}* mutant embryos displayed shortened OFT and small RV, structures generated by Isl1+ progenitor cells (Supplementary Fig. 5b), supporting the notion that endothelial RNF20 is important for SHF development.

Since Tie2 is expressed as early as E7.0, we sought to compare Isl1-Cre-mediated deletion of *Rnf20* with an endothelial-specific and temporally controlled approach. To this end, we employed the tamoxifen-inducible *Cdh5-CreERT2* endothelial-targeting line[60] (Fig. 4a,

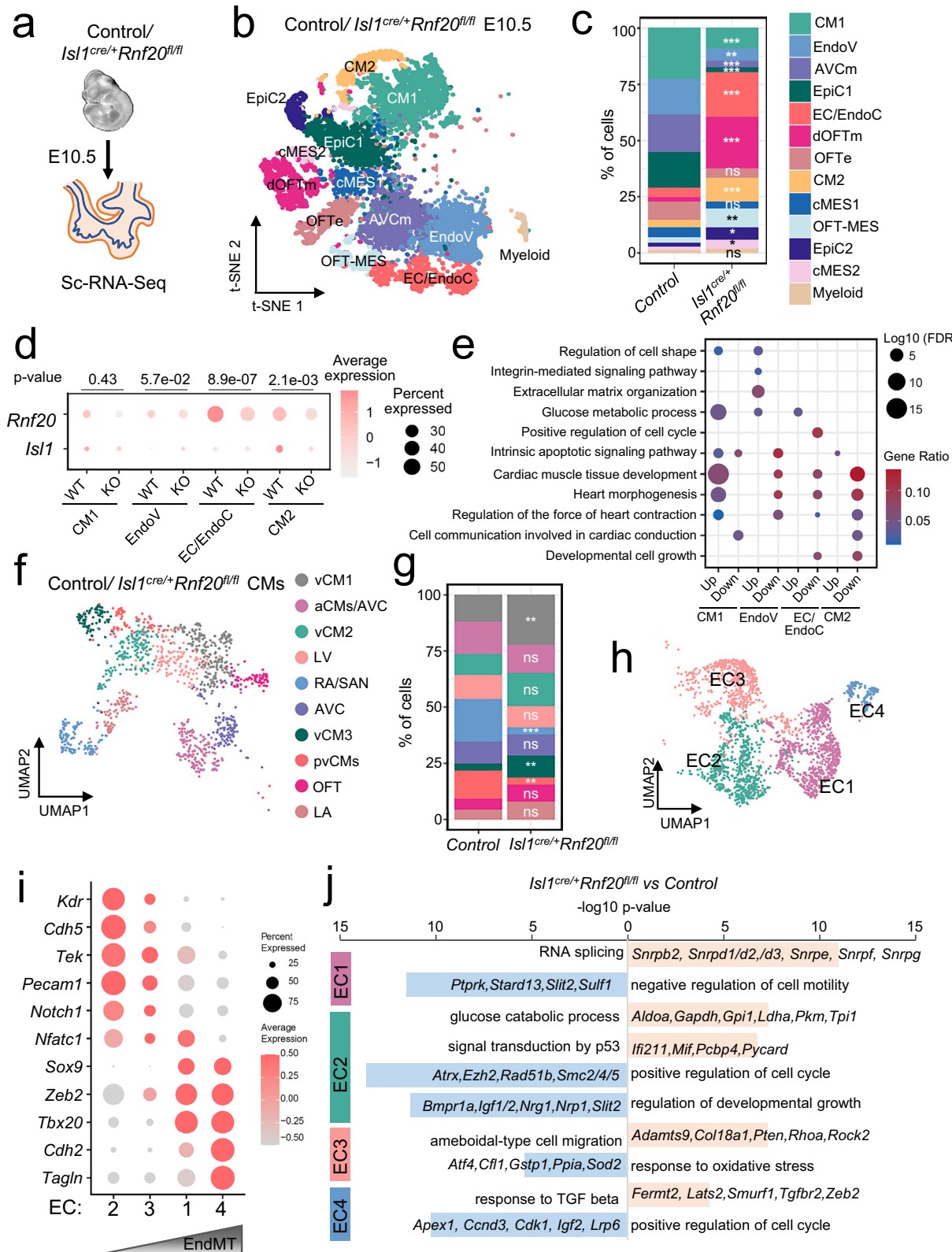

Supplementary Fig. 5c). Tamoxifen induction was performed at E8.5, when Isl1-Cre activity is known to be high. *Rnf20* mRNA levels (Fig. 4b) and H2Bub1 levels (Supplementary Fig. 5d) were significantly decreased specifically in endothelial cells, confirming efficient gene ablation following administration of tamoxifen from E8.5. Similarly to *Rnf20* deficiency in SHF progenitors and their derivatives, analysis of

image sequences recorded on E10.5 control hearts and hearts with endothelial deletion of Rnf20 (*Rnf20*[iEC-KO]) revealed highly arrhythmic beating with decreased amplitude and frequency compared to control embryos (Fig. 4c-e); one out of seven (1/7) control hearts showed arrhythmic beating behavior, while seven out of nine (7/9) *Rnf20*[iEC-KO] hearts exhibited arrhythmic contractility (Supplementary Fig. 5e,

**Fig. 3 | Conditional deletion of *Rnf20* in Isl1-derived cells alters cardiac cellular composition. a** Schematic representation of the heart dissection procedure used for scRNA-Seq. Hearts from five control and *Isl1^cre/+Rnf20^fl/fl* embryos (*n* = 5) were dissected for the experiment. **b** t-Stochastic neighbor embedding (t-SNE) plot of 10x genomics scRNA-Seq datasets of E10.5 control and *Isl1^cre/+Rnf20^fl/fl* hearts. CM1 and CM2, cardiomyocytes 1 and 2; EndoV endocardial valve cells, AVCm atrio-ventricular canal mesenchyme, EpiC1 and EpiC2 epicardial cells 1 and 2; EC/EndoC, endothelial cells/endocardial endothelial cells; dOFTm, dorsal outflow tract mesenchyme; dOFTe, dorsal outflow tract epithelium; cMES1 and cMES2, cardiac mesenchyme 1 and 2; OFT-MES, outflow tract mesenchyme; Myeloid cells. **c** Percentage of cells within the different clusters colored by cell types. *P*-values were calculated using a two-sided exact binomial test. **d** Dot plot showing *Rnf20* and *Isl1* expression in the selected cell populations from (**b**), separated by genotype. *P*-values were determined using the two-sided Wilcoxon rank-sum test provided by FindMarkers. **e** Dot plot depiction of representative gene ontology

terms in genes up- and downregulated in the different CM and EC populations of *Isl1^cre/+Rnf20^fl/fl* compared to control hearts. Uniform Manifold Approximation and Projection (UMAP) plot showing the re-clustering of CM1 and CM2 from panel b (**f**) and bar plot showing the percentage of cells within the different clusters colored by cell type (**g**). *P*-values were calculated using a two-sided exact binomial test (*\**p* < 0.05, **\**p* < 0.01, ***\**p* < 0.001). vCM1-3 ventricular cardiomyocytes 1-3, aCMs/AVC, atrial/ atrioventricular canal CMs, LV, left ventricle, RA/SAN, right atrium/sinoatrial node, pvCMs proliferative ventricular CMs, OFT outflow tract, LA left atrium. **h** UMAP analysis of EndoV, EC/EndoC and OFT-MES populations from panel b, presenting distinct EC populations. **i** Dot plot showing the expression of different endothelial and EndMT markers. **j** Representative GO terms in genes upregulated (light red) and downregulated (light blue) in the different EC populations in (**i**). *P*-values were calculated using a hypergeometric test as performed by Metascape (v.3.5).

Supplementary Table 3). Further, endothelial deletion of Rnf20 resulted in thinner compact and trabeculae layer, impaired interventricular septum (IVS) development at E11.5 (Fig. 4f-h, Supplementary Fig. 5f) and lethality by E14.5. Interestingly, in contrast to wild-type embryos, in which endocardial ECs typically show elongated nuclei and thin cell bodies lining the myocardium, Rnf20-depleted endocardial cells appeared disorganized and did not line the trabeculae, similar to the *Isl1^cre/+Rnf20^fl/fl* hearts (Fig. 4f, lower panels). We next studied whether the observed ventricle hypoplasia could be due to altered CM proliferation. Indeed, immunostaining for EdU incorporation in combination with the CM marker MF20 revealed a significant decrease in CM proliferation, while EC proliferation was not affected (Fig. 4i, j). TUNEL assays revealed no apoptotic cells in the hearts of both control and *Rnf20^iEC-KO* at E11.5 (Supplementary Fig. 5g), suggesting that the decreased cell number is due to decreased cell proliferation but not cell death.

Given that Rnf20 levels were reduced in patients with CHD with decreased SpO2 levels, we next studied the effect of endothelial-specific deletion of *Rnf20* during the postnatal period (Fig. 4k-n). Consistent with the phenotype observed in embryonic development, ablation of Rnf20 in ECs at P1 resulted in lethality after P7, a shift from the typical elongated and flattened morphology to a disorganized, morphologically disordered phenotype (Fig. 4l, Supplementary Fig. 5h, i), and abnormal sarcomeric organization (Fig. 4m, Supplementary Fig. 5j). Although we did not observe differences in capillary density, we detected significant decrease in CM proliferation (Fig. 4n).

To further study the molecular mechanisms underlying RNF20's function in ECs, we performed single-cell RNA-Seq of E11.5 control and *Rnf20^iEC-KO* embryos (Fig. 5a, b, Supplementary Fig. 6a, b, Supplementary Data 3). Similar to *Isl1^cre/+Rnf20^fl/fl* hearts, we detected an increased number of outflow tract mesenchymal cells (Fig. 5c). From the two CM populations detected, one showed a major decrease in cell numbers upon RNF20 loss (Fig. 5c). GO analysis revealed that genes downregulated in CMs were linked to heart morphogenesis, contraction, sarcomere organization, as well as cell junction assembly (Fig. 5d). Genes downregulated in EC/EndoC cells were associated with actin filament organization, homeostasis of cell number, heart morphogenesis, and endothelial cell differentiation, while upregulated genes were involved in the cellular response to environmental stimulus and fibroblast proliferation.

We next re-clustered the CM and EC populations to investigate alterations in Rnf20-deficient ECs in greater detail. We observed a dramatic decrease in proliferative ventricular CMs, while atrial CM number was increased, suggesting an important function of endothelial Rnf20 in ventricular cardiomyocyte proliferation (Fig. 5e-g, Supplementary Fig. 7a, b). Consistently, *Mki67*-proliferating ventricular CMs were much less in the *Rnf20^iEC-KO* mutants (Fig. 5f).

Re-clustering of EC identified five clusters; three clusters, expressing high levels of the EC marker genes *Kdr* and *Pecam1*, such as Vascular Endothelial Cells (VECs), EndoC and proliferative EndoC, and two expressing lower levels of *Kdr*, such as EndoV1 and EndoV2 (Fig. 5h-j Supplementary Fig. 7c-e). We observed a significant decrease in the EndoV1 population characterized by expression of EndMT-associated genes such as *Tbx20*, *Tgfb2*, and *Twist1* in the *Rnf20^iEC-KO* mutant hearts. In contrast, there was a marked increase in a distinct EndoV2 subpopulation co-expressing *Tgfb2* and *Twist1*, along with mesenchymal and structural genes, including *Cdh2* (N-cadherin), *Tagln* (SM22α), and various collagens such as *Col1a1*. This shift suggests a progression toward a more advanced EndMT state, marked by the acquisition of mesenchymal identity and extracellular matrix remodeling capacity. GO analysis revealed that, common upregulated genes were linked to signal transduction by p53 and erythrocyte homeostasis, while genes downregulated upon Rnf20 LOF were associated with regulation of developmental growth, cell-substrate adhesion, as well as response to growth factor stimulus, such as *Nrg1*, *Nrp1*, *Igf2*, similar to *Isl1^cre/+Rnf20^fl/fl* hearts (Fig. 5k, Supplementary Fig. 7f, Supplementary Data 4).

**Rnf20 inhibits endothelial-to-mesenchymal transition (EndMT)**

Histological analysis revealed that endocardial ECs show mesenchymal characteristics, while our *Isl1^cre/+Rnf20^fl/fl* and *Rnf20^iEC-KO* scRNA-Seq data identified an increased number of mesenchymal cells and higher expression of ECM genes, suggesting that Rnf20 might inhibit EndMT. To test the ability of *Rnf20*-deficient endocardial ECs to undergo EndMT, we performed a widely used cushion EndMT assay with microdissected OFT from E10.5 hearts cultured on collagen gel. We observed significantly more spindle-shaped cell with longer filopodia arising from *Isl1^cre/+Rnf20^fl/fl* and *Rnf20^iEC-KO* compared to control explants (Fig. 6a, b). Additionally, *Rnf20*-deficient HUVECs showed increased migratory behavior (Fig. 6c, Supplementary Fig. 8a) and decreased tube formation (Supplementary Fig. 8b), two hallmarks of EndMT, supporting the role of Rnf20 in inhibiting EndMT.

Given the upregulation of TGF-β signaling-associated genes in ECs upon Rnf20 loss, we next assessed pathway activation by staining for phosphorylated SMAD2 and SMAD3. We observed a significant increase in the number of p-SMAD2/p-SMAD3-positive cells in HUVECs following *RNF20* silencing (Fig. 6d). Notably, silencing *SMAD2* and *SMAD3* reversed the enhanced migratory capacity of RNF20-deficient HUVECs to control levels (Fig. 6e), indicating that activation of TGF-β signaling mediates the increased cell migration induced by RNF20 loss of function.

To further investigate the role of RNF20 in endothelial cell (EC) specification and endothelial-to-mesenchymal transition (EndMT), we silenced *Rnf20* during the mesoderm stage in a directed EC differentiation protocol from mouse embryonic stem cells (mESCs) (Fig. 6f). Flow cytometric analysis for PECAM1 expression at day 7 (D7) revealed no significant difference in EC yield between control and Rnf20-

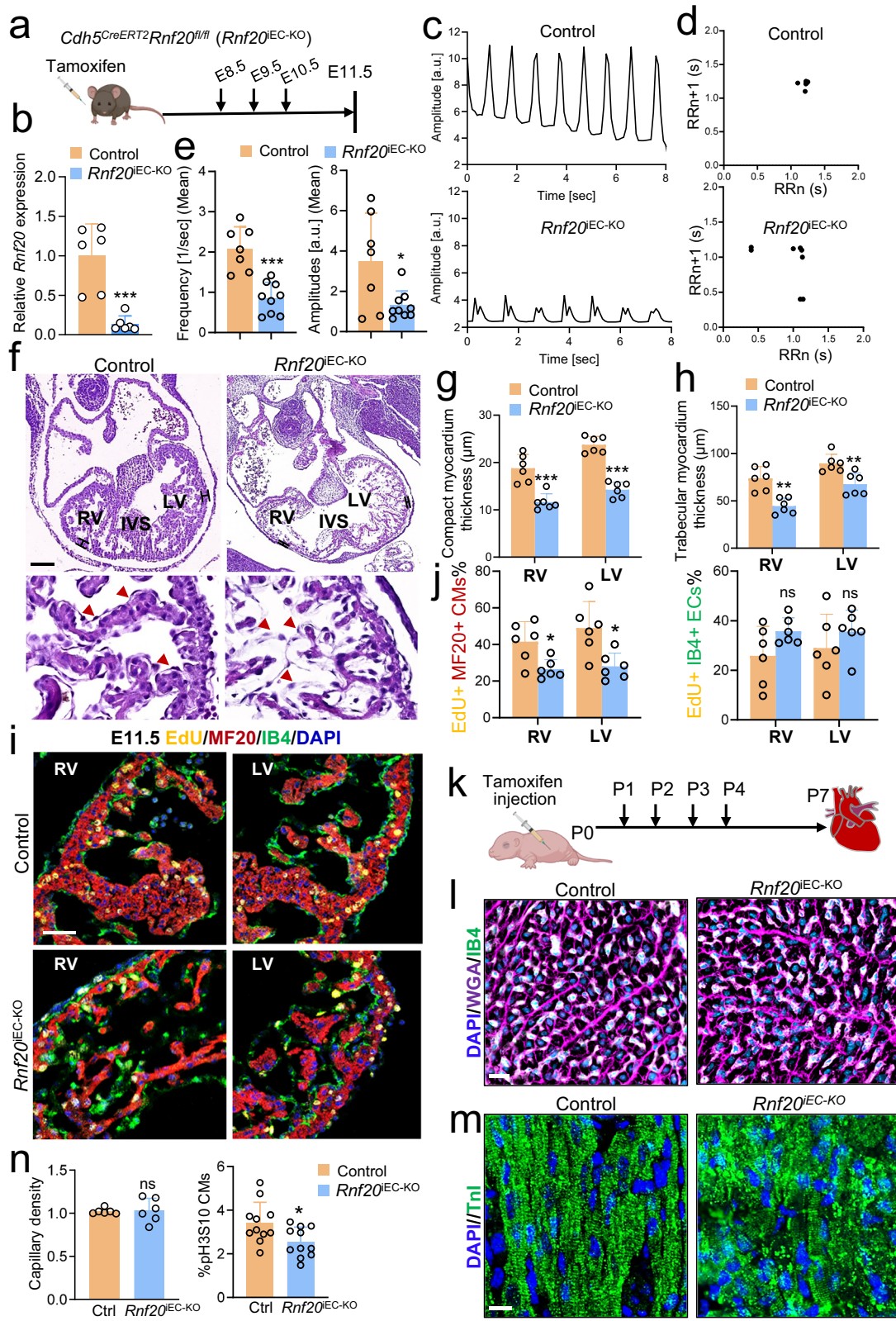

deficient cells (Fig. 6g, h), suggesting that Rnf20 is not essential for initial EC specification. However, transcriptomic analysis of sorted PECAM1+ cells showed a marked upregulation of genes associated with positive regulation of cell locomotion and extracellular matrix organization, indicative of a more migratory and mesenchymal-like phenotype (Fig. 6i, Supplementary Fig. 8c, Supplementary Data 5). Conversely, genes involved in response to growth factors and receptor

tyrosine kinase signaling were significantly downregulated, consistent with our in vivo observations. Comparison of differentially expressed genes in Rnf20-deficient mESC-derived ECs and HUVECs revealed a shared enrichment of genes promoting cell locomotion among the upregulated set, and genes regulating growth factor responses among the downregulated set, pointing to a conserved role of Rnf20 in endothelial cells (Fig. 6j).

**Fig. 4 | Endothelial RNF20 is essential for heart development and function.**
**a** Schematic representation of the experimental setup. Illustration of the mouse
created in BioRender.com. Dobreva, G. (2025) https://BioRender.com/1ea0j8q.
**b** Relative *Rnf20* mRNA expression in ECs isolated from E11.5 control (*Cdh5*[CreERT2]neg-
*Rnf20*[fl/fl], *n* = 6) and *Rnf20*[iEC-KO] (*Cdh5*[CreERT2]pos-*Rnf20*[fl/fl], *n* = 6) embryos. **c** Analysis of
image sequences recorded on control and *Rnf20*[iEC-KO] hearts using MYOCYTER.
**d** Poincaré plot derived from image sequences recorded on E11.5 control and
*Rnf20*[iEC-KO] hearts. **e** Frequency and amplitude extracted from image sequences
recorded on E11.5 control (*n* = 7) and *Rnf20*[iEC-KO] (*n* = 9) hearts using MYOCYTER.
**f** H&E staining of representative paraffin sections of E11.5 control and *Rnf20*[iEC-KO]
hearts showing a thinner compact layer, less developed trabeculae, disorganized
interventricular septum (left) and disorganized endocardial cells (red arrowheads in
the bottom right panel). LV, left ventricle; RV, right ventricle; IVS, interventricular
septum. Scale bars, 200 μm (whole heart), 50 μm (magnified regions). Quantifica-
tion of the thickness of the compact (**g**) and trabecular myocardium (**h**) in control

and *Rnf20*[iEC-KO] hearts (*n* = 6). Immunostaining with EdU, anti-MF20, anti-IB4 and
DAPI (**i**) and quantification of EdU-labeled CMs (EdU + /MF20+ cells) and EdU-
labeled ECs (EdU + /IB4+ cells) in RV and LV (**j**) of E11.5 control and *Rnf20*[iEC-KO]
embryos (*n* = 6). Scale bars, 50 μm. **k** Schematic representation of the experimental
setup for (**k–n**). Illustration of the pup created in BioRender.com. Dobreva, G.
(2025) https://BioRender.com/q0n71hy. **l** Representative Image of P7 control and
*Rnf20*[iEC-KO] heart sections co-stained with IB4, WGA and DAPI. Scale bars, 20 μm.
**m** Representative Image of P7 control and *Rnf20*[iEC-KO] heart sections co-stained with
DAPI and Troponin I. Scale bars, 20 μm. **n** Capillary density (ratio of the number of
CM versus ECs) (left panel, *n* = 6 per group) and quantification of CMs positive for
the mitotic marker phosphorylated-H3S10 (right panel, *n* = 11 per group) in control
and *Rnf20*[iEC-KO] hearts. Statistical analysis between two groups in (**b, e, g, h, j, n**) was
performed using an unpaired two-tailed Student's *t* test. Data are shown as
mean ± SEM. *$p < 0.05$, **$p < 0.01$, ***$p < 0.001$.

By day 14 of directed EC differentiation, control ECs formed
organized monolayers with robust VE-Cadherin (Cdh5) expression,
whereas Rnf20-deficient cells displayed disrupted monolayer integrity
and a substantial loss of VE-Cadherin expression (Fig. 6k). Notably,
many of these cells expressed the mesenchymal marker Snai1, further
supporting aberrant induction of EndMT in the absence of Rnf20.
Together, these findings indicate that while Rnf20 is dispensable for
early EC specification, it is essential for maintaining endothelial iden-
tity and preventing aberrant mesenchymal transition.

### Rnf20 inhibits nonphysiological angiocrine signaling promoting cell cycle withdrawal and abnormal contractility

We also observed a significant decrease in CM proliferation in *Isl1*[cre/
+]*Rnf20*[fl/fl] and *Rnf20*[iEC-KO] hearts, along with a decrease in the expression
of genes involved in developmental growth, regulation of heart con-
traction and cellular communication. To study the role of Rnf20 in ECs
in controlling CM behavior without the potential confounding side
effects of altered tissue perfusion in malformed hearts, we established
a co-culture system (Fig. 7a). We isolated cardiac ECs from control and
*Rnf20*[iEC-KO] hearts and, after 4-OHT addition for three days to induce
*Rnf20* ablation, co-cultured them with rat cardiomyocytes in a trans-
well indirect co-culture system. Rat CMs co-cultured with *Rnf20*[iEC-KO]
ECs exhibited significantly reduced contraction amplitude and fre-
quency, along with arrhythmic beating patterns (Fig. 7b-d, Supple-
mentary Table 4). In addition, Rnf20 loss led to a marked decrease in
the percentage of actively replicating EdU-positive cardiomyocytes,
indicating impaired proliferative capacity (Fig. 7e). Similar results we
observed in co-cultures of rat CMs with HUVECs (Fig. 7f, g). *RNF20*
silencing in HUVECs reduced the percentage of mitotic phospho-
Histone H3 (Ser10)-positive cardiomyocytes, while RNF20 over-
expression enhanced CM proliferation (Fig. 7g). Analysis of movies of
spontaneously beating CMs using MYOCYTER[55] revealed that silencing
of *RNF20* resulted in arrhythmic contractions (Fig. 7h, i) as well as
significantly reduced contraction amplitude and frequency (Fig. 7j),
similar to the phenotype observed in vivo, while overexpression of
RNF20 improved rhythmicity. These findings suggest that the
observed in vivo phenotypes are a direct consequence of *Rnf20* loss
in ECs.

As we observed a major impact of RNF20 LOF in ECs on CMs and
genes involved in cell-cell communication and ECM organization were
highly deregulated in *Isl1*[cre/+]*Rnf20*[fl/fl] and *Rnf20*[iEC-KO] hearts, we next
performed ligand-receptor analysis using NicheNet[61] to pinpoint the
angiocrine signals that may result in cardiomyocyte dysfunction
(Fig. 8a). Consistent with the cellular phenotype, *Nrg1*, *Nrp1*, *Igf1* and
*Igf2*, angiocrines with key roles in CM proliferation during cardiogen-
esis were decreased, while *Nppa*, *MIF*, *Tgfb1/2* as well as a number of
collagens were increased. Importantly, *RNF20* levels in TOF patients
showed a significant negative correlation with genes involved in
supramolecular fiber organization, including TGFB1 and other

canonical TGF-β target genes, such as *TGFB1I1*, *TAGLN*, *LOXL1*, col-
lagens, as well as *NPPA* and *NPPB* in CHD patients with decreased levels
of *RNF20* (Fig. 8b, Supplementary Fig. 9, Supplementary Data 6), fur-
ther supporting a function of the increased TGF-ß signaling in CHD
pathogenesis upon *RNF20* loss.

Next, we analyzed whether modulation of altered pathways and
angiocrines could rescue the decreased CM proliferation and
arrhythmic behavior (Fig. 8c). EdU labeling followed by immunos-
taining revealed a significant increase of CM proliferation back to
control levels by addition of NRG1. A similar rescue was observed
following silencing of *SMAD4*, central mediator of the TGF-β super-
family pathway, as well as silencing of SMAD2/3 (TGF-β pathway
effectors) or combination of collagens in RNF20-deficient HUVECs
(Fig. 8d, e). In contrast, addition of IGF2 or silencing the multi-
functional cytokine MIF did not improve CM proliferation (Fig. 8d,
Supplementary Fig. 10a).

Next, we examined how altered signaling pathways and angio-
crine factors affect CM contractility and rhythmicity (Fig. 9a). Func-
tional analysis revealed that silencing *SMAD2* and *SMAD3* in RNF20-
deficient HUVECs improved contraction amplitude, frequency, and
rhythmicity (Fig. 9b–d, Supplementary Data 7). Similarly, knockdown
of NPPA or collagen combinations enhanced rhythmicity and/or
contractile function (Fig. 9e–g, Supplementary Data 7). In contrast,
overexpression of NPPA or MIF induced pronounced arrhythmic
beating patterns (Supplementary Fig. 10b–d), highlighting their
potential contribution to CM dysfunction downstream of RNF20
function in ECs.

Together, these data support a key function of Rnf20 in endo-
thelial cells for the control physiological angiocrine signaling
instructing CM proliferation and rhythmicity.

### Rnf20 inhibits RNA polymerase II pause release at TGF-β targets by preventing TFIIS binding, while it stimulates growth factor gene expression through H2Bub1

Since RNF20 depletion in cardiac ECs and HUVECs revealed similar
effects on CM behavior, we next studied the mechanism behind the
transcriptional changes in RNF20-deficient ECs using ChIP-sequencing
for H2Bub1 and Pol II in HUVECs transfected with control siRNA and
siRNA against *RNF20*[41]. Genome-wide H2Bub1 levels were decreased in
*RNF20*-deficient HUVECs. However, the relative decrease of H2Bub1
levels was more prominent at genes downregulated in both *RNF20*-
knockdown HUVECs and *Rnf20*-deficient ESC-derived ECs (Fig. 10a, b).
These genes exhibited comparable levels of H2Bub1 enrichment at
both the promoter and gene body, consistent with active transcription
and efficient elongation. Among the genes showing significantly
reduced H2Bub1 levels were those involved in cholesterol metabolism
(e.g., *LDLR*), focal adhesion (*ZYX*), and growth factor signaling (*NRG1*
and *IGF2*), consistent with their transcriptional downregulation fol-
lowing *Rnf20* loss in ECs (Fig. 10c). VEGFC, which is typically

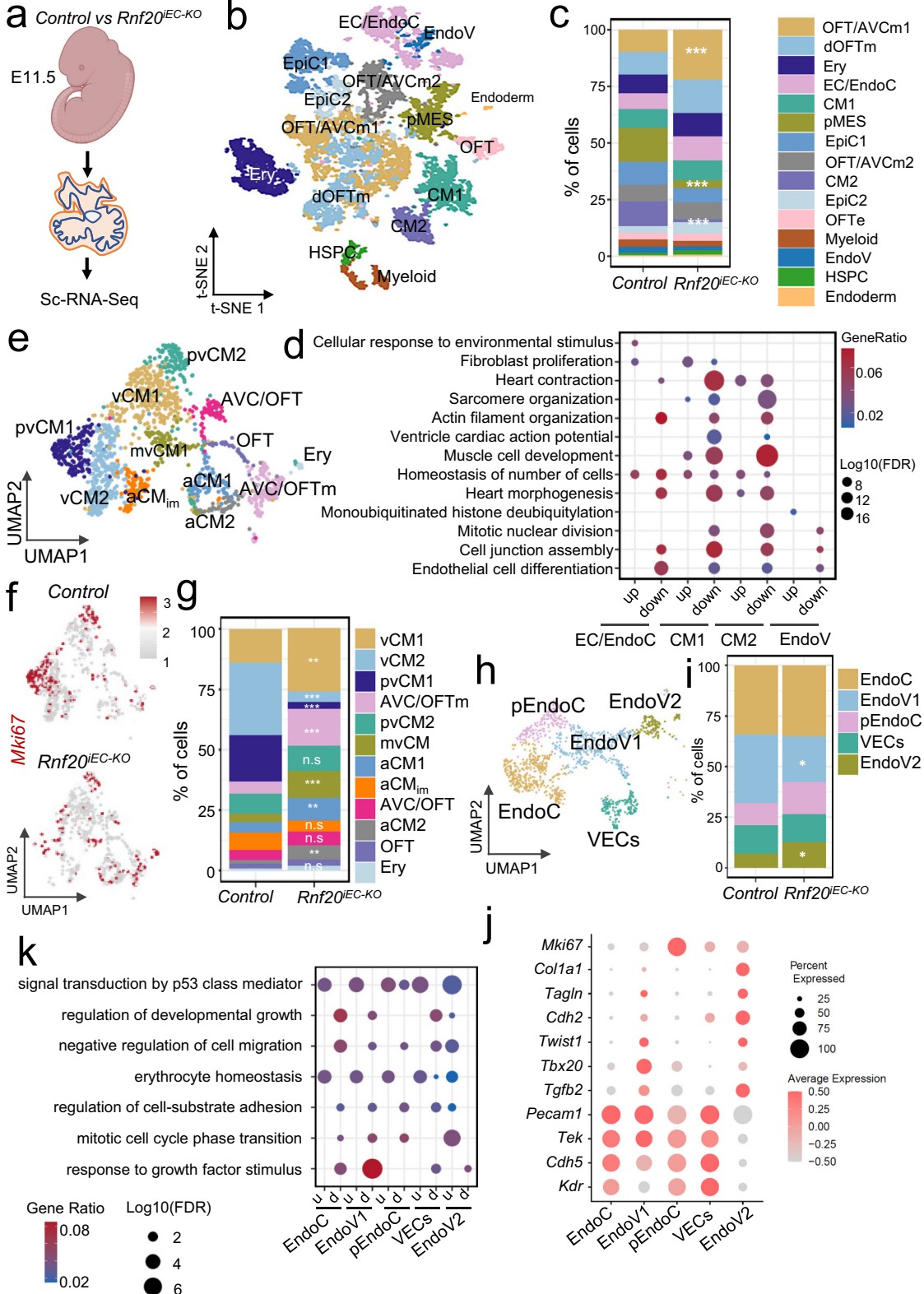

upregulated upon *RNF20* depletion in ECs[41], did not exhibit any significant changes in H2Bub1 levels.

Prior studies have demonstrated that Rnf20 modulates transcriptional pausing at genes exhibiting high promoter-proximal pausing in ECs[41]. Interestingly, by comparing the changes of the pausing index upon *RNF20* LOF at bona-fide Notch, TGF-β and Wnt

pathway genes, we observed that genes involved in the TGF-β pathway were highly paused and RNF20 loss resulted in significant decrease in the Pol II pausing index (Fig. 10d). Among genes that showed decreased transcriptional pausing were *TGFB1* and *TGFBR2* (Fig. 10e). Pol II ChIP-qPCR on TGF-β pathway genes in control and RNF20-knockdown HUVECs confirmed reduced Pol II pausing upon Rnf20

**Fig. 5 | Endothelial *Rnf20* loss results in defects in EC differentiation and ventricular CM proliferation. a** Schematic representation of the heart dissection procedure used for scRNA-Seq experiments. Illustration of the E11.5 embryo created in BioRender.com. Dobreva, G. (2025) https://BioRender.com/56nm0tt. Hearts from five control and *Rnf20*[iEC-KO] embryos (*n* = 5) were dissected for the experiment. **b** t-SNE plot of 10x genomics scRNA-Seq datasets of E11.5 control and *Rnf20*[iEC-KO] hearts. OFT/AVCm1 and 2, outflow tract/atrioventricular canal mesenchyme 1 and 2; dOFTm, dorsal outflow tract mesenchyme; Ery, erythrocytes; EC/EndoC, endothelial cells/endocardial endothelial cells; CM1 and CM2, cardiomyocytes 1 and 2; pMES, proliferative mesenchyme; EpiC1 and EpiC2, epicardial cells 1 and 2; OFTe, outflow tract epithelium; EndoV, endocardial valve cells; HSPC, hematopoietic stem and progenitor cells. **c** Percentage of cells within the different clusters colored by cell types. *P*-values were calculated using a two-sided exact binomial test. **d** dot plot depiction of representative gene ontology terms in up- and downregulated genes in the different CM and EC populations of *Rnf20*[iEC-KO] versus control hearts. **e** UMAP re-clustering of CM1 and CM2 from panel b, colored by CM type. vCM1 and vCM2, ventricular cardiomyocytes 1 and 2; pvCM1-2, proliferative ventricular CM1-2; mvCM mature ventricular CM, aCM1 and aCM2, atrial cardiomyocytes 1 and 2, aCM_{im} immature atrial cardiomyocytes, AVC/OFT atrioventricular canal/outflow tract, OFT outflow tract. **f** Feature plots of *Mki67* expression in the different CM populations in (**e**). Values represent normalized gene expression counts per cell. **g** Percentage of cells in each CM subtype, color-coded by cell type. P-values were calculated using a two-sided exact binomial test. **h** UMAP analysis of Pecam1+ECs, showing five distinct EC clusters. **i** Bar plot showing the percentage of cells across different EC types. *P*-values were calculated using a two-sided exact binomial test. **j** Dot plot displaying EC and EndMT marker gene expression in the EC clusters. EndoC endocardial ECs, pEndoC proliferative endocardial ECs, EndoV1-2 endocardial valve cells 1 and 2, VECs vascular ECs. **k** Representative GO terms in up- and downregulated genes in *Rnf20*[iEC-KO] in the different EC populations in (**h**).

---

depletion (Fig. 10f), while qPCR analysis revealed a corresponding upregulation of these genes (Fig. 10g).

RNF20 was shown to hinder the recruitment of TFIIS, which is required for the release of RNA polymerase II (Pol II) into active elongation at tumor-promoting genes[40]. To investigate whether RNF20 blocks TFIIS binding at TGF-β pathway genes, we performed TFIIS ChIP-seq in control and RNF20-knockdown HUVECs. Notably, TFIIS binding to TGF-β targets was significantly increased upon RNF20 depletion, suggesting a critical role for RNF20 in restraining TGF-β signaling by limiting TFIIS recruitment and function (Fig. 10h). Therefore, while RNF20-dependent decrease in H2Bub1 levels might be responsible for the transcriptional downregulation of RNF20-target genes, its function in promoting Pol II pausing is responsible for the elevated expression of genes involved in TGF-β signaling and response.

## Discussion

In this study, we discovered that RNF20 plays a central role in cardiac endothelial cell for heart development and CHD by maintaining EC identity and safeguarding physiological angiocrine signaling essential for proper cardiomyocyte proliferation and rhythmicity (Fig. 10i). Mechanistically, RNF20 exerts these effects by maintaining the critical balance between TGF-β and growth factor signaling during heart development. Loss of RNF20 led to enhanced TGF-β signaling in ECs, which is central to the induction of EndMT and cushion formation[9,19,62]. This was linked to RNF20's function in hindering TFIIS binding, thereby promoting Pol II promoter-proximal pausing. Importantly, bona-fide TGF-β pathway genes exhibited a higher pausing index compared to Notch and Wnt targets, and RNF20 deficiency resulted in significant Pol II release specifically on TGF-β targets. scRNA-Seq analysis and phenotypic characterization of SHF- and EC-specific *Rnf20* deletion revealed expansion of mesenchymal populations and enhanced outgrowth in cushion EndMT and wound healing assays, in line with elevated TGF-β signaling. A large number of upregulated genes upon Rnf20 LOF are ECM proteins, which are also upregulated in response to TGF-β signaling. Our data revealed that silencing critical components and downstream targets of TGF-β signaling, such as *SMAD2-4*, as well as a set of collagens, rescued the decreased CM proliferation and arrhythmic contractility upon RNF20-deficiency. Interestingly, collagens secreted from cardiac fibroblasts are required for HBEGF- and FGF2-induced cardiomyocyte proliferation[63], while excessive collagen production from RNF20-deficient ECs inhibited proliferation. Collagens are crosslinked through lysyl oxidases (LOX, LOXL), which increases their stiffness. Together with upregulation of *Col1a1/2*, *Col3a1*, *Col5a1/2/3*, *Col12a1*, lysyl oxidases were significantly upregulated upon Rnf20 LOF and in CHD patients, potentially contributing to increased cardiac stiffness beyond the physiological level and thereby limiting CM proliferation[64]. To what extent changes in biophysical and biochemical properties of the ECM impact the heterocellular crosstalk in the heart would be an important future direction to pursue.

Secondly, we discovered that RNF20 safeguards endothelial cell identity and provides crucial angiocrine signals essential for myocardial trabeculation, cardiomyocyte proliferation, and rhythmic contractility. Ablation of *Rnf20* using both SHF- and EC-specific Cre drivers led to a marked reduction in the expression of angiocrines known to promote cardiomyocyte proliferation, such as *Nrg1*, *Igf1* and *Igf2*. Similarly, genes involved in growth factor signaling were significantly downregulated in both mESC-derived ECs and HUVECs, indicating a conserved role for RNF20 in regulating their expression. *Nrg1*, downstream of endocardial Notch signaling[20], was also reduced. In a previous work, we demonstrated that RNF20 binds to the Notch Intracellular Domain (NICD) and controls Notch target gene expression via H2Bub1[41], suggesting that the decreased H2Bub1 at the *Nrg1* locus may result at least in part from impaired Notch signaling. *Nrg1* knockout mice do not survive past mid-embryogenesis (E10.5) due to defects in ventricular trabeculation[11,65]. Conversely, injecting NRG1 into the developing heart leads to hypertrabeculation of the ventricles[66]. Studies in isolated cardiomyocytes further revealed that NRG1 addition induces cell survival, hypertrophy and proliferation and is instrumental for cardiac repair and regeneration[67–69]. IGF1 and IGF2 are also secreted by the endocardium and acts together with NRG1 to promoter cardiomyocyte proliferation and morphogenesis[66]. Thus, the decreased NRG1 and IGF1/2 levels might be responsible for the reduced proliferation and defects in ventricular trabeculation caused by endothelial-specific Rnf20 loss. Supplementation of NRG1, but not IGF2 in co-cultures of wild-type CMs with Rnf20-deficient ECs rescued CM proliferation to the levels observed in co-cultures with control ECs, suggesting a more dominant role of NRG1 in mediating the RNF20 LOF phenotype; however, we cannot exclude the possibility of developmental stage-specific involvement of IGF2.

Compared with our previous study, which showed that RNF20 regulates VEGF-Notch signaling to control endothelial specification and vessel growth[41], the current work highlights its broader role in preserving endothelial identity and directing the instructive function of the endothelium to support heart morphogenesis and contractility by restraining TGF-β signaling and modulating angiocrine communication. Mechanistically, we observed a distinction in how RNF20 regulates VEGF and TGF-β signaling: at VEGF-signaling genes, it limits ERG activation and ERG-dependent Pol II pause release, whereas at TGF-β target genes, it restrains Pol II elongation by blocking recruitment of TFIIS, a factor essential for pause release.

While EndMT and disruption of angiocrine signaling were observed in both endothelial and SHF lineages upon RNF20 loss, only SHF-specific deletion resulted in marked reductions in right atrial and SAN cells. Given that the SAN serves as the heart's primary pacemaker by initiating and regulating electrical impulses that govern cardiac rhythmicity, these findings suggest that RNF20 may have a direct role in modulating SAN development and function. On the other hand, the more severe phenotype observed upon endothelial-specific ablation of

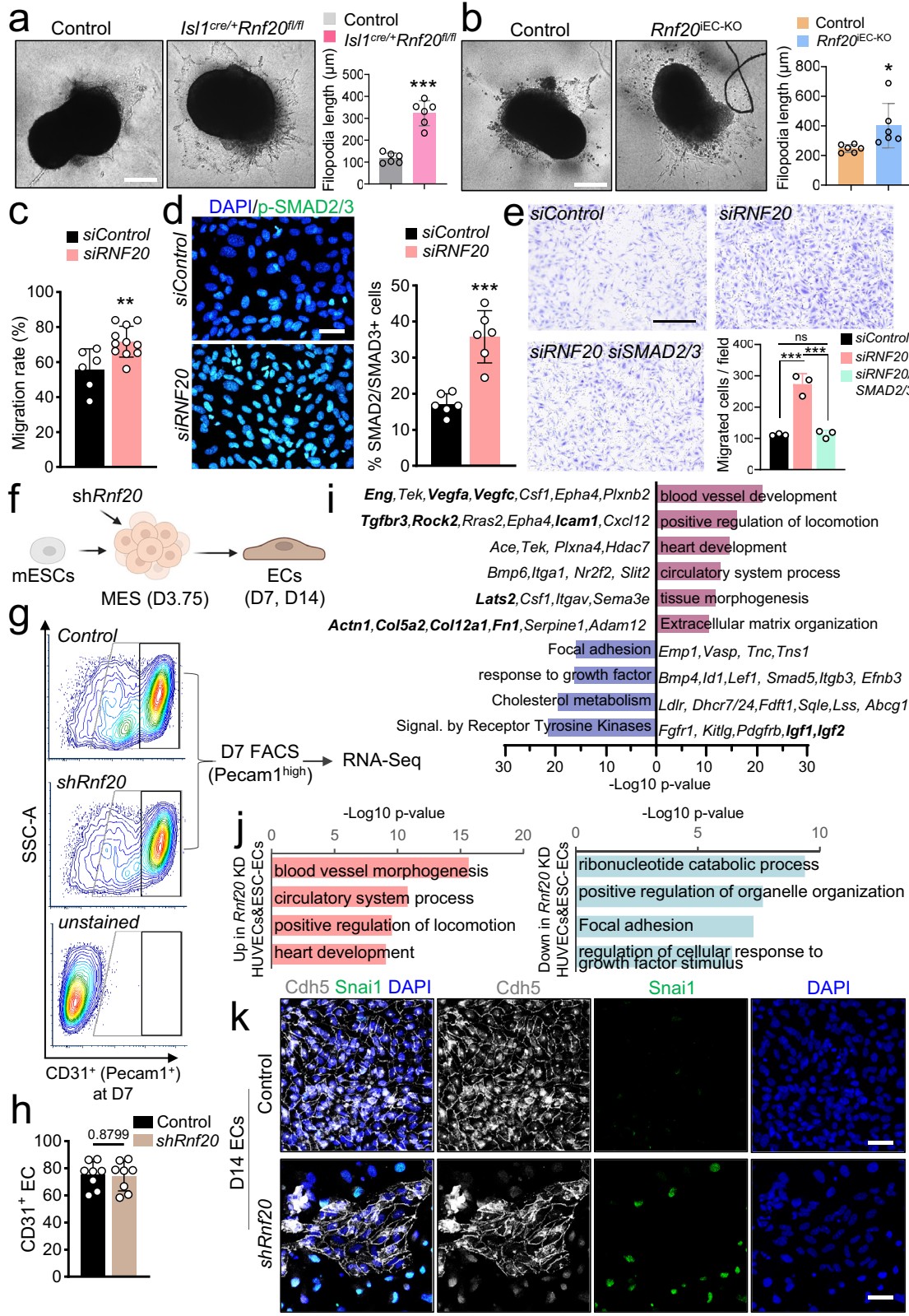

*Rnf20* might be due to its critical role in regulating EC state and function[41], not only in cardiac endothelial cells but also across the entire endothelium, which plays an instrumental role in tissue perfusion, and paracrine signaling essential for organogenesis.

Finally, we found that *RNF20* levels positively correlate with oxygen saturation and is significantly reduced in HUVECs upon exposure to hypoxia. Notably, RNF20 regulates HIF1α protein levels[42], while

hypoxic conditions and elevated HIF1α may in turn influence RNF20 expression or activity, suggesting a potential feedback loop relevant to cardiac development. Non-physiological hypoxia during early pregnancy, a known risk factor for CHDs, suppresses Isl1 expression via the HIF1α−HES1−SIRT1 axis[70]. Given that hypoxia can also enhance TGFβ signaling, this crosstalk may further disrupt key developmental gene programs required for normal cardiogenesis. Importantly, key

**Fig. 6 | Endothelial RNF20 inhibits EndMT.** Representative images of explants of proximal OFT from control ($n = 6$) and $Isl1^{cre/+}Rnf20^{fl/fl}$ ($n = 6$) embryos (**a**), and control ($n = 6$) and $Rnf20^{iEC-KO}$ ($n = 6$) embryos (**b**) at E10.5 cultured on collagen gel (left panels) and quantification of the filopodia length of outgrowing mesenchymal cells (right panel). Scale bars, 500 μm. **c** Quantification of migration rate in scratch assay in HUVECs transfected with control siRNA ($n = 6$) or siRNA against $RNF20$ ($n = 11$). **d** Immunostaining with anti-pSMAD2/3 and DAPI (**d**) and quantification of pSMAD2/3-positive cells in HUVECs transfected with control siRNA ($n = 6$) or $RNF20$ siRNA ($n = 6$). Scale bars, 20 μm. **e** Boyden chamber migration assay and quantification of migrated HUVECs transfected with control siRNA ($n = 3$), siRNA against $RNF20$ ($n = 3$) or $siRNF20$ together with siRNAs against SMAD2 and SMAD3 ($siRNF20/siSMAD2/3$) ($n = 3$). Scale bars, 500 μm. **f** Schematic representation of the directed mESC differentiation system into ECs with lentiviral transduction of a control and shRnf20 construct at the mesoderm stage (mESC, mouse embryonic stem cell; MES, mesoderm). Created in BioRender. Dobreva, G. (2025) https://BioRender.com/n7zzqua. **g** Representative FACS analyses of Pecam1⁺ (CD31⁺)

endothelial cells at day 7 of differentiation depicted in (**f**). Highly CD31-positive cells were FACS sorted for RNA-seq. **h** Percentage of Pecam1⁺ ECs at day 7, differentiated from ESCs transduced with control ($n = 8$) or shRnf20 construct ($n = 8$) at mesoderm stage. **i** Representative GO terms of genes enriched among significantly upregulated (violet) and downregulated (blue) genes in RNA-seq datasets of sorted Pecam1⁺ $shRnf20$ ECs versus control differentiated ECs ($n = 2$; padj <0.05). **j** Representative GO terms enriched among shared significantly upregulated (red) and downregulated (blue) transcripts (padj <0.05 for both datasets) upon RNF20 depletion in both HUVECs ($n = 3$) and mouse ESC differentiated ECs ($n = 2$). P-values were calculated using a hypergeometric test as performed by Metascape (v.3.5). **k** Immunostaining for Cdh5, Snai1 and DAPI of day 14 ECs derived from mESCs and representative images from three independent experiments with similar results. Scale bars, 40 μm. Statistical analysis in (**a**–**d**, **h**) was performed using an unpaired two-tailed Student's *t* test, and in (**e**) using one-way ANOVA with Šídák's correction. Data are shown as mean ± SEM. *$p < 0.05$, **$p < 0.01$, ***$p < 0.001$.

components and downstream effectors of TGF-β signaling were upregulated in TOF patients and negatively correlated with $RNF20$ expression, suggesting that targeting the RNF20-TGF-β axis may offer therapeutic benefit for patients with cyanotic CHD patients.

## Methods

### Animal studies
All animal experiments were performed according to the regulations issued by the Committee for Animal Rights Protection of the State of Hessen (Regierungspraesidium Darmstadt, Germany) and the Committee for Animal Rights Protection of the State of Baden-Württemberg (Regierungspraesidium Karlsruhe, Experimental protocol Az.: 35-9185.81/G-17/24).

### Mouse lines
The Rnf20tm1a(EUCOMM)Wtsi line was generated by microinjection of Rnf20 tm1a(EUCOMM)Wtsi ESCs, obtained from the European Conditional Mouse Mutagenesis Program (EUCOMM), into blastocysts, as detailed in ref. [41]. For the generation of a conditional (floxed) allele Rnf20tm1c(EUCOMM)Wtsi line, the Rnf20tm1a(EUCOMM)Wtsi mouse line was crossed with the germline FLP deleter mouse line. Cre deleter lines were bred to Rnf20tm1c(EUCOMM)Wtsi mice to induce specific deletion of $Rnf20$ in Isl1-positive, Tie2-positive and Cdh5-positive cells following the experimental schemes presented in the figures.

$Isl1^{Cre}$ mouse line (Isl1tm1(cre)Sev-C57BL/6)[53] was a kind gift from Sylvia Evans. The Tg(Tek-cre)12Flv line was obtained from Jackson Laboratory. The $Cdh5^{CreERT2}$ mouse line Tg(Cdh5-cre/ERT2)1Rha was obtained from Prof. Ralf Adams[60]. All mice used in this study were maintained on a C57BL/6 background.

### Cre-mediated gene ablation
To assess Cre recombination efficiency in the $Isl1^{Cre}$ mouse line, E10.5 embryos were dissected, and the left and right ventricles were separated under a stereomicroscope. Total RNA was extracted, followed by reverse transcription and qPCR using primers detailed in Supplementary Data 8. To induce endothelial-specific ablation, (Z)−4-hydroxytamoxifen (H7904, Sigma), dissolved in medium-chain triglycerides (3274, Caelo), was administered intraperitoneally to pregnant control and Rnf20iEC-KO ($Cdh5^{CreERT2}$pos-$Rnf20^{fl/fl}$) mice at a dose of 2 mg per 30 g body weight once daily for three consecutive days. To assess gene deletion efficiency, embryos were harvested at E11.5, and ECs were isolated using anti-CD31-conjugated Dynabeads via magnetic sorting, following the manufacturer's instructions. RNA extraction, reverse transcription, and qPCR were performed as described below. Primers used for qPCR analysis are listed in Supplementary Data 8.

### Human study approval and ethical standard
All studies involving human subjects were conducted in accordance with the regulations of the Heidelberg Medical Faculty of Heidelberg University, with ethical standards laid down in the vote S-157/2013 and its later amendments. Informed consents were obtained from parents/legally authorized representatives of participants.

### Cell lines and cell culture
HEK293T cells were purchased from ATCC (CRL-3216) and cultured in DMEM supplemented with 10% fetal bovine serum (FBS, Thermo Fisher Scientific, 10270106), 1% penicillin-streptomycin (Thermo Fisher Scientific, 15140122) and 2 mM L-glutamine (Thermo Fisher Scientific, 25030123).

HUVEC cells were cultured in Endothelial Cell Growth Medium MV 2 (PromoCell, C-22022).

For primary cardiac EC isolation from P4 mice, hearts were isolated and incubated for 30 min at 37 °C on a rotating shaker with 2 ml of digestion solution containing 0.05 mg/ml DNase I (Worthington, 2139) and 1 mg/ml Collagenase I (Worthington, 4197) diluted in HBSS (Thermo Fisher Scientific, 14175053) per heart until complete digestion. Reaction was stopped by the addition of EGM-MV2 medium (PromoCell, C-22022) supplemented with 1x penicillin/streptomycin and 10% FBS. Cells were centrifuged at $400 \times g$ and 4 °C for 10 min, resuspended in 1 ml of EGM-MV2 medium and incubated with 5 μl CD31(BD Biosciences, 553370)-coupled to Dynabeads (Invitrogen, 11035) for 30 min at room temperature. ECs were then captured by a magnet.

Rat CMs (from 1 to 3-day-old postnatal rats) were isolated via Percoll-Gradient and cultured in PAN BIOTECH M199 media (Biolab, P04-07500), supplemented with 10% FBS.

Murine control E14-Nkx2-5-EmGFP ESCs[71] were maintained on mitomycin (Sigma-Aldrich, M4287) treated mouse embryonic fibroblasts in KnockOut™ DMEM (Thermo Fisher Scientific, 10829018) supplemented with 10% KnockOut serum replacement (Thermo Fisher Scientific, 10828028), 2 mM L-glutamine, 0.04 mM 2-mercaptoethanol (Sigma-Aldrich, M3148), 0.1 mM non-essential amino acids (Thermo Fisher Scientific, 11140035), 1 mM sodium pyruvate (Thermo Fisher Scientific, 11360070), 4.5 mg/ml D-glucose, 1% penicillin-streptomycin and 1000 U/ml leukemia inhibitory factor (LIF ESGRO, Millipore, ESG1107).

### Directed differentiation of mESC into ECs
Embryonic stem cells (ESCs) were differentiated into endothelial cells (ECs) employing a modified protocol from ref. [72]. In brief, murine ESCs (mESCs) were dissociated and prepared feeder-free through pre-plating for 45 min. Subsequently, cells were seeded onto dishes coated with 0.1% gelatin (Sigma-Aldrich, #G9391) in a culture medium

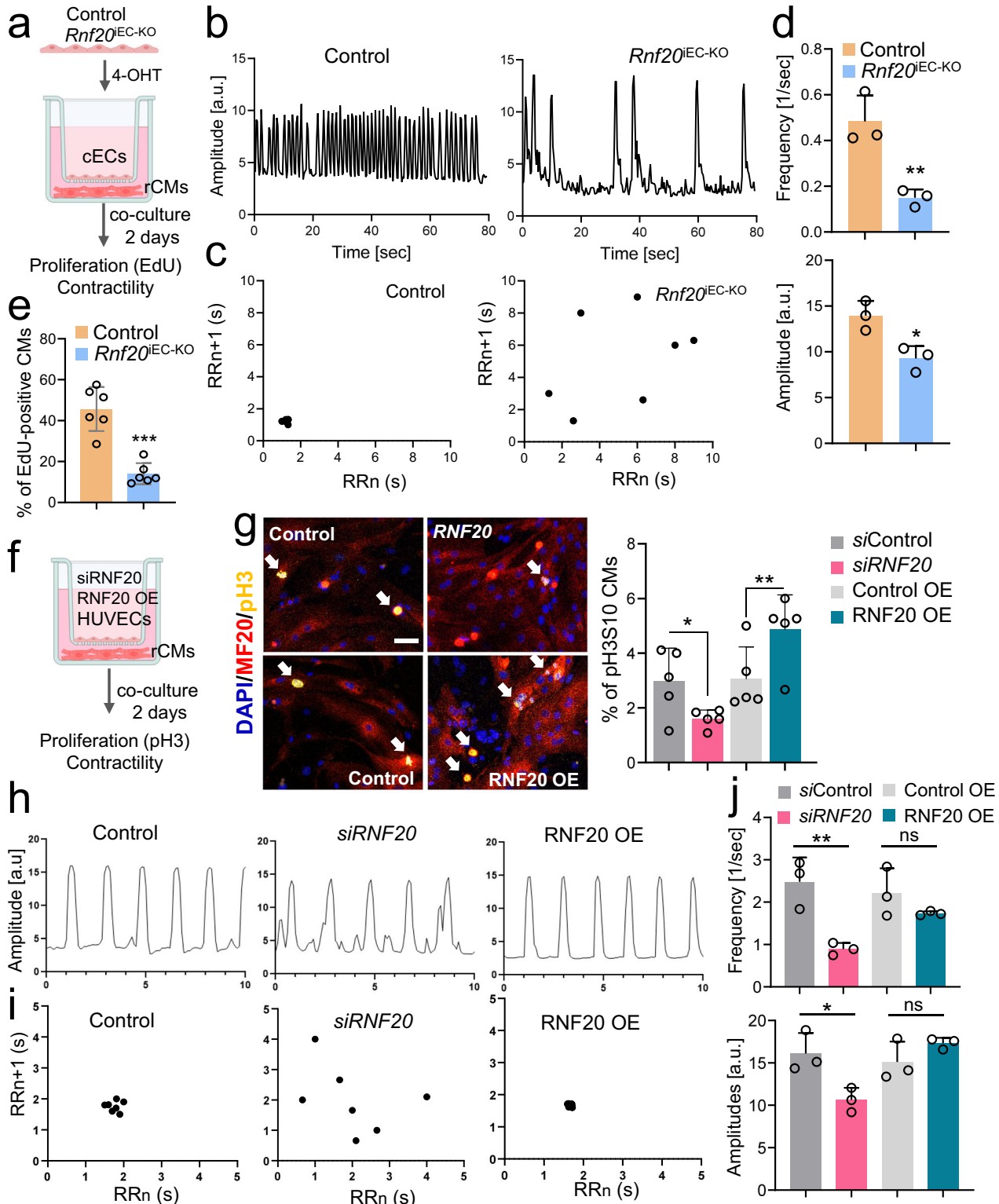

consisting of a 1:1 mixture of Neurobasal Medium (Thermo Fisher Scientific, #21103049) and DMEM F-12 Medium (Thermo Fisher Scientific, # 21331020), supplemented with 0.5X N2 Supplement (Thermo Fisher Scientific, #17502048), 0.5X B27 Supplement (Thermo Fisher Scientific, #17504044), 1X Penicillin/Streptomycin, 1X L-Glutamine, 0.05% Bovine Albumin Fraction V (Thermo Fisher Scientific, #15260037), 150 μM monothioglycerol (Sigma-Aldrich, #M6145), 10 ng/ml Bone Morphogenetic Protein 4 (BMP4, R&D Systems, #314-BP), and 2000 U/ml Leukemia Inhibitory Factor (LIF) for 3 days.

Differentiation was initiated by seeding 75,000 cells/ml into ultra-low attachment petri dishes in differentiation medium composed of 75% Iscove's Modified Dulbecco's Medium (IMDM, Thermo Fisher Scientific, #12440061) and 25% Ham's F-12 Nutrient Mix (Thermo Fisher Scientific, #21765029), supplemented with 0.5X N2 Supplement, 0.5X B27 Supplement without vitamin A (Thermo Fisher Scientific, #12587010), 1X Penicillin/Streptomycin, 50 μg/ml L-Ascorbic Acid (Sigma-Aldrich, #A4403), and 450 μM monothioglycerol. After 48 h (Day 2), embryoid bodies (EBs) were harvested, washed with PBS, and enzymatically

**Fig. 7 | Endothelial RNF20 regulates cardiomyocyte contractility and rhythmicity. a** Schematic representation of the transwell co-culture system: mouse ECs isolated from control and $Rnf20^{iEC-KO}$ hearts were treated with $10 \mu M$ 4-Hydroxytamoxifen for 72 h and subsequently plated on the top compartment of a transwell plate. 12 h beforehand rat CMs were plated in the bottom compartment and co-cultured with ECs for 2 days. **b** Plot of contraction amplitude and speed of beating rat CMs co-cultured with control or$Rnf20^{iEC-KO}$ ECs extracted from video sequences using MYOCYTER. **c** Poincaré plot derived from image sequences of beating rat CMs co-cultured with control or $Rnf20^{iEC-KO}$ ECs. **d** Frequency and amplitude of image sequences recorded on rCMs after co-culture with ECs isolated from control ($n = 6$) or $Rnf20^{iEC-KO}$ ($n = 6$) hearts. **e** Percentage of proliferating EdU-positive rat CMs co-cultured with ECs isolated from control ($n = 6$) or $Rnf20^{iEC-KO}$ ($n = 6$) hearts. **f** Schematic representation of the transwell co-culture system: HUVECs were either transfected with control siRNA or siRNA against $RNF20$ or infected with control or $RNF20$ overexpressing (OE) lentivirus for 48 h, following a

co-culture with rat CMs for 2 days. **g** Immunostaining for pH3S10 and MF20 (left panels) and percentage of mitotic pH3S10-positive rat CMs (right panel) after co-culture with HUVECs transfected with control siRNA ($n = 5$) or siRNA against $RNF20$ ($n = 5$) or infected with control ($n = 5$) of RNF20 OE lentivirus ($n = 5$). Scale bars, $50 \mu m$. **h** Plot of contraction amplitude and speed of beating of rat CMs co-cultured with control, $siRNF20$ silenced or $RNF20$ OE HUVECs extracted from video sequences using MYOCYTER. **i** Poincaré plot derived from image sequences of beating rat CMs co-cultured with control, $siRNF20$ silenced or $RNF20$ OE HUVECs. **j** Frequency and amplitude of image sequences recorded on rCMs after co-culture with control ($n = 3$), $siRNF20$ silenced ($n = 3$) or $RNF20$ OE HUVECs ($n = 3$). Transwell co-culture system schematics in (**a, f**) created in BioRender.com. Dobreva, G. (2025) https://BioRender.com/yr6530z. Statistical analysis between two groups in (**d, e, g, j**), was performed using an unpaired two-tailed Student's $t$ test. Data are shown as mean ± SEM. *$p < 0.05$, **$p < 0.01$, ***$p < 0.001$.

dissociated with 0.05% trypsin at room temperature by pipetting up and down 15 times. The single-cell suspension was counted, and cells were re-aggregated by seeding 150,000 cells/ml in differentiation medium further supplemented with vascular endothelial growth factor A (VEGFA, R&D Systems, #293-VE-010), Activin A (R&D Systems, #338-AC), and BMP4 (the concentration was adjusted for each lot).

Following a 40-h incubation period (Day 3.75; mesodermal stage), cells were dissociated as previously described and plated for EC differentiation at 300,000 cells/cm² onto dishes coated with 0.03% Matrigel (Corning, #354230) in Endothelial Cell Differentiation Medium composed of a 1:1 mixture of IMDM and F-12 Nutrient Mix, supplemented with 0.5X N2 Supplement, 0.5X B27 Supplement, 0.05% lipid-rich BSA (Thermo Fisher Sicientific, # 11020013), 50 ng/ml L-Ascorbic Acid, 50 ng/ml monothioglycerol, 50 ng/ml VEGF, 10 ng/ml bFGF, and 2 ng/ml BMP4. After 30–60 min when cells have attached, the medium was removed and control or shRnf20 lentiviral supernatants produced in EC differentiation media were added on top of the cells. Next day, viral supernatants were replaced by EC differentiation media. Medium was replaced daily until day 7. At day 7 the differentiating EC cells were characterized by FACS and RNA-Seq analysis or cultured until day 14 in EGM-MV2 (Promocell) media for further analysis.

**Flow cytometry analysis and cell sorting**
For FACS analysis, cells were enzymatically dissociated, washed with PBS, and subsequently blocked in FACS buffer (PBS supplemented with 0.4% BSA) for 5 min at ambient temperature. Cells were stained with an APC-conjugated anti-CD31 (Pecam-1) antibody (1:40, eBioscience, #17-0311-80) to identify endothelial cells (ECs). Following two washes with FACS buffer, Sytox Blue (1:2000, Invitrogen, # S34857) was applied to exclude dead cells. Flow cytometric acquisition was performed on a BD FACSAria IIu under stringent sterility and sensitivity conditions. Data analysis was carried out using FCS Express 7 software.

**Silencing using RNA interference**
siRNA-mediated knockdown in HUVECs was performed by transfection of ON-TARGETplus Human $RNF20$ siRNA (L-007027-00-0005, Horizon), ON-TARGETplus Human $COL5A2$ siRNA (L-011016-00-0005, Horizon), ON-TARGETplus Human $COL5A3$ siRNA (L-013060-00-0005, Horizon), ON-TARGETplus Human $COL3A1$ siRNA (L-011012-00-0005, Horizon), ON-TARGETplus Human $COL12A1$ siRNA (L-011834-00-0005, Horizon), ON-TARGETplus Human $NPPA$ siRNA (L-012729-00-0005, Horizon), ON-TARGETplus Human SMAD2 siRNA (L-003561-00-0005, Horizon), ON-TARGETplus Human SMAD3 siRNA (L-020067-00-0005, Horizon) or ON-TARGETplus Non-targeting Control siRNA as a control, using Lipofectamin RNAiMAX (#13778075, Thermo Fisher Scientific) according to the manufacturer's instructions. The following modifications from the manufacturer's protocol were used for transfection: 30 nM siRNA (final concentration) was transfected with 2 μl Lipofectamine RNAiMAX reagent in 1 ml of complete EGM-MV2 medium per six

well overnight. Next day cells were supplemented with fresh medium. After 60 h of transfection, cells were used for follow-up experiments (RNA-Seq, co-culture with rat CMs).

**Lentiviral transduction**
For shRNA-mediated knockdown in mESC-derived endothelial cells, HEK293T cells were used to produce lentiviral particles encoding control (pLKO) or $shRNF20$ (sequence: GCTCAGAAGAAGCTCCATGAT) obtained from The RNAi Consortium (TRC) shRNA library. HEK293T cells were seeded in 6-well plates and transfected at approximately 80% confluency. Transfection was performed using $1 \mu g$ CMVΔR8.74 packing plasmid and $0.5 \mu g$ VSV.G envelope plasmid with X-tremeGENE™ HP DNA Transfection Reagent (Roche, 6366236001) in a total of 1 ml of HEK cell medium (DMEM high glucose supplemented with 10% FCS, 1x penicillin/streptomycin, 1x GlutaMAX, 1x NEAA, 1x sodium pyruvat) overnight. Next morning, the transfection media was replaced with 2 ml EC differentiation medium without growth factors, composed of 1:1 mixture of IMDM and F-12 Nutrient Mix, supplemented with 0.5X N2 supplement, 0.5X B27 supplement, 0.05% lipid-rich BSA, 50 ng/ml L-ascorbic acid, 50 ng/ml monothioglycerol. Lentiviral supernatants were collected 48 h and 72 h post-transfection, centrifuged at $4000 \times g$ for 20 min to remove cellular debris, and the cleared supernatant was harvested. Prior to use, the lentiviral supernatant was supplemented with $8 \mu g/ml$ polybrene (final concentration) and growth factor (50 ng/ml VEGF, 10 ng/ml bFGF, and 2 ng/ml BMP4) to obtain a complete EC differentiation medium. This medium was applied to mESC-derived mesodermal cells to deliver the control and shRNA lentiviral construct during EC differentiation.

For overexpression in HUVECs, HEK293T cells were plated one day early on 6-well plate. Transfection started when the cells reached around 70–80% confluence. Overexpressing lentiviral library plasmids were transfected together with $0.975 \mu g$ CMVΔR8.74 packing plasmid and $0.525 \mu g$ VGV.G. X-tremeGENE™ HP DNA Transfection Reagent (Roche, 6366236001) was used according to the manufacturer's instructions. After 24 h, medium was changed to EGM-MV2 medium (PromoCell, C-22022). Virus-containing supernatants were harvested after 48 h and used to infect HUVECs at 60-70% confluence. Polybrene (10 mg/ml) was added 1:1000 to facilitate infection. After 24 h, cells were washed five times with PBS and used in the following experiments.

**Co-culture of ECs with CMs**
Overexpressing/silenced (40,000 cells in 24 well plates) HUVECs were co-cultured with rat CMs in a transwell chamber (Sarstedt, 83.3932.041) in complete EGM-MV2 medium (PromoCell, C-22022) supplemented with 1x penicillin/streptomycin. For the co-culture, rCMs were plated in the lower chamber and left to attach overnight. Next day HUVECs were plated in the upper chamber. After 48 h, rCMs were imaged for contractility analysis or fixed for immunofluorescence staining. ECs isolated from P4 mice, were seeded in a plate pre-coated with Collagen I

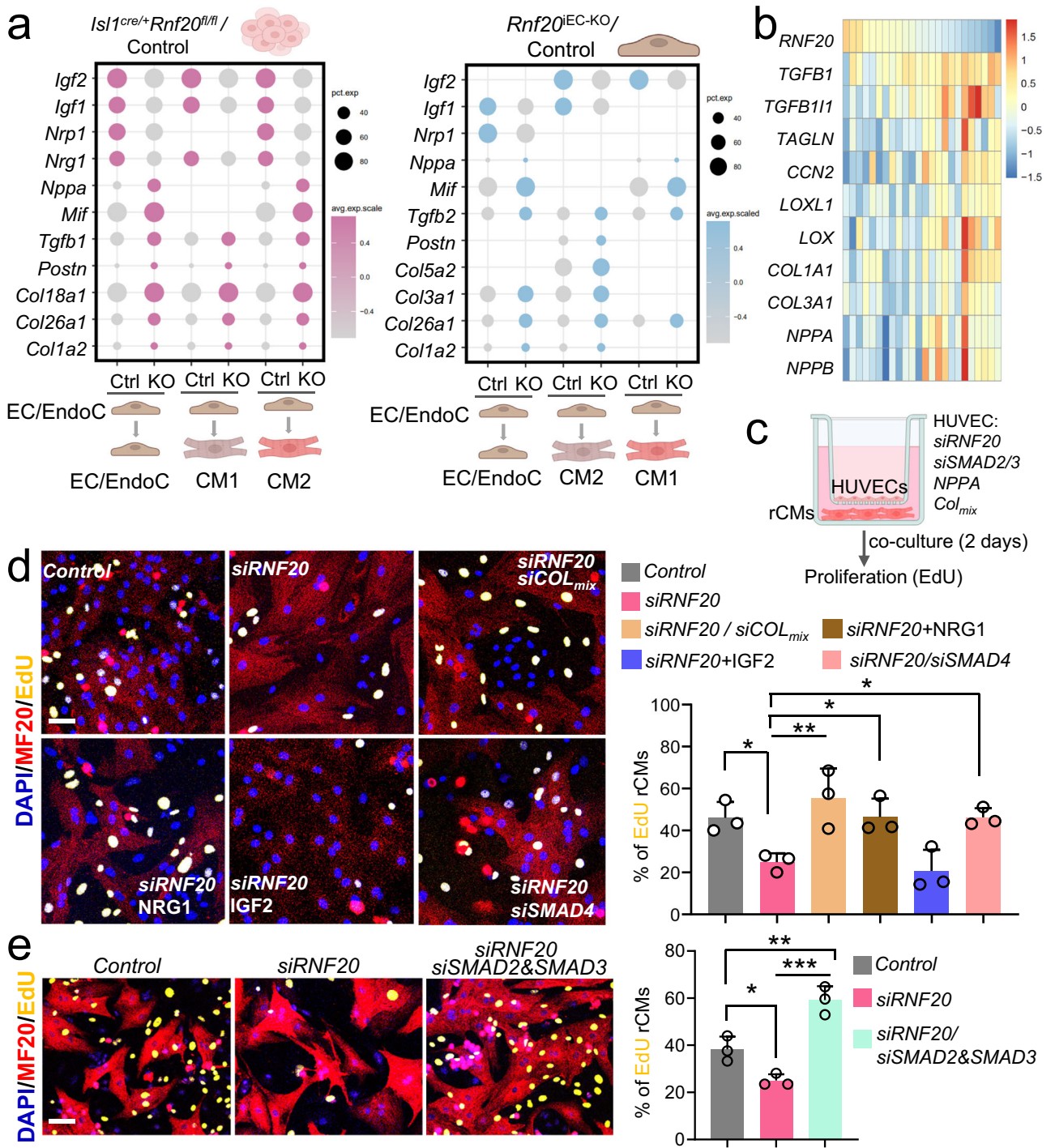

**Fig. 8 | Rnf20-dependent EC-derived factors regulate CM proliferation.**
**a** Schematic representation of the NicheNet analysis of significant ligand–receptor pairs considering differentially expressed ligands in ECs and receptors on CMs in *Isl1*[cre/+]*Rnf20*[fl/fl] and *Rnf20*[iEC-KO] versus control embryos. The dot plots depict ligands contributing to the CM response in *Rnf20*[iEC-KO] vs control embryos. Cardiac cell type schematics created in BioRender.com. Dobreva, G. (2025) https://BioRender.com/ce62pb8. **b** Heatmap representation of the expression of *Rnf20* and TGF-β members/ downstream targets in CHD patients. **c** Schematic representation of the co-culture experiment design involving rCMs and transfected HUVECs. Transwell co-culture system schematics created in BioRender.com. Dobreva, G. (2025) https://BioRender.com/yr6530z. **d** Immunostaining for EdU and MF20 (left panels) and percentage of proliferating EdU-positive CMs in co-cultures with HUVECs

transfected with control siRNA (*n* = 3), siRNA against *RNF20* (*siRNF20*, *n* = 3), *siRNF20* supplemented with recombinant NRG1 (*n* = 3) or IGF2 (*n* = 3), or *siRNF20* together with siRNAs against various collagens (*n* = 3) or *SMAD4* (*n* = 3), determined by EdU-incorporation for 24 h (right panel). Scale bars, 50 μm. **e** Immunostaining for EdU and MF20 (left panels) and percentage of proliferating EdU-positive CMs in co-cultures with HUVECs transfected with control siRNA (*n* = 3), siRNA against *RNF20* (*siRNF20*, *n* = 3), or *siRNF20* together with siRNAs against *SMAD2* and *SMAD3* (*n* = 3), determined by EdU-incorporation for 24 h (right panel) (*n* = 3 per group). Scale bars, 50 μm. Statistical analysis between multiple groups in (**d, e**), was performed using one-way-ANOVA with Šídák's correction. Data are shown as mean ± SEM. **p* < 0.05, ***p* < 0.01, ****p* < 0.001.

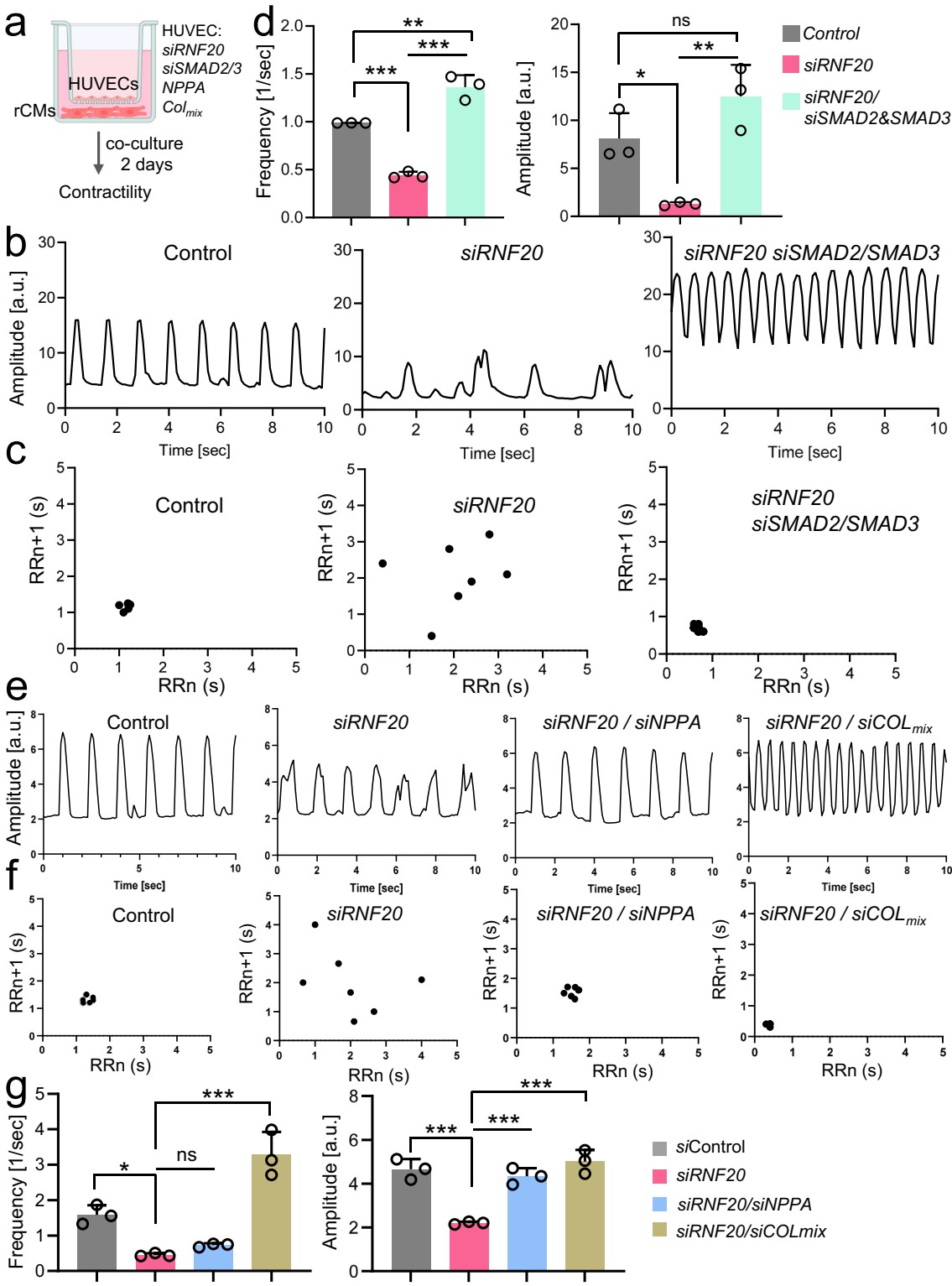

(ThermoFischer, A1048301) overnight. 10 nM 4-Hydroxytamoxifen was added for 3 days to induce *Rnf20* ablation. Afterwards, cells were washed in PBS and co-cultured with rat CMs as described above.

### Contractility analysis

Movies of beating embryonic hearts and rat CMs were acquired using Leica DMI8 microscope and a 5× or 10× objective. Amplitude and contractility profiles were obtained by analysis with the ImageJ macro MYOCYTER[55].

Poincaré plot analysis was performed to quantify beat-to-beat variability in cardiac rhythm. RR intervals were calculated from time-lapse recordings by measuring the time between successive cardiac contraction peaks ($RR_n$ and $RR_{n+1}$) by using MYOCYTER. Each $RR_n$ was plotted against the subsequent $RR_{n+1}$ to generate Poincaré plots.

### EndMT collagen gel assay

EndMT assay was performed according to ref. 73. Proximal outflow tract (pOFT) was microdissected and cut longitudinally to expose the

**Fig. 9 | Rnf20-dependent EC-derived factors regulate CM contractility.**
**a** Schematic representation of the co-culture experiment design involving rCMs and transfected HUVECs. Transwell co-culture system schematics created in BioRender.com. Dobreva, G. (2025) https://BioRender.com/yr6530z. **b** Plot of contraction amplitude and beating speed of rat CMs co-cultured with HUVECs transected with control siRNA, *siRNF20*, or *siRNF20* together with siRNAs against *SMAD2* and *SMAD3* extracted from video sequences using MYOCYTER. **c** Poincaré plot derived from image sequences of beating rat CMs co-cultured with HUVECs transfected with control siRNA, *siRNF20*, or *siRNF20* together with siRNAs against *SMAD2* and *SMAD3*. **d** Frequency and amplitude of image sequences recorded on rCMs co-cultured with HUVECs transfected with control siRNA ($n = 3$), *siRNF20*

($n = 3$), or *siRNF20* together with siRNAs against *SMAD2* and *SMAD3* ($n = 3$). **e** Plot of contraction amplitude and beating speed of rat CMs co-cultured with HUVECs transected with control siRNA or siRNA against the indicated genes extracted from video sequences using MYOCYTER. **f** Poincaré plot derived from image sequences of beating rat CMs co-cultured with HUVECs transfected with control siRNA or siRNA against the indicated genes. **g** Frequency and amplitude of image sequences recorded on rCMs co-cultured with HUVECs transfected with control siRNA ($n = 3$), *siRNF20* ($n = 3$), or *siRNF20* together with siRNA against *NPPA* ($n = 3$), or siRNAs against various collagens ($n = 3$). Statistical analysis between multiple groups in **d, g**, was performed using one-way-ANOVA with Šídák's correction. Data are shown as mean ± SEM. *$p < 0.05$, **$p < 0.01$, ***$p < 0.001$.

endocardial surface and placed on collagen gel (1 mg/ml collagen I from rat tail (Ibidi, 50201) in 10x M199 (ThermoFischer, 11825015)). The explants were allowed to attach to the gel for 6 h and then cultured in M199 medium with 1% FBS for 72 h. Images were taken by Leica DMI8 microscope and a 5× or 10× objective.

### Migration assays

For the scratch assay, HUVECs were cultured in full medium until 100% confluency was achieved. A scratch was introduced to the monolayer utilizing a 200 μl pipette tip, followed by a washing step to eliminate any detached cells and debris. Images were captured immediately after the scratch to document the initial state of the cell monolayer (time-point 0). The cells were then incubated at 37 °C with 5% $CO_2$ for a duration of 7 h, after which additional images were captured to assess cell migration (timepoint 7 h). All images were acquired utilizing a Leica DMI8 microscope, and the analysis of cell migration was conducted using ImageJ/Fiji software.

For the Boyden chamber migration assay, HUVECs (100,000 cells in 24 well plates inserts) were seeded into the upper chamber of transwell inserts (8 μm pore size; Corning) in serum-free Endothelial Cell Growth Medium MV 2 (PromoCell, C-22022). The lower chamber was filled with Endothelial Cell Growth Medium MV 2 (PromoCell, C-22022) with supplements. After incubation at 37 °C for 24 h, non-migrated cells remaining on the upper surface of the membrane were removed. Migrated cells on the lower surface were fixed in 4% paraformaldehyde and stained with crystal violet.

### Tube formation assay

Tube formation assays were performed in μ-Slide Angiogenesis slides according to the manufacturer's instruction (Ibidi, 81506).

### Immunostaining

For immunostainings, cells were fixed on cover slips with 4% PFA for 10 min. After washing with PBS, cells were permeabilized for 30 min with 0.3% Triton X-100 in PBS and staining was performed. Phospho-SMAD2 (Ser465/467)/SMAD3 (Ser423/425) primary antibody (1:100, Cell Signaling, D27F4) was used in blocking buffer. Incubation with primary antibody was performed overnight at 4 °C in a humidified chamber. After three washing steps with PBS, combinations of the corresponding secondary antibodies, conjugated to Alexa Fluor 555 or Alexa Fluor 488 (1:500, Thermo Fisher Scientific) with and DAPI (1:10,000) were added in blocking buffer and incubated for 2 h at room temperature. The slides were then washed three times with PBS. Sections were mounted with Mowiol 4-88 (Millipore, 475904) mounting medium. Image acquisition was done using a Zeiss LSM 710 confocal microscope.

For EdU staining and analysis, 0.02 mmol/L EdU was added to primary rat CMs, for 24 h. After three washes with PBS, cells were fixed with 4% PFA for 10 min. After washes with PBS, cells were permeabilized for 30 min with 0.3% Triton X-100 in PBS and staining was performed using the Click-iT™ EDU Alexa Fluor™ 488 Kit (ThermoFischer, C10337) following the manufacturers' instructions.

### Histology

Embryos were fixed in 4% paraformaldehyde and dehydrated by increasing percentage of ethanol, followed by paraffin embedding. Paraffin-embedded samples were cut into 7 μm sections. Hematoxylin and Eosin Y staining (Sigma-Aldrich, GHS116, HT110216) was performed according to the manufacturer's instructions.

Detection of apoptotic cells was performed using FragEL DNA Fragmentation Detection Kit according to the manufacturer's instructions (Sigma-Aldrich, QIA33).

Detection of proliferating cells was performed using the Click-iT™ EDU kit, followed by staining with HRP-conjugated secondary antibody as described in the section below.

Stained sections were mounted with ROTI®Histokitt (Carl Roth, 6638.1).

Postnatal hearts were rapidly excised and transferred into PBS to flush residual blood while the heart was still contracting. Afterwards the hearts were immersed in 0.3 M KCl solution to arrest contraction and ensure relaxation of cardiac tissue. Immediately after, hearts were embedded in OCT within a cryomold placed in a petri dish over chilled pentane cooled in liquid nitrogen for rapid freezing. Embedded tissue blocks were stored at −80 °C until sectioning.

Cryosections of 7 μm thickness were prepared using a Cryostat at −30 °C and mounted onto Superfrost Plus slides. Sections were air-dried at RT, then fixed for 20 mins in 4% PFA. After fixation, slides were washed twice with PBS before proceeding to Immunofluorescence staining.

### Immunofluorescence staining of tissue sections

For immunofluorescence staining on paraffin sections, slides were first dewaxed. Heat-induced epitope retrieval was performed by boiling for 10 minutes at 400 W in a pressure cooker filled with freshly prepared 10 mM sodium citrate solution with a pH of 6.0. After cooling, the slides were blocked in a humidified chamber for 1 h in blocking buffer consisting of 0.3% Triton X-100 and 3% BSA in PBS. The following primary antibodies were used in blocking buffer: anti-myosin heavy chain (MF20) (1:10, DSHB), anti-phospho-Histone H3 (Ser10) (1:200, Cell Signalling #9701S), anti-Ubiquityl-Histone H2B (Lys120) (1:200, Cell Signaling, 5546). Incubation with primary antibody was performed overnight at 4 °C in a humidified chamber. After three washing steps with PBS, combinations of the corresponding secondary antibodies, conjugated to Alexa Fluor 555 or Alexa Fluor 488 (1:500, Thermo Fisher Scientific) with the conjugated primary antibody Isolectin B4 (IB4) Alexa Fluor 647 (1:200, Thermo Fisher Scientific, I32450) and DAPI (1:10,000) were added in blocking buffer and incubated for 2 h at room temperature. The slides were then washed three times with PBS. Sections were mounted with Mowiol 4-88 (Millipore, 475904) mounting medium. Image acquisition was done using a Zeiss LSM 700 confocal microscope.

For immunofluorescence staining on cryosections, permeabilization and blocking was performed in PBS containing 10 % FCS and 0.5% Triton X-100 for 1 h.

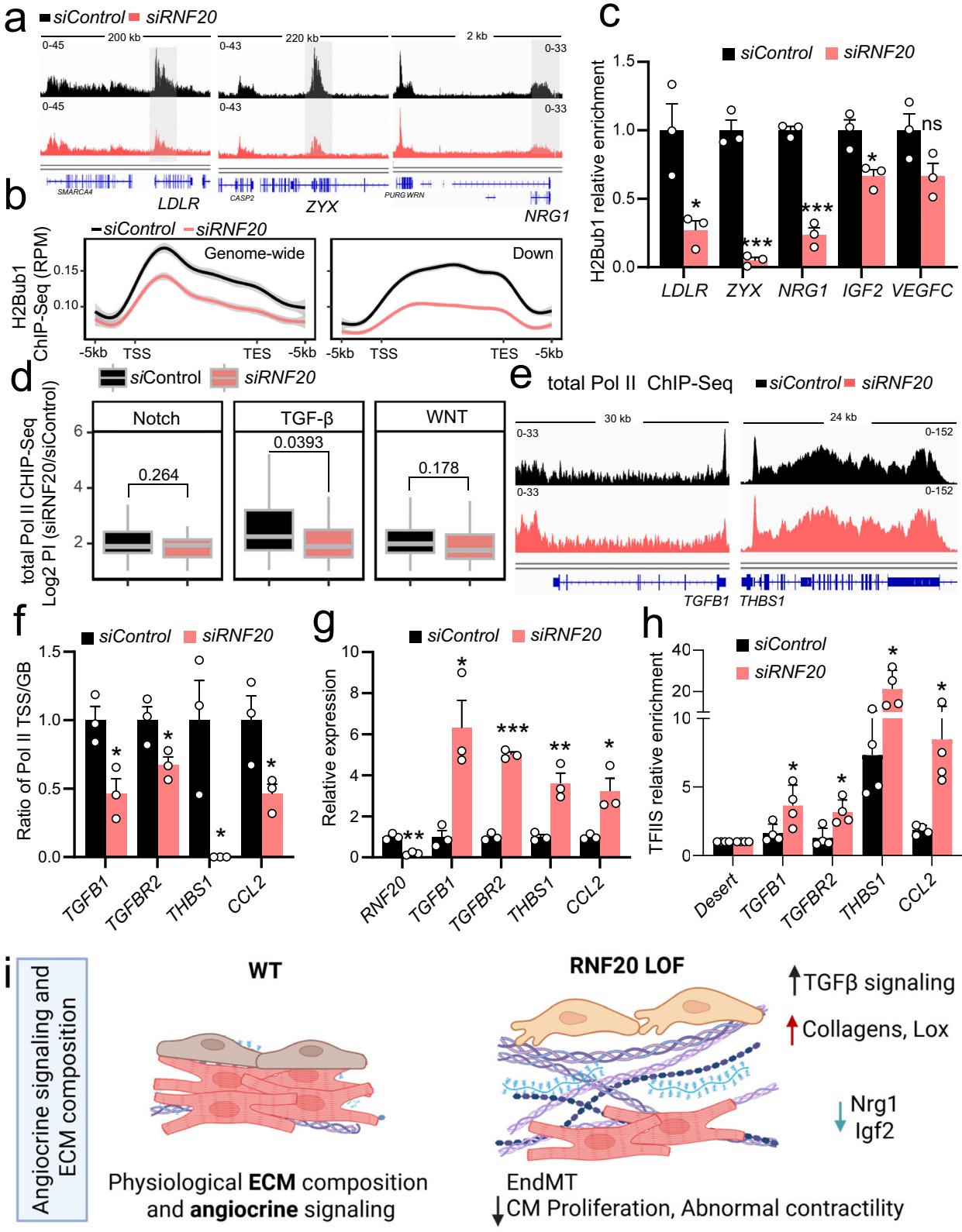

For direct staining with conjugated Antibodies, sections were incubated for 2 h at RT with WGA-alexa-fluor-633, (1:100, Invitrogen #W21404) Isolectin B4-Alexa-fluor-488, (1:100 Invitrogen #I21411) and DAPI (1:10.000), all diluted in blocking buffer.

For staining with primary and secondary antibodies, sections were stained overnight at 4 °C with primary antibodies against CD31 (1:100, BD #553370), pH3S10 (1:200, Cell Signalling #9701S) or Troponin I

(1:100, abcam, ab56357) diluted in blocking buffer. The next day, slides were washed three times and incubated for 1 h at RT in the dark with Donkey anti-rabbit Alexa Fluor™ 488 and donkey anti-rat Alexa Fluor™ 633-conjugated (1:400, Invitrogen) secondary antibodies along with DAPI in blocking buffer.

Finally, sections were washed in PBS containing 0.5% Triton X-100 three times and once in PBS alone, followed by mounting with an

**Fig. 10 | RNF20 restrains Pol II pause release at TGF-β targets, while promoting growth factor gene expression via H2Bub1.** a Genome tracks of H2Bub1 ChIP-Seq reads in HUVECs transfected with control siRNA (*siControl*) of siRNA against *RNF20* (*siRNF20*). The gray boxes highlight gene loci with lower H2Bub1 levels compared to neighboring genes. **b** Average genome-wide H2Bub1 ChIP-Seq signal as well as H2Bub1 ChIP-Seq signal at genes downregulated upon RNF20 LOF in both HUVECs and mESC-derived ECs. **c** H2Bub1 enrichment at *LDLR, ZYX, NRG1, IGF2* and *VEGFC* and *Pdk1* in HUVECs transfected with control siRNA ($n = 3$) or siRNA against *RNF20* ($n = 3$), determined by H2Bub1 Chip-qPCR. **d** Log2 of the Pol II pausing index (PI) at genes involved in Notch ($n = 63$), TGF-β ($n = 83$) and WNT ($n = 113$) signaling. Pausing index was calculated from Pol II ChIP-Seq datasets of HUVECs transfected with control siRNA ($n = 3$) or siRNA against RNF20 ($n = 3$). Boxplots show the median (central line), the 25th and 75th percentiles (box), and the 5th and 95th percentiles (whiskers). *P* values were calculated using a two-sided Wilcoxon test. **e** Genome tracks of total Pol II ChIP-Seq reads in HUVECs transfected with control siRNA (*siControl*) of siRNA against *RNF20* (*siRNF20*). **f** Ratio of the Pol II enrichment at the TSS and GB of *TGFB1, TGFBR2, THBS1* and *CCL2* in control ($n = 3$) and *siRNF20* ($n = 3$) HUVECs, determined by Pol II Chip-qPCR. TSS, Transcription Start Site; GB, Gene Body ($n = 3$). **g** Relative mRNA expression of *TGFB1, TGFBR2, THBS1* and *CCL2* in control ($n = 3$) and *siRNF20* ($n = 3$) HUVECs. **h** TFIIS enrichment at *TGFB1, TGFBR2, THBS1* and *CCL2* in control and *siRNF20* HUVECs, determined by TFIIS Chip-qPCR ($n = 4$ independent ChIP experiments). **i** Model of the function of RNF20 in ECs for proper cardiac morphogenesis and function. Created in BioRender. Dobreva, G. (2025) https://BioRender.com/xfgieqh. Statistical analysis between two groups in (**c, f, g, h**) was performed using an unpaired two-tailed Student's t-test. Data are shown as mean ± SEM. *$p < 0.05$, **$p < 0.01$, ***$p < 0.001$.

antifade mounting medium. Sections were stored at 4 °C until imaging. Imaging was performed with a Zeiss Axio Scan Z.1 and a Zeiss LSM700 confocal microscope.

## RNA isolation, RT-PCR, and real-time PCR
RNA was isolated using the TRIzol RNA Isolation Reagent (Invitrogen, 15596018), according to manufacturer's instructions. For real-time PCR analysis cDNA was synthesized with the High Capacity cDNA Reverse Transcription Kit (#4368813, Applied Biosystems). Real-time PCR was performed using the SYBR GREEN PCR master mix (#A25742, Applied Biosystems,) or qPCR BIO SyGreen Blue Hi-ROX (Nippon Genetics). Cycle numbers were normalized to validated housekeeping genes and, for samples with low RNA input, also to 18S rRNA to ensure reliable quantification. Details are provided in the Source Data. Primers used for RT-qPCR analysis are listed in Supplementary Data 8.

## RNA-Sequencing and analysis
RNA for RNA-Seq analysis was isolated using the RNeasy Plus Universal Mini kit (Qiagen, 73404). NGS were performed either on NextSeq 550 Illumina System at the NGS Core Facility or by BGI on a BGISEQ-500 platform.

High-quality trimmed reads were aligned to the mm10 reference genome using STAR (v. 2.7.10a) with the parameters '--alignIntronMin 20 --alignIntronMax 500000'[74]. The resulting SAM files were utilized to create tag libraries with default settings using 'makeTagDirectory.' Quantification of mapped reads was performed with the 'analyzeRepeats.pl' function (using 'rna mm10 -strand both -count genes –noadj') from HOMER (v. 5.1)[75].

Differential expression analysis was carried out and normalized using DESeq2 (v. 3.4.2). The 'rpkm.default' function from EdgeR (v. 3.42.4) was employed to calculate Reads Per Kilobase of Million Reads (RPKM). Differentially regulated genes were identified based on a *p*-value of 0.05.

BAM files were merged using bamtools merge (v. 2.5.1) with default settings, and then converted to BigWig files using 'bamCoverage' (v. 3.5.1) with the following parameters: '-bs 20 --smoothLength 40 -p max --normalizeUsing RPKM -e 150.'

For gene ontology pathway analysis, the 'compareCluster' function from clusterprofiler (v. 4.8.1) and the web-based tool DAVID were employed.

## 10x Genomics scRNA-Sequencing and analysis
Hearts from E10.5 Isl1-Rnf20 were dissected and after swift genotyping five control (Isl1+/-Rnf20+/+ or Isl1+/+Rnf20+/fl) and five Isl1+/-Rnf20fl/fl (*Rnf20* KO) hearts were pooled. Similarly, hearts from five E11.5 Cdh5CreERT2-positive Rnf20fl/fl embryos and five E11.5 control Cdh5CreERT2-negative Rnf20fl/fl embryos were pooled. Pooled hearts were further digested with 0.05 mg/ml DNase I (Worthington, 2139) and 1 mg/ml Collagenase I (Worthington, 4197) diluted in HBSS (Thermo Fisher Scientific, 14175053) until complete digestion.

Reaction was stopped by adding DMEM medium containing 10% FBS and after centrifugation resuspended in FACS buffer (10% FBS in PBS). Living cells were FACS sorted by staining with SYTOX™ Blue Dead Cell Stain (1:2000, Thermo Fisher Scientific, S34857) and then diluted at a density of 1000 cells/μl in 0.4% BSA. 8000 cells were used for library preparation according to the Chromium Next GEM Single Cell 3' Kit v3.1 (10xGenomics, 1000269) with the Chromium Next GEM Chip G Single Cell Kit (10xGenomics, 1000127) and sequenced on a BGISEQ-500 platform.

Raw files were analyzed using the CellRanger (v.7.1.0) count function with specific settings, such as '--expected-cells 3000,' and the 'refdata-gex-mm10-2020-A' genome[76]. The filtered feature matrices served as input files for data integration using Seurat (v. 4.3.0). Low-quality cells were filtered out by selecting only those cells expressing more than 900 genes per cell. Clusters were identified using the 'FindClusters' function with specific parameters (resolution=0.5, random.seed=1, and algorithm=1). The UMAP or TSNE plots were generated using RunUMAP and RunTSNE (dims=1.30, n.neighbors= 30).

In all single-cell datasets, marker genes were identified using the 'FindAllMarkers' function with specific parameters (min.pct=0.25, logfc.threshold=0.25, test.use=Wilcox). Differentially regulated genes between control (Co) and *Rnf20* knockout (Ko) cells in each cluster were identified with 'FindMarkers' using ident.1=Ko_cells, ident.2=Co_cells, min.pct=0.05, logfc.threshold=0.25, and test.use=Wilcox.

To isolate cardiomyocytes (CMs) for the scRNA-Seq analysis, CM clusters (CM1 and CM2) were selected and reclustered at a resolution of 0.8. To identify differences in endothelial cells (ECs) between Isl1[cre/+]-Rnf20[fl/fl] KO and control embryos, EndoV, EC/EndoC and OFT-MES were selected and reclustered at a resolution of 0.2. For the E11.5 datasets of control and *Cdh5*[CreERT2]-Rnf20[fl/fl] KO, similar clustering settings as those used for the Isl1 scRNA-Seq data at E10.5 were applied to select CM populations.

Publicly available scRNA-Seq gene expression lists from E7.75 and E8.25 embryos[50] were downloaded from NCBI (GSE108963), and first heart field (FHF) and Second heart field (SHF) were defined similar to (44). FHF cells were sorted by Nkx2-5 with low expression of Isl1. SHF were cells sorted by Isl1 with low expression of Nkx2-5. Differential expression analysis between SFH and FHF cells was performed using 'FindMarkers' with ident.1 = E7.75_Isl1_positive_cells, ident.2 = E7.75_Nkx2_5_positive_cells, min.pct = 0.05, logfc.threshold=0.25, and test.use=Wilcox.

Single-cell RNA-seq data from TOF patients and healthy donors were obtained from GEO accession GSE203274. Donor and TOF samples were merged and normalized following the approach described in [https://doi.org/10.32604/chd.2023.047689], with the modification that only cells expressing between 200 and 9000 genes and showing a mitochondrial gene expression ratio below 0.2% were retained. Dimensionality reduction and visualization were performed using the RunUMAP function with the parameters: reduction = "harmony", dims = 1:10, and n.neighbors = 25. Differential gene expression between ToF

and donor samples for each cell type was calculated using the Find-Markers function. Differences in cell type proportions per sample were assessed using a two-sided binomial exact test implemented in R.

## Ligand–receptor analysis from scRNA- sequencing data

Ligand–receptor interaction analysis was performed using NicheNet (v2.0.0)[61], a computational framework that predicts intercellular communication by integrating ligand–receptor interactions with downstream signaling and transcriptional responses. In our study, a ligand was considered relevant if the corresponding angiocrine gene was highly expressed in endothelial cells (EC/EndoC), its receptor was expressed in cardiomyocytes (CMs), and the known downstream target genes of this signaling axis were differentially expressed in CMs.

We applied the nichenet_seuratobj_cluster_de function using the following parameters: expression_pct = 0.10, lfc_cutoff = 0.25, filter_top_ligands = FALSE, top_n_ligands = 100, and top_n_targets = 200. This function operates on a Seurat-normalized expression matrix, identifies differentially expressed genes (DEGs) per cluster, and maps them onto the NicheNet reference database (version curated as of July 2023).

## Chromatin immunoprecipitation (ChIP) and ChIP-sequencing data analysis

Pol II (Rpb1 NTD, 14958S, Cell Signalling; 1 μg/IP) and TFIIS ChIP (TCEA1, Abcam, ab185947; 1 μg/IP) were done as described before[41]. For ChIP-qPCR, 0.1 ng of purified DNA was used per reaction. Primers used for ChIP-qPCR analysis are listed in Supplementary Data 8.

RNA Polymerase II and H2Bub ChIP-seq data from GEO under accession number [GSE212524] were integrated with our analysis following standard processing pipelines. Trimmed reads were aligned to the mouse hg38 genome using Bowtie2 (v. 2.4.4) with default settings.

Bam files were visualized on the gene body of the RefSeq transcriptome using the 'ngs.plot.r' function from ngsplot (v. 2.47.1) with the following settings: '-G hg38 -R genebody -c contig.txt -L 5000 -FL 200 -D refseq.' The output text file from ngsplot was processed in an R script to create smoothed line plots for H2Bub-ChIP-seq enrichment genome-wide and Rnf20-targets in both HUVECs and mESC-derived ECs with the 'geom_smooth' function (method = 'gam', size=1) from ggplot2 (v. 3.4.2).

The Pol II pausing index was calculated using the PIC repository[77], where the Pol II enrichment at the transcription start site (TSS) (−50 to 300 bp) was divided by the enrichment of Pol II from 300 bp to 3 kb after the termination site. The Pol II data were then plotted on pathway gene lists. Gene lists related to the TGFB (GO:0141091), Notch (GO:0007219), and WNT signaling (GO:0016055) were obtained from (https://www.informatics.jax.org). Boxplots were generated in R using 'geom_boxplot' (outlier.shape=NA, width=0.8) from ggplot2 (v. 3.4.2). P-values within each group were calculated using the 'wilcox_test' function from the rstatix (v. 0.7.2) library.

## Statistics and reproducibility

The number of animals used in each study is reported in the figures and the figure legends. All experiments were performed at least three independent times in biological replicates and the respective data were used for statistical analyses. The 'n' values indicated in the figure legends always refer to biological replicates, such as mice, embryos, or cells. For the co-culture analysis presented in Figs. 7d–g, 8d, e, 9d, g the 'n' values indicated in the figure legends correspond to endothelial cells isolated from individual mice or to HUVECs transfected with different siRNAs or overexpressing distinct constructs, which were subsequently co-cultured with neonatal rat cardiomyocytes. Differences between groups were assessed using an unpaired two-tailed Student's $t$ test or ANOVA multiple comparisons test. P-values represent significance $*P < 0.05$, $**P < 0.01$, $***P < 0.001$. Barplots and Boxplots were produced with GraphPad Prism 8 software.

## Reporting summary

Further information on research design is available in the Nature Portfolio Reporting Summary linked to this article.

## Data availability

Chromatin immunoprecipitation (ChIP)-sequencing, bulk and single-cell RNA sequencing data generated in this study have been deposited in GEO, accession number GSE246928. Bulk RNA-seq of Tetralogy of Fallot (TOF) patients has been deposited in GEO, accession number GSE305562. Processed data are included in the Supplementary Data 1-8. RNA Polymerase II, H2Bub1 ChIP-seq and RNA-seq data from HUVECs have been previously deposited in GEO, accession number GSE212524[41] [https://www.ncbi.nlm.nih.gov/geo/query/acc.cgi?acc=GSE212524]. Single-cell RNA-seq data from Tetralogy of Fallot (TOF) patients and healthy donors were retrieved from GSE203274[48] [https://www.ncbi.nlm.nih.gov/geo/query/acc.cgi?acc=GSE203274]. Single-cell RNA-seq from E7.75 and E8.25 mouse embryos were retrieved from GSE108963[50] [https://www.ncbi.nlm.nih.gov/geo/query/acc.cgi?acc=GSE108963]. Source data are provided with this paper.

## Code availability

The full code is available at the indicated GitHub repository (https://github.com/jcorderJC12/03Rnf20_CHD_NatC) and additional information is available upon request to the corresponding authors.

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

## Acknowledgements

We would like to thank the FlowCore Mannheim, the Preclinical Models and NGS Core Facilities of the Medical Faculty Mannheim at Heidelberg University for excellent support. We are grateful to Boyan Garvalov for careful reading of the manuscript and the constructive comments. This work was supported by the CRC 1366 (Projects A03, A06), the CRC1550 (Project A03) funded by the DFG, the DZHK (81Z0500202), funded by BMBF and the Baden-Württemberg foundation special program 'Angioformatics single cell platform'.

## Author contributions

Y.D., N.T.-E., R.G., Y.R., Y.C., A.S., O.L., Y.W., E.C. and A.A. performed the experiments. Y.C., M.M., P.G., T.L., M.G. collected clinical data and CHD specimens. J.C. performed the bioinformatic analysis. C.C.W., G.B., M.P., T.W., R.O. and J.H. provided reagents and valuable intellectual input. Y.D., N.T.-E., R.G., J.C. and G.D. designed the experiments, analyzed the data and wrote the manuscript. All authors discussed the results and commented on the manuscript.

## Funding

## Competing interests

The authors declare no competing interests.

## Additional information

¹Department of Cardiovascular Genomics and Epigenomics, European Center for Angioscience (ECAS), Medical Faculty Mannheim, Heidelberg University, Mannheim, Germany. ²Max Planck Institute for Heart and Lung Research, Bad Nauheim, Germany. ³Division of Pediatric Cardiac Surgery, Department of Cardiac Surgery, University Hospital Heidelberg, Heidelberg, Germany. ⁴Department of Pediatric and Congenital Cardiology, University Hospital Heidelberg, Heidelberg, Germany. ⁵Department of Cardiovascular Physiology, ECAS, Medical Faculty Mannheim, Heidelberg University, Mannheim, Germany. ⁶Ploidy and Organ Physiology Laboratory, ECAS, Heidelberg University, Mannheim, Germany. ⁷Université de Lorraine, CNRS, Laboratoire IMoPA, UMR 7365, Nancy, France. ⁸Berlin Institute of Health at Charité—Universitätsmedizin Berlin, Berlin, Germany. ⁹Max Delbrück Center for Molecular Medicine in the Helmholtz Association, Berlin, Germany. ¹⁰Helmholtz-Institute for Translational AngioCardioScience (HI-TAC) of the Max Delbrück Center for Molecular Medicine in the Helmholtz Association (MDC) at Heidelberg University, 69117 Heidelberg, Germany. ¹¹German Centre for Cardiovascular Research (DZHK), Mannheim, Germany. ¹²Experimental Pharmacology, ECAS, Medical Faculty Mannheim, Heidelberg University, Mannheim, Germany. ¹³Cardiovascular Pharmacology, ECAS, Medical Faculty Mannheim, Heidelberg University, Mannheim, Germany. ¹⁴These authors contributed equally: Yanliang Dou, Nalan Tetik-Elsherbiny, Rui Gao. ✉e-mail: Julio.Cordero@medma.uni-heidelberg.de; Gergana.Dobreva@medma.uni-heidelberg.de

