## [Peer Review file · Nature Communications]

Endothelial RNF20 Suppresses Endothelial-to-Mesenchymal Transition and Safeguards Physiological Angiocrine Signaling to Prevent Congenital Heart Disease

Corresponding Author: Professor Gergana Dobрева

Version 0:

Reviewer comments:

Reviewer #1

(Remarks to the Author)

In this manuscript, Drs. Cordero/Dobрева and colleagues examine the role of the histone H2B ubiquitin (H2Bub1) ligase RNF20 in heart development, and its possible association with congenital heart disease (CHD). An RNF20 mutation has been previously identified in CHD patients, with mouse and in vitro models demonstrating an important role for RNF20 in heart development and cardiomyocyte maturation, respectively.

Gene expression (RNAseq) is examined from surgically removed right ventricular outflow tract tissue in cyanotic Tetralogy of Fallot (ToF) patients. Of a large number of dysregulated genes, expression of RNF20 is significantly reduced. Analysis of heterozygous Rnf20 murine ESCs or Rnf20 morpholino targeted zebrafish embryos suggests an accelerated differentiation of cardiac progenitors to a cardiomyocyte fate, with potentially impaired endothelial differentiation. This is supported by RNAseq analysis of differentiated mESCs, which suggests expanded “first heart field” and decreased “second heart field” associated gene expression with decreased Rnf20. In mice, Isl1:Cre-based SHF deletion of Rnf20 results in a spectrum of cardiac defects including thinned ventricular walls and poor cardiac contractility, with embryonic death by E14.5. Similar phenotypes are noted with endothelial-specific deletion of Rnf20, suggesting a later function for Rnf20 (subsequent to function in the SHF progenitors) to modulate EC signaling to CMs for proper ECM formation and CM proliferation/maturation. This is supported in elegant EC/CM co-culture experiments where Rnf20 function is altered in ECs.

This work represents a broad examination of Rnf20 function in heart development, with extensive in vitro, in vivo and genomics analyses. Overall, the manuscript is clearly written, with well-presented data.

Minor Comments:

1. A very minor point, but on page 4 Isl1 is discussed as a SHF marker, while it may be more specific to the aSHF? This is highly stage-specific.
2. Top of page 8: “haploinsufficient cells” should be clarified to the type of cell (mESCs in this case).
3. The acronym ECs is used for both endocardial cells (Introduction and some text) and endothelial cells (Methods page 24 and other sections). The mESC differentiation protocols used are making endothelial cells, as opposed to endocardial cells. This should be clarified.
4. Figure 3: PTA and other histological findings should be more clearly demonstrated for non-experts, as should measures of RV size.

Major Comments:

1. For the initial patient sample RNAseq analysis, it is not clear how many samples were used, what “control” healthy tissue was used, etc.

2. The zebrafish data as presented is difficult to interpret. What controls (RNA rescue, etc.) have been attempted in the MO model? There is a clear effect on overall development (Figure S1F). If premature differentiation of SHF cells is being claimed in this model, *Isl1* expression and other analysis of SHF markers should be included.

3. Figures 3 and S2: it is confusing why effects on CM death (at e11.5) and proliferation are seen in both the LV and RV, as well as H2Bub1 immunostaining, given that a SHF-specific deletion of *Rnf20* was carried out. Presumably, LV CMs and ECs should maintain functional *Rnf20*.

4. The interpretation of scRNAseq data in the first paragraph of page 11 is puzzling: greater “aSHF” and “pSHF” cell proportions are seen in the *Rnf20* lof model, which is attributed to potential “defects in aSHF progenitor differentiation”. This seems to be directly opposed to conclusions for mESC and zebrafish analyses. The cell types as presented are also poorly defined.

5. Overall, the statement that “RNF20 governs the diversification of cardiac progenitors in the first and second heart fields and is crucial for the development of the SHF” (Discussion, page 18) requires a more detailed analysis of FHF and SHF markers at appropriate stages prior to heart assembly. Many of the gene expression changes noted in e10.5-14.5 hearts may be secondary to effects on heart function and development, as also seen in EC-specific *Rnf20* lof models used.

6. The much greater severity of the EC-specific deletion of *Rnf20* vs that seen with SHF/*Isl1*-based deletion should be discussed in greater detail. Further, data should be presented on the extent of Cre-mediated removal of both *Rnf20* floxed alleles in these models (is full deletion being reached)?

Reviewer #2

(Remarks to the Author)

Y. Dou, N. Tetik-Elsherbiny, R. Gao and collaborators proposed a manuscript entitled “RNF20 at the Nexus of Cardiac Progenitor Diversification and Angiocrine Signaling in Congenital Heart Disease” for publication in Nature Communications. In this study the authors highlighted the important role of *Rnf20*, an ubiquitin ligase responsible for the monoubiquitination H2Bub1, in cardiac development.

The authors state the following: (1) downregulation of RNF20 in patients presenting a Tetralogy of Fallot with oxygen saturation below 88%; (2) precocious differentiation of cardiac progenitors due to *Rnf20* downregulation in mESC in vitro and in zebrafish in vivo; (3) impaired endothelial differentiation in vitro after *Rnf20* downregulation; (4) decreased cardiomyocyte proliferation together with a thinner compact and trabecular myocardium in *Isl1-cre;Rnf20^{fl/fl}* mouse embryos at E10.5; (5) shortened outflow tract, a small right ventricle and a decrease ventricular cardiomyocyte proliferation in *Cdh5-cre;Rnf20^{fl/fl}* mouse embryos at E11.5; (6) lethality in *Cdh5-cre;Rnf20^{fl/fl}* mouse embryos at P1, accompanied by a poor cardiomyocyte organization; (7) role for *Rnf20* in the inhibition of the endothelial-to-mesenchymal transition in the cardiac outflow tract; (8) decreased PolII pausing index at upregulated genes and TGF- β effectors after *Rnf20* knockdown in endothelial cells in vitro; (9) increased TGF- β signaling in the context of *Rnf20* loss, which is associated with the decreased cardiomyocyte proliferation.

The authors showed that *Rnf20* acts both in cardiac progenitors of the second heart field and in endocardial cells, supporting it as a candidate in the development of congenital heart defects. The authors perform an extensive investigation of the role of *Rnf20* in both cardiac progenitors and endocardial cells, using a significant combination of approaches and models. The proposed study will be of interest for investigators in the field of cardiac development and congenital heart disease.

Unfortunately, a rigorously demonstrated mechanistic understanding of why diminished RNF20 dose causes the observed effects or observed changes in gene expression is not offered. Instead, a very broad range of models and assays is brought to bear, confirming the importance of RNF20 without providing insight into its specific role. Some experiments are over interpreted, and many claims are made that are not substantiated by the data included in the manuscript, as detailed below. A good example is the aggressive claim that RNF20 governs the diversification of cardiac progenitors in the first and second heart fields. Most certainly the authors demonstrate that the gene is required in the SHF. However, the authors fail to rigorously demonstrate a requirement for RNF20 in the diversification of cardiac progenitors.

A recently published paper on the role of RNF20 in heart development and maturation (PMID: 37967007) indicated a time-specific role for RNF20 across cardiac maturation— basically that RNF20 is important for stage specific gene expression and thereby physiology. It is no wonder then that the authors of the current manuscript observe a myriad of defects in the various RNF20 mutants analyzed.

Specific Remarks to the Author

Figure 1. Transcriptional profiling of TOF hearts was done long after development is complete, so anything observed is related to post-natal physiology concerns, not development. Identified genes may also have a relevant impact on development, but the use of post-natal profiling to identify genes important for development is not a logical approach. There is no evidence that TOF hearts experience diminished SpO₂ in utero when development is occurring. Cyanotic heart disease only impacts SpO₂ after birth, well after cardiac morphogenesis is complete. The authors should also be aware that TOF, while classically defined by four anatomic concerns, is actually one anatomic defect: Malalignment of the conal septum. This singular anatomic abnormality is responsible for RVOT obstruction, overriding aorta, and VSD. The RV hypertrophy is a secondary (acquired) phenomenon.

1.1. In Figure 1a., the authors presented the evolution of the expression of three genes (SCARB1, IVD and AMD1) in comparison to the oxygen saturation measured in patients. Are these genes already known to be correlated to SpO₂ and could serve as positive and negative controls of this analysis?

1.2. The expression unit is missing for each plots of Figure 1a and must be added.

1.3. The number of genes positively and negatively correlated to SpO₂ should be added to the text and/or to Figure 1b.

1.4. In Figure 1c., the authors showed an heatmap representing differentially expressed genes correlating with differences in oxygen saturation. It is unclear how many genes are represented on this heatmap. 3,394 genes are indicated in the text but only 1,906 are significantly misregulated in Supplementary Table S2. Why this difference in gene number?

Precisions related to the DEG number (total, upregulated and downregulated) must be added into the text and in Figure 1c.

Figure 2. The authors perform transcriptional profiling on cellular differentiations and conclude the specific aspects of development are altered in specific ways based on pathway term analysis of DEGs. There is no due diligence or experimental design for the transcriptional profiling – how many replicates were performed? How many DEGs identified? What statistical thresholds were applied – in the methods it states a p-value of 0.05 was applied – was this an adjusted p-value? What timepoints were assessed? How were timepoints compared to one another? The results suggest that some aspects of heart development may be perturbed but specific statements about this are not possible to conclude from the data presented.

2.1. In Figure 2a., it would be useful to add directly to the figure the range of days of the mESC differentiation protocol where the different cell types shown appear.

2.2. In Figure 2c., the GFP⁺ gates appear different between Control and Rnf20^{+/-}. The authors should add the gating strategy in a Supplementary Figure?

2.3. To explain Figure 2e the authors said “At cardiomyocyte stage (day 10) Rnf20^{+/-} CMs were highly positive for cardiac Troponin I (cTnI), with a more organized sarcomeric structure and increased sarcomeric length, suggesting accelerated CM differentiation”.

- Immunofluorescence analysis being qualitative and not quantitative per se, what is meant by “highly positive for cardiac Troponin I”? If you are referring to the larger surface area of the Troponin I⁺ groups of cells in the Rnf20^{+/-} condition compared to control, an adequate area quantification followed by a statistical testing must be performed.

- A specific quantitative analysis of the sarcomere length must be performed to conclude on an “increased sarcomeric length”.

Figure 3.

The *Isl1*-Cre deletion of Rnf20 certainly results in a heart defect. Certainly, there is a heart defect. However, the authors claim the mutant hearts have persistent truncus arteriosus, which is unsubstantiated. Do the author mean there is a single outflow vessel? It is a combined aorta PA without separation, or an aorta or a pulmonary artery? These have distinct etiologies and causes. No common trunk is shown with detail enabling verification. No numbers are included in comparison between mutant or control, no statistics are provided. This is true for almost many claimed defects across the paper.

3.1. In Figure 3b., the authors used *Isl1* and *Nkx2-5* as marker genes for the second heart field and the first heart field respectively. However, *Nkx2-5* expression is not specific to the FHF, as it is also expressed by SHF progenitors at E7.75, as notably explain in the article where the data were generated and others (DOI: 10.1016/j.ydbio.2014.02.023).

3.2. To study cardiac contractility, the authors used MYOCYTER software and presented representative traces. To be able to conclude on abnormal contraction amplitude, a specific quantification of peaks amplitude should be performed on different biological replicates for both *Isl1*-cre;Rnf20^{fl/fl} and control traces and followed by statistical testing. These results must be added along with representative traces.

3.3. Regarding the number of *Isl1*-cre;Rnf20^{fl/fl} and control hearts showing arrhythmic behavior, a contingency table should be build and a statistical test should be applied to test the arrhythmic behavior to genotype association.

3.4. For the analysis of H2Bub1 deposition, the authors conclude on “a more pronounced decrease in the RV” compared to the LV in both *Isl1*-cre;Rnf20^{fl/fl} CMs and ECs. However, no direct comparison has been made, so this conclusion is premature. A direct statistical testing between *Isl1*-cre;Rnf20^{fl/fl} right ventricle and *Isl1*-cre;Rnf20^{fl/fl} left ventricle samples must be performed to be able to conclude this.

4.1. To investigate deeper the impact of Rnf20 loss in second heart field progenitors (*Isl1*⁺) of mouse E10.5 embryos, the authors performed a scRNA-seq analysis. In Figure 4a., the schematic explanation is misleading as it represents the looping heart tube only, but SHF progenitors (which are located in the embryo trunk, dorsally from the heart tube) were identified by the analysis. A more detailed explanation of the dissection strategy must be provided.

4.2. Related to Figure 4b., a UMAP or t-SNE representation of the control and Rnf20-loss samples separately must be provided in Supplement to present the cell distribution per genotype.

4.3. The authors used previously published gene sets for cell annotation of Figures 4b and 4f. Heatmaps showing the expression of these markers per cluster must be provided in Supplement to assess the quality of the annotation.

4.4. In Figures 4d. and 4g., the authors concluded on differences in cell type proportion between genotypes, but no statistical evidence was provided. It is unclear whether the authors processed different biological replicates per genotype, that could have allowed them to definitely conclude on cell type proportion differences. In the present version of the data analysis, no statistically robust conclusion can be claimed, as alternate explanations, for example dissection bias, could be implicated in the observed differences.

4.5. In Figure 4i., the authors identified five EC populations.

- A heatmap of top marker genes specific to each population must be provided in Supplement to better characterize each EC population.

- The authors provided some explanations for cluster EC3, EC4 and EC5, but no information is provided for cluster EC1 and EC2. What about these two clusters?

4.6. There are few details concerning the bioinformatic processing of the scRNA-seq data, in term of cell quality selection

(mitochondrial content, cell cycle regression), normalization and batch effect correction? It would be important for reproducibility aspect, to make available the entire code used in this study.

5.1. The authors performed in Figure 5 a similar analysis as Figure 3., but here focused on Rnf20-loss in EC specifically. Regarding the number of Rnf20iEC-KO and control hearts showing arrhythmic behavior, a contingency table should be build and a statistical test should be applied to test the arrhythmic behavior to genotype association.

5.2. As for Figure 3e., to be able to conclude on abnormal contraction amplitude in Figure 5d., a specific quantification of peaks amplitude on different biological replicates should be performed for both Rnf20iEC-KO and control traces, followed by statistical testing.

5.3. For the explanation of Figures 5k. and 5m., the authors said that the “ablation of Rnf20 in ECs at P1 resulted in lethality, disorganized ECs, abnormal CM alignment and sarcomeric organization”. Together with representative pictures, it will be more robust to provide quantifications of EC organization, CM alignment and sarcomeric organization on multiple cells, from different biological replicates, followed by an appropriate statistical test.

6.1. To investigate deeper the impact of Rnf20 loss in endothelial cells (Cdh5+) in mouse E11.5 embryos, the authors performed a scRNA-seq analysis. Figure 6 was constructed in the same way than Figure 4, so the comments are similar for both Figure. The schematic explanation presented in Figure 6a., is misleading as it represents only the heart, but SHF progenitors (which are located in the embryo trunk, dorsally from the heart tube) were identified by the analysis. A more detailed explanation of the dissection strategy must be provided.

6.2. Related to Figure 6b., a UMAP or t-SNE representation of the control and Rnf20-loss samples separately must be provided in Supplement to present the cell distribution per genotype.

6.3. It's unclear which marker genes the authors used for cell annotation of Figures 6b and 6f. If they are the same as for Figure 4, it must be indicated in the manuscript. Heatmaps showing the expression of these markers per cluster must be provided in Supplement to assess the quality of the annotation.

6.4. In Figures 6d. and 6g., the authors concluded on differences in cell type proportion between genotypes, but no statistical evidence was provided. It is unclear whether the authors processed different biological replicates per genotype, that could have allowed them to definitely conclude on cell type proportion differences. In the present version of the data analysis, no statistically robust conclusion can be claimed, as a dissection bias could be implicated in the observed differences.

7.1. To study cardiac contractility of rat cardiomyocytes co-cultured with HUVECs treated or not with siRNA, the authors used MYOCYTER software and presented representative traces in Figure 7j. To be able to conclude on arrhythmic events and abnormal contraction amplitude, specific analyses must be performed using Poincare plots and quantification of peaks amplitude, on different biological replicates for both control and conditions traces. These results must be added along with representative traces.

8.1. In Figure 8d., the authors presented representative contraction traces of rat cardiomyocytes co-cultured with HUVECs treated or not with siRNA. In the description of the Figure 8d., the authors conclude on the “overexpression of NPPA, LOXL2 and MIF resulted in highly arrhythmic beating behavior”. The conclusion is overstated as the related trace presented only a higher frequency but no obvious arrhythmic behavior. To be able to conclude on arrhythmic events and abnormal contraction amplitude, specific analyses must be performed using Poincare plots and quantification of peaks amplitude, on different biological replicates for both control and conditions traces. These results must be added along with representative traces.

Reviewer #3

(Remarks to the Author)

Dou et al. investigated the causes of Tetralogy of Fallot (ToF), one of the most common types of cyanotic Congenital Heart Disease (CHD), and discovered that the H2Bub ligase RNF20 was one of the down-regulated genes in patients with low peripheral oxygen saturation (<88%). Since RNF20 mutation has been associated with CHD, they decided to focus on the connection between RNF20 and heart development. Utilizing a series of cell culture, zebrafish, and conditional knockout mouse models, and combining these with sc-RNA-seq analysis, they concluded that RNF20-H2Bub mediates SHF formation by modulating optimal EndoMT during cardiac development. The discovery that Rnf20 in cardiac endothelial cells (ECs) is required for Second Heart Field (SHF) development, and its role in modulating the interaction between ECs and cardiomyocytes (CMs) through optimal TGF-beta and angiocrine signaling, is interesting. However, this paper falls short in providing mechanistic insights and could benefit from improved data interpretation, better organization, and a writing style more accessible to a broader audience.

Major comments:

1. The two key themes of this paper are TGF-beta and angiocrine signaling. However, the introduction provides an insufficient overview regarding the key knowledge gaps in the interaction between these two pathways. Additionally, they did not include two recent publications that are highly relevant to this paper: PMID: 38038666 and PMID: 37967007, particularly the study by Barish et al.

2. Their efforts using mouse embryonic stem cells (mESCs) demonstrated that Rnf20 haploinsufficiency promoted cardiomyocyte (CM) differentiation but inhibited endothelial cell (EC) differentiation. However, it is clear from their own and published results that Rnf20 haploinsufficiency, which is not equivalent to loss of function, has no impact on cardiogenesis during heart development in mice. Therefore, this set of data does not align well with the current research theme.

3. Their initial efforts using a zebrafish model provided a hint that Rnf20 might regulate Isl1-mediated Second Heart Field (SHF) development. This hypothesis was confirmed using an Isl1-Cre mouse model. They observed decreased cardiomyocyte (CM) proliferation while endothelial cells (ECs) were not affected, indicating an altered interaction between these two cell populations due to the absence of Rnf20. However, their conclusion that 'Rnf20 potentially plays a role in both CMs and endocardial ECs for SHF development' (page 10) is not clearly substantiated by their data.

4. The description of the subsequent scRNA-seq analysis in the Isl1-Cre mouse heart was diffuse and rambling, yet it ultimately supports a role for Rnf20 in SHF development. However, the interpretation of how RNF20 deficiency alters transcription in endocardial endothelial cells (ECs), particularly in terms of increased gene expression in response to TGF-beta, was not clearly conveyed. Additionally, the conclusion that endothelial cells (ECs) play an important role in cardiac morphogenesis appears redundant, considering the well-established role of ECs, particularly through endothelial-to-mesenchymal transition (EndoMT), in the formation of heart valves and septa.

5. Further genetic analysis in mice led to the intriguing discovery that Rnf20 in ECs is essential for SHF development. scRNA-seq revealed a potential mesenchymal characteristic in the EC population following Rnf20 ablation, hinting at enhancement of endMT. The inhibitory role of RNF20 in endMT was subsequently confirmed using two in vitro endMT assays.

6. They attempted to establish that RNF20-H2Bub regulates RNA polymerase II (RNA pol II) pause-release in TGF-beta genes in HUVECs using ChIP-seq analysis. However, it has already been clearly demonstrated that BRE1A (RNF20) and H2Bub are not associated with the release of RNA pol II into the gene body (PMID: 26279188). RNF20-H2Bub has been shown to promote chromatin accessibility during cardiomyocyte maturation (PMID: 37967007) and to modulate the enhancer activity of estrogen-responsive genes. The authors should investigate these possible mechanisms in the regulation of TGF-beta genes in ECs.

7. Finally, their ligand-receptor analysis led to a surprising discovery: the expression of angiocrines such as Nrg1, Nrp1, Igf1, and Igf2 was significantly decreased. Furthermore, the addition of Nrg1 and the knockdown of the TGF-beta effector SMAD4 could rescue the compromised cardiomyocyte (CM) proliferation caused by Rnf20 deficiency in a co-culture model. However, the explanation of the 'NicheNet' analysis was missing, particularly concerning its ability to identify altered signaling pathways with a sensitivity surpassing that of sc-RNA-seq. The input source for the 'NicheNet' analysis is unclear, but if sc-RNA-seq data were used, a closer integration of these results with their respective sc-RNA-seq dataset would have in a smoother presentation.

8. Furthermore, the mechanisms by which RNF20 may regulate the expression of angiocrine signaling genes need to be addressed.

Version 1:

Reviewer comments:

Reviewer #1

(Remarks to the Author)

The authors have largely addressed my comments/concerns. The focus on endocardial/endothelial functions of Rnf20 has greatly strengthened this manuscript.

Reviewer #2

(Remarks to the Author)

The previous version of this manuscript included a very large range of different experiments with different messages, some of which was not well validated. Now the authors focus on the endothelial function of RNF20 and TGF-b signaling. The focus on the endothelial function of RNF20 and the link with TGF-b signaling rather than "SHF specification" is appreciated. This allows allows more complete validation than the previous version of the manuscript and better reflects the novelty of this study. In this regard new experiments such as the depleted Rnf20 in endothelial cells by directed differentiation of mESCs is a nice addition to the endothelial-specific KO in-vivo. Given that the authors have previously published on the role of RNF20 in vessel growth (PMID: 39322771), they should include a detailed discussion of the specific advances in insight from the current manuscript over the prior one.

We would like to express our sincere gratitude to all the reviewers for their appreciation of our work and especially for their thoughtful and constructive comments, which have significantly helped us to improve the quality of our manuscript and to clarify several important points. In response to the reviewers' feedback, we have revised our focus to more specifically examine the role of RNF20 in endothelial cells, where *RNF20* expression is reduced in Tetralogy of Fallot (TOF) patients, as well as in multipotent second heart field (SHF) progenitors, which give rise to endothelial cells, cardiomyocytes, and other cardiac lineages, and display markedly higher RNF20 levels compared to first heart field progenitors. To address the concerns raised, we have conducted a series of additional experiments, which are outlined in detail in the following point-by-point response.

Reviewers' comments are in italic:

Reviewer #1 (Remarks to the Author):

In this manuscript, Drs. Cordero/Dobrova and colleagues examine the role of the histone H2B ubiquitin (H2Bub1) ligase RNF20 in heart development, and its possible association with congenital heart disease (CHD). An RNF20 mutation has been previously identified in CHD patients, with mouse and in vitro models demonstrating an important role for RNF20 in heart development and cardiomyocyte maturation, respectively.

Gene expression (RNAseq) is examined from surgically removed right ventricular outflow tract tissue in cyanotic Tetralogy of Fallot (ToF) patients. Of a large number of dysregulated genes, expression of RNF20 is significantly reduced. Analysis of heterozygous Rnf20 murine ESCs or Rnf20 morpholino targeted zebrafish embryos suggests an accelerated differentiation of cardiac progenitors to a cardiomyocyte fate, with potentially impaired endothelial differentiation. This is supported by RNAseq analysis of differentiated mESCs, which suggests expanded "first heart field" and decreased "second heart field" associated gene expression with decreased Rnf20. In mice, Isl1:Cre-based SHF deletion of Rnf20 results in a spectrum of cardiac defects including thinned ventricular walls and poor cardiac contractility, with embryonic death by E14.5. Similar phenotypes are noted with endothelial-specific deletion of Rnf20, suggesting a later function for Rnf20 (subsequent to function in the SHF progenitors) to modulate EC signaling to CMs for proper ECM formation and CM proliferation/maturation. This is supported in elegant EC/CM co-culture experiments where Rnf20 function is altered in ECs.

This work represents a broad examination of Rnf20 function in heart development, with extensive in vitro, in vivo and genomics analyses. Overall, the manuscript is clearly written, with well-presented data.

Response: We thank the reviewer for the appreciation of our manuscript and for the constructive comments, which have significantly helped us to improve the quality and clarity of our work.

Minor Comments:

1. A very minor point, but on page 4 Isl1 is discussed as a SHF marker, while it may be more specific to the aSHF? This is highly stage-specific.

Response: We agree that Isl1 expression is highly stage-specific. We have clarified this point in the revised manuscript (page 8, lines 12-15), as follows: Isl1 is a well-established marker of second heart field (SHF) progenitors in the mouse. While it is initially expressed in both anterior and posterior SHF domains, its expression becomes progressively restricted to anterior SHF derivatives, particularly those contributing to the outflow tract and right ventricle^{13, 14, 47, 48}.

2. Top of page 8: "haploinsufficient cells" should be clarified to the type of cell (mESCs in this case).

Response: We thank the reviewer for highlighting these shortcomings in the original text. In the revised manuscript, we have removed the data obtained using *Rnf20* haploinsufficient (*Rnf20*^{+/-}) cells, which carry only one functional allele. Instead, we now employ a cell culture model that more accurately supports the central message of the manuscript (Figure 6f-k). Based on the collective feedback from all reviewers, we decided to sharpen the focus of the manuscript on the essential role of *Rnf20* specifically in endothelial cells.

To enable a better comparison with our in vivo experimental models, we depleted *Rnf20* in mesodermal precursors and assessed its impact on endothelial cells using a protocol for directed differentiation of mESCs in endothelial cells (Fig. 6f). Flow cytometric analysis for PECAM1 expression at day 7 (D7) revealed no significant difference in EC yield between control and *Rnf20*-deficient cells (Fig. 6g, 6h), suggesting that *Rnf20* is not essential for initial EC specification. However, transcriptomic analysis of sorted PECAM1⁺ cells showed a marked upregulation of genes associated with positive regulation of cell locomotion and extracellular matrix organization, indicative of a more migratory and mesenchymal-like phenotype (Fig. 6i, Supplementary Fig. 8c, Supplementary Table 4). Conversely, genes involved in response to growth factors and receptor tyrosine kinase signaling were significantly downregulated,

consistent with our in vivo observations. Comparison of differentially expressed genes in Rnf20-deficient mESC-derived ECs and HUVECs revealed a shared enrichment of genes promoting cell locomotion among the upregulated set, and genes regulating growth factor responses among the downregulated set, pointing to a conserved role of Rnf20 in endothelial cells (Fig. 6j).

Fig. 6

Fig. 6: Endothelial RNF20 inhibits EndMT. **f** Schematic representation of the directed mESC differentiation system into endothelial cells with lentiviral transduction of a control and *shRnf20* construct at the mesoderm stage (mESC, mouse embryonic stem cell; MES, mesoderm; EC, endothelial cells). **g** Representative FACS analyses of Pecam1⁺ (CD31⁺) endothelial cells at day 7 of differentiation depicted in **f**. Highly CD31-positive cells were FACS sorted for RNA-seq. **h** Percentage of Pecam1⁺ ECs at day 7. **i** Representative GO terms of genes enriched among significantly upregulated (violet) and downregulated (blue) genes in RNA-seq datasets of sorted Pecam1⁺ *shRnf20* ECs versus control differentiated ECs (*p*adj < 0.05). **j** Representative GO terms enriched among shared significantly upregulated (red) and downregulated (blue) transcripts (*p*adj < 0.05 for both datasets) upon RNF20 depletion in both HUVECs and mouse ESC differentiated ECs. **k** Immunostaining for Cdh5, Snai1 and DAPI of day 14 ECs derived from mESCs.

By day 14 of directed EC differentiation, control ECs formed organized monolayers with robust VE-Cadherin (Cdh5) expression, whereas Rnf20-deficient cells displayed disrupted monolayer integrity and a substantial loss of VE-Cadherin expression (Fig. 6k). Notably, many of these cells expressed the mesenchymal marker Snai1, further supporting premature or aberrant induction of EndMT in the absence of Rnf20. Together, these findings indicate that while Rnf20 is dispensable for early EC specification, it is essential for maintaining endothelial identity and preventing aberrant mesenchymal transition.

3. The acronym ECs is used for both endocardial cells (Introduction and some text) and endothelial cells (Methods page 24 and other sections). The mESC differentiation protocols used are making endothelial cells, as opposed to endocardial cells. This should be clarified.

Response: We thank the reviewer for pointing this out. In the revised manuscript, we now use the abbreviation "ECs" for endothelial cells, while "endocardial endothelial cells" are referred to as "endocardial ECs" to distinguish between the two.

4. Figure 3: PTA and other histological findings should be more clearly demonstrated for non-experts, as should measures of RV size.

Response: We thank the reviewer for this comment. We have now added dotted lines in Fig. 2d to indicate the location of the great vessels (aorta and pulmonary artery), in contrast to the single vessel observed in the knockout. Additionally, we have included a quantification of ventricular size in Fig. 2e.

Fig. 2

Major Comments:

1. For the initial patient sample RNAseq analysis, it is not clear how many samples were used, what “control” healthy tissue was used, etc.

Response: We apologize for not clearly conveying the rationale behind including these datasets. The critical role of *RNF20* in heart development has been demonstrated through exome sequencing studies identifying a de novo *RNF20* mutation in patients with congenital heart disease (CHD) ⁶. In addition, previously published bulk RNA-sequencing data from

Supplementary Fig. 1a

Supplementary Fig. 1a Volcano plot of differentially expressed data taken from (GSE256516). Dots representing *RNF20* as well as CHD-related genes, such as *JAG1*, *NOTCH1*, *NOTCH2*, and *TBX5* are highlighted.

Fig. 1: *RNF20* levels are reduced in TOF patients and positively correlate with SpO₂ levels. **a** Dot plot showing the expression of *RNF20* in different cell populations separated by Donor and Tetralogy of Fallot (TOF) patients from (GSE203274). P-value after Wilcox test from FindMarkers. **b** Representative genes and enriched GO terms in marker genes for EC2. **c** Spearman correlation between *RNF20* expression and clinical traits in TOF patients. **d** Dot plot showing the correlation between SpO₂ levels and *RNF20* gene expression in samples from TOF patients. **e** Relative mRNA expression of *RNF20*, *GLUT1*, *PDK1* and *LOX* in HUVECs cultured in normoxia and hypoxia (1% oxygen level, 24h) determined by qPCR.

Fig. 1

Tetralogy of Fallot (TOF) patients show reduced *RNF20* expression (Supplementary Fig. 1a). Analysis of recent single-cell RNA sequencing data further supports this finding, revealing downregulation of *RNF20* specifically in endothelial cells, mural cells, and cardiac fibroblasts from TOF patients compared to controls (Figure 1a, Supplementary Fig.1b–d).

In the initial version of the manuscript, we utilized RNA-sequencing data from a cohort of TOF patients to identify gene expression signatures associated with available clinical parameters (n=25, Supplementary Table 1). However, this approach somewhat diluted the central message of the study. In the revised manuscript, we therefore focused our use of the RNA-sequencing data to assess correlations between *RNF20* expression and key clinical parameters, including age, peripheral oxygen saturation (SpO₂) at admission (preoperative), and right ventricular pressure, resulting from structural abnormalities in these patients. Our analysis revealed a significant positive correlation between *RNF20* expression and SpO₂, suggesting that reduced oxygen levels after birth may contribute to *RNF20* downregulation. In line with the significant positive correlation observed between *RNF20* expression and SpO₂, experimental exposure of cells to hypoxic conditions resulted in a marked reduction in *RNF20* mRNA levels, accompanied by altered expression of hypoxia-induced genes (Figure 1d, 1h). Postnatal hypoxia can further induce alterations in cardiac structure, function, and cellular signaling. Therefore, our manuscript includes data from a conditional postnatal *RNF20* deletion model to investigate the consequences of endothelial-specific *RNF20* loss after birth (Fig. 4k–n).

Notably, in a recent study from our group, we identified a critical role for *RNF20* in regulating HIF1 α protein levels³⁶. Conversely, hypoxia affects *RNF20* expression, suggesting a potential interplay between *RNF20* and HIF1 α in congenital heart defects associated with chronic hypoxia. We have incorporated this point into the Discussion (page 21, lines 5-15).

2. The zebrafish data as presented is difficult to interpret. What controls (RNA rescue, etc.) have been attempted in the MO model? There is a clear effect on overall development (Figure S1F). If premature differentiation of SHF cells is being claimed in this model, Isl1 expression and other analysis of SHF markers should be included.

Response: We apologize for not clearly explaining the experimental setup. In the revised manuscript, following the reviewers' comments and recommendations, we have refocused our study on the key role of *RNF20* in endothelial cells and have conducted a more detailed characterization of the mechanisms by which endothelial *Rnf20* regulates heart development and function in the mouse. Accordingly, we have removed the previously included zebrafish data, as it falls outside the revised scope of the manuscript.

3. Figures 3 and S2: it is confusing why effects on CM death (at e11.5) and proliferation are seen in both the LV and RV, as well as H2Bub1 immunostaining, given that a SHF-specific deletion of *Rnf20* was carried out. Presumably, LV CMs and ECs should maintain functional *Rnf20*.

Response: We thank the reviewer for this important question. We believe that cardiomyocyte death observed at E11.5 is a secondary consequence of cardiovascular defects, as such cell death typically occurs within hours to 1-2 days following the onset of hemodynamic or structural disturbances. Therefore, the majority of our functional and cellular analyses were performed at developmental stages preceding detectable cell death.

It is indeed surprising that H2Bub1 levels are reduced in the left ventricle (LV), given that RNF20 is specifically depleted in the right ventricle (RV), but not in the LV (Fig. 2b, Supplementary Fig. 2f, 2g). H2Bub1 levels are dynamically regulated by the coordinated actions of E3 ubiquitin ligases, such as RNF20/RNF40, and deubiquitinases (DUBs). Indeed, qPCR analysis on microdissected RV and LV tissue revealed increased *Rnf40* expression in the RV, likely reflecting a compensatory response, and decreased expression in the LV (Supplementary Fig. 2h). Thus, although *Rnf20* is specifically deleted in SHF derivatives, the secondary downregulation of *Rnf40* in the LV may contribute to the observed reduction of H2Bub1 levels (Supplementary Fig. 2f, 2g).

Fig. 2

Supplementary Fig. 2h

Fig. 2b Relative *Rnf20* mRNA expression in dissected RV and LV of E10.5 control and *Isl1^{cre/+}Rnf20^{fl/fl}* hearts.

Fig. 2h Relative *Rnf40* mRNA expression in RV and LV of E10.5 control and *Rnf20^{Isl1^{cre/+}Rnf20^{fl/fl}}* hearts.

4. The interpretation of scRNAseq data in the first paragraph of page 11 is puzzling: greater “aSHF” and “pSHF” cell proportions are seen in the *Rnf20* lof model, which is attributed to potential “defects in aSHF progenitor differentiation”. This seems to be directly opposed to conclusions for mESC and zebrafish analyses. The cell types as presented are also poorly defined.

Response: We apologize for not clearly articulating our findings in the previous version. We acknowledge that the cell types, as originally presented, were not well defined. In the revised manuscript, we have clarified the identity of the relevant cell populations and provided additional details to support their definition (Fig. 3b and Supplementary Fig. 3a, b).

Fig. 3

Fig. 3: *Rnf20* loss alters cardiac cellular composition. **a** Schematic representation of the heart dissection procedure used for scRNA-Seq. Hearts from five control and *Isl1^{cre/+}Rnf20^{fl/fl}* embryos ($n = 5$) were dissected for the experiment. **b** t-Stochastic neighbour embedding (t-SNE) plot of 10x genomics scRNA-Seq datasets of E10.5 control and *Isl1^{cre/+}Rnf20^{fl/fl}* hearts. CM1 and CM2, cardiomyocytes 1 and 2; EndoV, endocardial valve cells; AVcM, atrioventricular canal mesenchyme; EpiC1 and EpiC2, epicardial cells 1 and 2; EC/EndoC, endothelial cells/endocardial endothelial cells; dOFTm, dorsal outflow tract mesenchyme; dOFTe, dorsal outflow tract epithelium; cMES1 and cMES2, cardiac mesenchyme 1 and 2; OFT-MES, outflow tract mesenchyme; Myeloid cells. **c** Percentage of cells within the different clusters colored by cell types. P-value obtained from binomial exact test.

Our RNA-seq analyses were performed on dissected E10.5 hearts, which no longer contain undifferentiated progenitor cells. Our intention was to refer specifically to aSHF-derived lineages. In the revised manuscript, we have clarified the annotation of cell populations identified in the single-cell RNA-seq analysis. The population previously labeled as aSHF corresponds to cells in the distal outflow tract mesenchyme (dOFTm), which exhibit mesenchymal characteristics (Fig. 3b). This population is significantly expanded in the mutants (Fig. 3c), consistent with the increased EndMT observed in the commonly used OFT cushion EndMT assay (Fig. 6a), and with the enhanced migratory capacity of endothelial cells upon *Rnf20* loss of function (Fig. 6b–e).

The population previously labeled as pSHF corresponds to cardiac mesenchymal cells expressing pSHF markers, now designated as cMES2. The observed increase in this population is likely due to enhanced EndMT of pSHF-derived endocardial cells following *Rnf20* loss.

Supplementary Fig. 3.

Supplementary Fig. 3. Single cell analysis of control and *Isl1^{cre/+}Rnf20^{fl/fl}* hearts. **a** t-SNE plot separated by genotype and colored according to cell type. **b** Heatmap showing the top 3 markers for each cell type on the left, and markers used to define the cell populations from Fig. 3b on the right.

5. Overall, the statement that “*RNF20* governs the diversification of cardiac progenitors in the first and second heart fields and is crucial for the development of the SHF” (Discussion, page 18) requires a more detailed analysis of FHF and SHF markers at appropriate stages prior to heart assembly. Many of the gene expression changes noted in e10.5-14.5 hearts may be secondary to effects on heart function and development, as also seen in EC-specific *Rnf20* lof models used.

Response: We highly appreciate the reviewer’s thoughtful comment regarding the need for a more detailed analysis of FHF and SHF markers at earlier developmental stages. In light of this and the feedback from the other reviewers, we have revised our focus to specifically examine the role of *Rnf20* in endothelial cells, where *RNF20* expression is reduced in TOF patients, as well as in second heart field progenitors, which exhibit significantly higher *Rnf20* levels compared to first heart field progenitors (Fig. 2a). Importantly, *Rnf20* is highly expressed in the *Isl1*-derived endocardial lineages (Supplementary Fig. 3c-e), where our findings reveal its critical role in limiting EndMT and maintaining physiological angiocrine signaling, which is essential for normal heart development and function.

Fig. 2

Supplementary Fig. 3

Fig. 2a Dot plot of *Rnf20* expression in FHF and SHF cells of sorted *Isl1*⁺ and *NKX2.5*⁺ E7.75 embryos⁴⁴. First heart field (FHF) and second heart field (SHF). FHF population were defined as cells sorted by *Nkx2-5* with negligible expression of *Isl1*. SHF cells were sorted by *Isl1*. $P=0.021$ after Wilcox test from FindMarkers. **Supplementary Fig. 3:** **c** Dot plot illustrating the expression of *Isl1* and *Rnf20* across various cell types. **d** UMAP plot depicting cell clusters within the *Isl1* lineage in E8.75 embryos⁴⁹. **e** Dot plot displaying marker genes for CMs, ECs, and proliferative populations.

While our data indicate that *Rnf20* influences cardiac progenitor diversification, as noted by the reviewer, a comprehensive investigation of its role in the early specification of FHF and SHF lineages would require additional in-depth analyses at earlier time points and the use of earlier Cre drivers. Such an approach, although important, would extend beyond the scope of the current study and detract from the central message of the revised manuscript.

6. The much greater severity of the EC-specific deletion of *Rnf20* vs that seen with SHF/*Isl1*-based deletion should be discussed in greater detail. Further, data should be presented on the extent of Cre-mediated removal of both *Rnf20* floxed alleles in these models (is full deletion being reached)?

Response: We thank the reviewer for highlighting this point. In our study, we employed two endothelial Cre lines: the constitutive *Tie2*-Cre and the tamoxifen-inducible *Cdh5*-Cre, induced at E8.5. *Tie2* is expressed early (starting at E7.0) in endothelial lineages, including angioblasts, the yolk sac, embryonic vasculature, and endocardium. This early expression likely accounts for the more severe phenotype observed in *Tie2*^{Cre}*Rnf20*^{fl/fl} mutant embryos. To compare the endothelial-specific and *Isl1*-Cre-mediated deletion of *Rnf20*, we employed the tamoxifen-inducible *Cdh5*-CreERT2. The E8.5 induction time point was chosen because *Isl1*-Cre-mediated recombination is efficient at this stage. Correspondingly, *Cdh5*-CreERT2 mediates recombination in *Cdh5*-expressing endothelial cells from E8.5 onward, targeting primarily mature endothelial cells and the endocardium. We have also performed qPCR analysis to confirm the efficient deletion of *Rnf20* in the different mutant lines.

Fig. 2b**Fig. 4b****Fig. 2b** Relative *Rnf20* mRNA expression in dissected RV and LV of E10.5 control and *Isl1^{cre/+}Rnf20^{fl/fl}* hearts.**Fig. 4b** Relative *Rnf20* mRNA expression in ECs isolated from E11.5 control (*Cdh5^{CreERT2}neg-Rnf20^{fl/fl}*) and *Rnf20^{IEC-KO}* (*Cdh5^{CreERT2}pos-Rnf20^{fl/fl}*) embryos.

This has now been addressed in the Results section (page 11, lines 20-21; Page 12, lines 1-7) and the Discussion section, where we delineate the phenotypic similarities and differences observed following *Rnf20* ablation in the second heart field versus the endothelium (page 20, lines 23-27; page 21, lines 1-4).

Reviewer #2 (Remarks to the Author):

Y. Dou, N. Tetik-Elsherbiny, R. Gao and collaborators proposed a manuscript entitled “RNF20 at the Nexus of Cardiac Progenitor Diversification and Angiocrine Signaling in Congenital Heart Disease” for publication in Nature Communications. In this study the authors highlighted the important role of Rnf20, an ubiquitin ligase responsible for the monoubiquitination H2Bub1, in cardiac development.

*The authors state the following: (1) downregulation of RNF20 in patients presenting a Tetralogy of Fallot with oxygen saturation below 88%; (2) precocious differentiation of cardiac progenitors due to Rnf20 downregulation in mESC in vitro and in zebrafish in vivo; (3) impaired endothelial differentiation in vitro after Rnf20 downregulation; (4) decreased cardiomyocyte proliferation together with a thinner compact and trabecular myocardium in *Isl1-cre;Rnf20fl/fl* mouse embryos at E10.5; (5) shortened outflow tract, a small right ventricle and a decrease ventricular cardiomyocyte proliferation in *Cdh5-cre;Rnf20fl/fl* mouse embryos at E11.5; (6) lethality in *Cdh5-cre;Rnf20fl/fl* mouse embryos at P1, accompanied by a poor cardiomyocyte organization; (7) role for Rnf20 in the inhibition of the endothelial-to-mesenchymal transition in the cardiac outflow tract; (8) decreased PolII pausing index at upregulated genes and TGF- β effectors after Rnf20 knockdown in endothelial cells in vitro; (9) increased TGF- β signaling in the context of Rnf20 loss, which is associated with the decreased cardiomyocyte proliferation. The authors showed that Rnf20 acts both in cardiac progenitors of the second heart field and in endocardial cells, supporting it as a candidate in the development of congenital heart defects. The authors perform an extensive investigation of the role of Rnf20 in both cardiac progenitors and endocardial cells, using a significant combination of approaches and models.*

The proposed study will be of interest for investigators in the field of cardiac development and congenital heart disease.

Unfortunately, a rigorously demonstrated mechanistic understanding of why diminished RNF20 dose causes the observed effects or observed changes in gene expression is not offered. Instead, a very broad range of models and assays is brought to bear, confirming the importance of RNF20 without providing insight into its specific role. Some experiments are over interpreted, and many claims are made that are not substantiated by the data included in the manuscript, as detailed below. A good example is the aggressive claim that RNF20 governs the diversification of cardiac progenitors in the first and second heart fields. Most certainly the authors demonstrate that the gene is required in the SHF. However, the authors fail to rigorously demonstrate a requirement for RNF20 in the diversification of cardiac progenitors.

A recently published paper on the role of RNF20 in heart development and maturation (PMID: 37967007) indicated a time-specific role for RNF20 across cardiac maturation– basically that RNF20 is important for stage specific gene expression and thereby physiology. It is no wonder then that the authors of the current manuscript observe a myriad of defects in the various RNF20 mutants analyzed.

Response: We thank the reviewer for the constructive comments, which have helped us to significantly improve and refine the focus of our manuscript. As the reviewer correctly noted, Rnf20 plays a key role in transcriptional regulation, and its deficiency is expected to result in pleiotropic effects, including multiorgan abnormalities. In light of this and the feedback from the other reviewers, we have revised our focus to more specifically examine the role of RNF20 in endothelial cells, where *RNF20* expression is reduced in Tetralogy of Fallot (TOF) patients, as well as in multipotent second heart field (SHF) progenitors, which give rise to endothelial cells, cardiomyocytes, and other cardiac lineages, and display markedly higher RNF20 levels compared to first heart field progenitors. We now provide more detailed mechanistic insights into how RNF20 controls Endothelial-to-Mesenchymal Transition (EndMT) and angiocrine signaling pathways that are essential for cardiogenesis.

Specific Remarks to the Author

Figure 1. Transcriptional profiling of TOF hearts was done long after development is complete, so anything observed is related to post-natal physiology concerns, not development. Identified genes may also have a relevant impact on development, but the use of post-natal profiling to identify genes important for development is not a logical approach. There is no evidence that TOF hearts experience diminished SpO2 in utero when development

is occurring. Cyanotic heart disease only impacts SpO₂ after birth, well after cardiac morphogenesis is complete.

The authors should also be aware that TOF, while classically defined by four anatomic concerns, is actually one anatomic defect: Malalignment of the conal septum. This singular anatomic abnormality is responsible for RVOT obstruction, overriding aorta, and VSD. The RV hypertrophy is a secondary (acquired) phenomenon.

Response: We fully agree with the reviewer and apologize for not clearly conveying the rationale behind including these datasets. The critical role of *RNF20* in heart development has been demonstrated through exome sequencing studies identifying a de novo *RNF20* mutation in patients with congenital heart disease (CHD)⁶. In addition, previously published bulk RNA-sequencing data from Tetralogy of Fallot (TOF) patients show reduced *RNF20* expression (Supplementary Fig. 1a). Analysis of recent single-cell RNA sequencing data further supports this finding, revealing downregulation of *RNF20* specifically in endothelial cells, mural cells, and cardiac fibroblasts from TOF patients compared to controls (Figure 1a, Supplementary Fig.1b–d).

In the initial version of the manuscript, we utilized RNA-sequencing data from a cohort of TOF patients to identify gene expression signatures associated with available clinical parameters (n=25, Supplementary Table 1). However, this approach somewhat diluted the central message of the study. In the revised manuscript, we therefore focused our use of the RNA-sequencing data to assess correlations between *RNF20* expression and key clinical parameters, including age, peripheral oxygen saturation (SpO₂) at admission (preoperative), and right ventricular pressure, resulting from structural abnormalities in these patients, as noted by the reviewer. A more detailed analysis of the correlation between different gene sets and various clinical parameters will be addressed in a separate manuscript.

Our analysis revealed a significant positive correlation between *RNF20* expression and SpO₂, suggesting that reduced oxygen levels after birth may contribute to *RNF20* downregulation. In line with the significant positive correlation observed between *RNF20* expression and SpO₂, experimental exposure of cells to hypoxic conditions resulted in a marked reduction in *RNF20* mRNA levels, accompanied by altered expression of hypoxia-induced genes (Figure 1d, 1e). Postnatal hypoxia can further induce alterations in cardiac structure, function, and cellular signaling. Therefore, our manuscript includes data from a conditional postnatal *Rnf20* deletion model to investigate the consequences of endothelial-specific RNF20 loss after birth (Fig. 4k–n).

Notably, in a recent study from our group, we identified a critical role for RNF20 in regulating HIF1 α protein levels³⁶. Conversely, hypoxia affects *RNF20* expression, suggesting a potential

interplay between RNF20 and HIF1 α in congenital heart defects associated with chronic hypoxia. We have incorporated this point into the Discussion (page 21, lines 5-15).

Supplementary Fig. 1a

Supplementary Fig. 1a Volcano plot of differentially expressed data taken from (GSE256516). Dots representing *RNF20* as well as CHD-related genes, such as *JAG1*, *NOTCH1*, *NOTCH2*, and *TBX5* are highlighted.

Fig. 1: *RNF20* levels are reduced in TOF patients and positively correlate with SpO₂ levels. **a** Dot plot showing the expression of *RNF20* in different cell populations separated by Donor and Tetralogy of Fallot (TOF) patients from (GSE203274). P-value after Wilcoxon test from FindMarkers. **b** Representative genes and enriched GO terms in marker genes for EC2. **c** Spearman correlation between *RNF20* expression and clinical traits in TOF patients. **d** Dot plot showing the correlation between SpO₂ levels and *RNF20* gene expression in samples from TOF patients. **e** Relative mRNA expression of *RNF20*, *GLUT1*, *PDK1* and *LOX* in HUVECs cultured in normoxia and hypoxia (1% oxygen level, 24h) determined by qPCR.

Fig. 1

1.1. In Figure 1a., the authors presented the evolution of the expression of three genes (*SCARB1*, *IVD* and *AMD1*) in comparison to the oxygen saturation measured in patients. Are these genes already known to be correlated to SpO₂ and could serve as positive and negative controls of this analysis?

Response: As mentioned above, in the revised manuscript we have sharpened the focus on *RNF20* and now show that *RNF20* levels correlate significantly with SpO₂ levels, but not with age or right ventricular pressure (Fig. 1c, Fig. 1d). In addition, we demonstrate that *RNF20* expression is significantly reduced in HUVECs upon exposure to hypoxia (Fig. 1e).

1.2. *The expression unit is missing for each plots of Figure 1a and must be added.*

Response: We are sorry for these deficiencies in the text, the expression units have been added to the revised Figure 1d.

1.3. *The number of genes positively and negatively correlated to SpO2 should be added to the text and/or to Figure 1b.*

Response: As mentioned above, in the revised manuscript we have refined the focus on *RNF20*, and therefore the previously included data have been omitted. Instead, we now provide Supplementary Table 7, which lists genes showing significant positive and negative correlations with *RNF20* expression in TOF patients, based on Spearman correlation analysis. In addition, we include a heatmap highlighting a subset of negatively correlated genes involved in TGF- β signaling (Fig. 8b).

1.4. *In Figure 1c., the authors showed an heatmap representing differentially expressed genes correlating with differences in oxygen saturation. It is unclear how many genes are represented on this heatmap. 3,394 genes are indicated in the text but only 1,906 are significantly misregulated in Supplementary Table S2. Why this difference in gene number? Precisions related to the DEG number (total, upregulated and downregulated) must be added into the text and in Figure 1c.*

Response: We are sorry for not clarifying this sufficiently. The 3,394 genes are indicated in the text, are these which show positive or negative correlation. The genes presented in Supplementary Table S2 are genes that are differentially expressed between the two groups (in patients with SpO₂<88% and patients with SpO₂ \geq 88%). However, in light of the reviewers' comments, we have now decided to focus specifically on *RNF20* and its correlation with clinical parameters. A more detailed analysis of the correlation between different gene sets and various clinical parameters will be addressed in a separate manuscript.

Figure 2. The authors perform transcriptional profiling on cellular differentiations and conclude the specific aspects of development are altered in specific ways based on pathway term analysis of DEGs. There is no due diligence or experimental design for the transcriptional

profiling – how many replicates were performed? How many DEGs identified? What statistical thresholds were applied – in the methods it states a p-value of 0.05 was applied – was this an adjusted p-value? What timepoints were assessed? How were timepoints compared to one another? The results suggest that some aspects of heart development may be perturbed but specific statements about this are not possible to conclude from the data presented.

Response: We thank the reviewer for highlighting these shortcomings in the original text. In the revised manuscript, we have removed the data obtained using *Rnf20* haploinsufficient (*Rnf20*^{+/-}) cells, which carry only one functional allele. Instead, we now employ a cell culture model that more accurately supports the central message of the revised manuscript (Figure 6f–k).

Fig. 6

Fig. 6: Endothelial RNF20 inhibits EndMT. **f** Schematic representation of the directed mESC differentiation system into endothelial cells with lentiviral transduction of a control and shRnf20 construct at the mesoderm stage (mESC, mouse embryonic stem cell; MES, mesoderm; EC, endothelial cells). **g** Representative FACS analyses of

Pecam1⁺ (CD31⁺) endothelial cells at day 7 of differentiation depicted in f. Highly CD31-positive cells were FACS sorted for RNA-seq. h Percentage of Pecam1⁺ ECs at day 7. i Representative GO terms of genes enriched among significantly upregulated (violet) and downregulated (blue) genes in RNA-seq datasets of sorted Pecam1⁺ *shRnf20* ECs versus control differentiated ECs (padj < 0.05). j Representative GO terms enriched among shared significantly upregulated (red) and downregulated (blue) transcripts (padj < 0.05 for both datasets) upon RNF20 depletion in both HUVECs and mouse ESC differentiated ECs. k Immunostaining for Cdh5, Snai1 and DAPI of day 14 ECs derived from mESCs.

As already mentioned, based on the collective feedback from all reviewers, we decided to sharpen the focus of the manuscript on the essential role of *Rnf20* specifically in endothelial cells. To enable a better comparison with our *in vivo* experiments, we depleted *Rnf20* in mesodermal precursors and assessed its impact on endothelial cells using a protocol for directed differentiation of mESCs in endothelial cells (Fig. 6f).

Flow cytometric analysis for PECAM1 expression at day 7 (D7) revealed no significant difference in EC yield between control and *Rnf20*-deficient cells (Fig. 6g, 6h), suggesting that *Rnf20* is not essential for initial EC specification. However, transcriptomic analysis of sorted PECAM1⁺ cells showed a marked upregulation of genes associated with positive regulation of cell locomotion and extracellular matrix organization, indicative of a more migratory and mesenchymal-like phenotype (Fig. 6i, Supplementary Fig. 8c, Supplementary Table 4). Conversely, genes involved in response to growth factors and receptor tyrosine kinase signaling were significantly downregulated, consistent with our *in vivo* observations. Comparison of differentially expressed genes in *Rnf20*-deficient mESC-derived ECs and HUVECs revealed a shared enrichment of genes promoting cell locomotion among the upregulated set, and genes regulating growth factor responses among the downregulated set, pointing to a conserved role of *Rnf20* in endothelial cells (Fig. 6j).

By day 14 of directed EC differentiation, control ECs formed organized monolayers with robust VE-Cadherin (*Cdh5*) expression, whereas *Rnf20*-deficient cells displayed disrupted monolayer integrity and a substantial loss of VE-Cadherin expression (Fig. 6k). Notably, many of these cells expressed the mesenchymal marker *Snai1*, further supporting premature or aberrant induction of EndMT in the absence of *Rnf20*. Together, these findings indicate that while *Rnf20* is dispensable for early EC specification, it is essential for maintaining endothelial identity and preventing aberrant mesenchymal transition.

For the RNA-seq data two independent pools were generated and adjusted p-value (padj) less than 0.05 was used for the analysis, this is now specified in the figure legend (Fig. 6i, 6j). In addition, we have included a schematic diagram outlining the experimental setup and differentiation time points used in the analyses presented in Fig. 6f-k.

2.1. In Figure 2a., it would be useful to add directly to the figure the range of days of the mESC differentiation protocol where the different cell types shown appear.

Response: As mentioned above, to allow a more direct comparison with our in vivo experiments, we depleted *Rnf20* in mesodermal precursors (Fig. 6f) and assessed its impact on endothelial cell differentiation at days 7 and 14 of differentiation, using a protocol for the directed differentiation of mESCs into endothelial cells. The time points of *Rnf20* silencing and subsequent analyses are indicated in Fig. 6f.

2.2. In Figure 2c., the GFP+ gates appear different between Control and *Rnf20*^{+/-}. The authors should add the gating strategy in a Supplementary Figure?

Response: Indeed, there was an error in the gating strategy in the initial submission. A gating strategy for our revised experimental setup (Fig. 6f-k) is included in Supplementary Fig. 8c.

2.3. To explain Figure 2e the authors said “At cardiomyocyte stage (day 10) *Rnf20*^{+/-} CMs were highly positive for cardiac Troponin I (cTnI), with a more organized sarcomeric structure and increased sarcomeric length, suggesting accelerated CM differentiation”.

- Immunofluorescence analysis being qualitative and not quantitative per se, what is meant by “highly positive for cardiac Troponin I”? If you are referring to the larger surface area of the Troponin I+ groups of cells in the *Rnf20*^{+/-} condition compared to control, an adequate area quantification followed by a statistical testing must be performed.
- A specific quantitative analysis of the sarcomere length must be performed to conclude on an “increased sarcomeric length”.

Response: In response to the reviewers' feedback, we have refined the focus of our manuscript to better characterize the role of *Rnf20* in endothelial cells during heart development, as its function in cardiomyocytes has already been well described. To avoid diluting the central message, we now limit our phenotypic characterization to the endothelial lineage.

Figure 3. The Isl1-Cre deletion of Rnf20 certainly results in a heart defect. Certainly, there is a heart defect. However, the authors claim the mutant hearts have persistent truncus arteriosus, which is unsubstantiated. Do the author mean there is a single outflow vessel? It is a combined aorta PA without separation, or an aorta or a pulmonary artery? These have distinct etiologies and causes. No common trunk is shown with detail enabling verification. No numbers are included in comparison between mutant or control, no statistics are provided. This is true for almost many claimed defects across the paper.

Response: We thank the reviewer for pointing this out. Indeed, as *Isl1^{cre/+}Rnf20^{fl/fl}* embryos do not survive beyond E14.5 and already exhibit developmental delay at this stage, we agree that it is more accurate to describe the phenotype as a "single outflow vessel," as the reviewer suggested. We have now included quantitative data in Figure S2d to support this observation.

Supplementary Fig. S2b

	Ao/PA	Single vessel	Total
Control (WT)	6	0	6
Isl1^{cre/+}Rnf20^{fl/fl}	0	5	5
Total	6	5	11
Fisher's exact test			p = 0.0022

Supplementary Fig. 2b Contingency table showing the incidence of single vessel in control and *Isl1^{cre/+}Rnf20^{fl/fl}* hearts. Statistical significance was assessed using Fisher's exact test (p = 0.0022).

3.1. In Figure 3b., the authors used *Isl1* and *Nkx2-5* as marker genes for the second heart field and the first heart field respectively. However, *Nkx2-5* expression is not specific to the FHF, as it is also expressed by SHF progenitors at E7.75, as notably explain in the article where the data were generated and others (DOI: 10.1016/j.ydbio.2014.02.023).

Response: We thank the reviewer for pointing this out. We have now reanalyzed the single-cell RNA sequencing (scRNA-seq) data from E7.75 mouse embryos⁴⁴ and observed

significantly higher expression of *Rnf20* in SHF progenitors (*Isl1^{pos}*) compared to FHF precursors (*Nkx2.5^{pos} Isl1^{neg}*) (Fig. 2a). To further illustrate the distinction between these populations, we have also plotted established markers for FHF and SHF, which demonstrate clear separation between the two lineages.

Fig. 2a Dot plot of *Rnf20* expression in FHF and SHF cells of sorted *Isl1⁺* and *NKX2.5⁺* E7.75 embryos⁴⁴. First heart field (FHF) and second heart field (SHF). FHF population were defined as cells sorted by *Nkx2-5* with negligible expression of *Isl1*. SHF cells were sorted by *Isl1*. P=0.021 after Wilcox test from FindMarkers

3.2. To study cardiac contractility, the authors used MYOCYTER software and presented representative traces. To be able to conclude on abnormal contraction amplitude, a specific quantification of peaks amplitude should be performed on different biological replicates for both *Isl1-cre;Rnf20fl/fl* and control traces and followed by statistical testing. These results must be added along with representative traces.

Response: We thank the reviewer for this valuable suggestion. As requested, we have now performed a specific quantification of peak amplitudes and frequency across multiple biological replicates for both *Isl1^{cre/+}Rnf20^{fl/fl}* and control hearts. The results, including statistical analysis, have been added to the revised manuscript alongside the representative MYOCYTER traces (Fig. 2i). To quantify cardiac arrhythmia, we utilized Poincaré plot analysis based on RR intervals extracted from image sequences processed with MYOCYTER. Specifically, after identifying contraction peaks in the cardiomyocyte beating trace, the time interval between one peak and the next was defined as RR_n , and the subsequent interval as RR_{n+1} . These paired intervals (RR_n, RR_{n+1}) were then plotted against each other in a two-dimensional scatter plot, generating the Poincaré plot, which allows visualization and quantification of beat-to-beat variability (Fig. 2h, Supplementary Table 2).

Fig. 2

Fig. 2: **g** Analysis of image sequences recorded on E10.5 control and *Isl1^{cre/+}Rnf20^{fl/fl}* hearts using MYOCYTER⁴⁹. **h** Poincaré plot of image sequences recorded on E10.5 control and *Isl1^{cre/+}Rnf20^{fl/fl}* hearts. **i** Frequency and amplitude of image sequences recorded on E10.5 control and *Rnf20 Isl1^{cre/+}Rnf20^{fl/fl}* hearts using MYOCYTER⁴⁹.

3.3. Regarding the number of *Isl1-cre;Rnf20fl/fl* and control hearts showing arrhythmic behavior, a contingency table should be build and a statistical test should be applied to test the arrhythmic behavior to genotype association.

Response: We have now provided a contingency table showing the incidence of arrhythmic behavior in *Isl1^{cre/+}Rnf20^{fl/fl}* and control hearts. Statistical significance was assessed using Fisher's exact test ($p = 0.0039$) (Figure S2C).

Supplementary Fig. 2c

	Arrhythmic	Normal Rhythm	Total
Control (WT)	1	11	12
Isl1^{cre/+}Rnf20^{fl/fl}	5	1	6
Total	6	12	18
Fisher's exact test			p = 0.0039

Supplementary Fig. 2c Contingency table showing the incidence of arrhythmic behavior in control and *Isl1^{cre/+}Rnf20^{fl/fl}* hearts. Statistical significance was assessed using Fisher's exact test (p = 0.0039).

3.4. For the analysis of H2Bub1 deposition, the authors conclude on "a more pronounced decrease in the RV" compared to the LV in both *Isl1-cre;Rnf20fl/fl* CMs and ECs. However, no direct comparison has been made, so this conclusion is premature. A direct statistical testing between *Isl1-cre;Rnf20fl/fl* right ventricle and *Isl1-cre;Rnf20fl/fl* left ventricle samples must be performed to be able to conclude this.

Response: In the revised manuscript, we further elaborate on the decrease in H2Bub1 levels, and the statement in question is no longer included in the text (page 9, lines 10-11). Specifically, H2Bub1 levels are dynamically regulated by the coordinated actions of E3 ubiquitin ligases, such as RNF20/RNF40, and deubiquitinases (DUBs). Thus, the observed decrease in H2Bub1 in the LV may result from secondary changes in other components of this regulatory machinery. Indeed, our qPCR analysis of microdissected RV and LV tissue revealed increased *Rnf40* expression in the RV, likely reflecting a compensatory response, and decreased expression in the LV, which may contribute to the reduced H2Bub1 levels (Figure S2h).

4.1. To investigate deeper the impact of *Rnf20* loss in second heart field progenitors (*Isl1+*) of mouse E10.5 embryos, the authors performed a scRNA-seq analysis. In Figure 4a., the schematic explanation is misleading as it represents the looping heart tube only, but SHF progenitors (which are located in the embryo trunk, dorsally from the heart tube) were identified by the analysis. A more detailed explanation of the dissection strategy must be provided.

Response: We apologize for not clearly articulating our findings in the previous version. Our RNA-seq analyses were performed on dissected E10.5 hearts, which as the reviewer noted no longer contain undifferentiated progenitor cells. Our intention was to refer specifically to aSHF-derived lineages. In the revised manuscript, we have clarified the annotation of cell populations identified in the single-cell RNA-seq analysis. The population previously labeled as aSHF corresponds to cells in the distal outflow tract mesenchyme (dOFTm), which exhibit mesenchymal characteristics (old Fig. 4b, in the revised manuscript Fig. 3b). This population is significantly expanded in the mutants (Fig. 3c), consistent with the increased EndMT

observed in the commonly used OFT cushion EndMT assay (Fig. 6a), and with the enhanced migratory capacity of endothelial cells upon *Rnf20* loss of function (Fig. 6b–e).

Fig. 3

Fig. 3: *Rnf20* loss alters cardiac cellular composition. **a** Schematic representation of the heart dissection procedure used for scRNA-Seq. Hearts from five control and *Isl1^{cre/+}Rnf20^{fl/fl}* embryos (n = 5) were dissected for the experiment. **b** t-Stochastic neighbour embedding (t-SNE) plot of 10x genomics scRNA-Seq datasets of E10.5 control and *Isl1^{cre/+}Rnf20^{fl/fl}* hearts. CM1 and CM2, cardiomyocytes 1 and 2; EndoV, endocardial valve cells; AVCm, atrioventricular canal mesenchyme; EpiC1 and EpiC2, epicardial cells 1 and 2; EC/EndoC, endothelial cells/endocardial endothelial cells; dOFTm, dorsal outflow tract mesenchyme; dOFTe, dorsal outflow tract epithelium; cMES1 and cMES2, cardiac mesenchyme 1 and 2; OFT-MES, outflow tract mesenchyme; Myeloid cells. **c** Percentage of cells within the different clusters colored by cell types. P-value obtained from binomial exact test.

The population previously labeled as pSHF corresponds to cardiac mesenchymal cells expressing pSHF markers, now designated as cMES2. The observed increase in this population is likely due to enhanced EndoMT of pSHF-derived endocardial cells following *Rnf20* loss.

4.2. Related to Figure 4b., a UMAP or t-SNE representation of the control and *Rnf20*-loss samples separately must be provided in Supplement to present the cell distribution per genotype.

4.3. The authors used previously published gene sets for cell annotation of Figures 4b and 4f. Heatmaps showing the expression of these markers per cluster must be provided in Supplement to assess the quality of the annotation.

Response to 4.2 and 4.3: We have now provided t-SNE plot separated by genotype (control and *Isl1^{cre/+}Rnf20^{fl/fl}* hearts) and colored according to cell type in Figure Supplementary Fig. 3a. In addition, we have included heatmaps showing the expression of the marker genes used for cell annotation in Supplementary Fig. 3b. Markers for all clusters shown in Fig. 3b, 3f, and 3h are now provided in Supplementary Table 3.

Supplementary Fig. 3.

Supplementary Fig. 3. Single cell analysis of control and *Isl1^{cre/+}Rnf20^{fl/fl}* hearts. a t-SNE plot separated by genotype and colored according to cell type. **b** Heatmap showing the top 3 markers for each cell type on the left, and markers used to define the cell populations from Fig. 3b on the right.

In the revised manuscript, rather than clustering *Neb1⁺/Tnni3⁺/Myh6⁺* cardiomyocytes, we have now reclustered the cardiomyocyte populations (CM1 and CM2) at a resolution of 0.8 (Fig. 3f, 3g). We provide a t-SNE plot separated by genotype (control and *Isl1^{cre/+}Rnf20^{fl/fl}* hearts) and colored by cell type in Supplementary Fig. 4a, along with heatmaps displaying the expression of marker genes used for cell type annotation in Supplementary Fig. 4b.

Fig. 3

Supplementary Fig. 4

Fig. 3f, g Uniform Manifold Approximation and Projection (UMAP) plot showing the re-clustering of CM1 and CM2 from panel b (f) and bar plot showing the percentage of cells within the different clusters colored by cell type (g). vCM1-3, ventricular cardiomyocytes 1-3; aCMs/AVC, atrial/atrioventricular canal CMs; LV, left ventricle; RA/SAN, right atrium/sinoatrial node; pvCMs, proliferative ventricular CMs; OFT, outflow tract; LA, left atrium. **Supplementary Fig. 4a** t-SNE plot separated by genotype and colored according to CM subtype. **b** Heatmap showing the top 3 markers for each CM subtype on the left, and markers used to define the cell populations from Fig. 3f on the right.

4.4. In Figures 4d. and 4g., the authors concluded on differences in cell type proportion between genotypes, but no statistical evidence was provided. It is unclear whether the authors processed different biological replicates per genotype, that could have allowed them to definitely conclude on cell type proportion differences. In the present version of the data analysis, no statistically robust conclusion can be claimed, as alternate explanations, for example dissection bias, could be implicated in the observed differences.

Response: We thank the reviewer for pointing this out. The single-cell RNA-seq analysis was performed using pooled dissected hearts from five control and five *Isl1^{cre/+}Rnf20^{fl/fl}* knockout embryos. To assess whether the observed frequency of a specific cell population in the *Rnf20* knockout condition was significantly changed than expected based on the wild-type distribution, we used a binomial test. This statistical test evaluates whether the observed underrepresentation is statistically significant compared to the expected frequency (the expected proportion under the null hypothesis). The details have now been included in the revised Materials and Methods section as well as the corresponding figure legend. The observed significant differences align with the phenotypic changes discussed in response to reviewer comment 4.1.

Fig. 2

Fig. 3b t-Stochastic neighbour embedding (t-SNE) plot of 10x genomics scRNA-Seq datasets of E10.5 control and *Isl1^{cre/+}Rnf20^{fl/fl}* hearts. CM1 and CM2, cardiomyocytes 1 and 2; EndoV, endocardial valve cells; AVcM, atrioventricular canal mesenchyme; EpiC1 and EpiC2, epicardial cells 1 and 2; EC/EndoC, endothelial cells/endocardial endothelial cells; dOFTm, dorsal outflow tract mesenchyme; dOFTe, dorsal outflow tract epithelium; mCMs, mature cardiomyocytes; cMES1 and cMES2, cardiac mesenchyme 1 and 2; OFT-MES, outflow tract mesenchyme; Myeloid cells. **c** Percentage of cells within the different clusters colored by cell types. P-value obtained from binomial exact test. **f, g** Uniform Manifold Approximation and Projection (UMAP) plot showing the re-clustering of CM1 and CM2 from panel b (**f**) and bar plot showing the percentage of cells within the different clusters colored by cell type (**g**). vCM1-3, ventricular cardiomyocytes 1-3; aCMs/AVC, atrial/atrioventricular canal CMs; LV, left ventricle; RA/SAN, right atrium/sinoatrial node; pvCMs, proliferative ventricular CMs; OFT, outflow tract; LA, left atrium.

4.5. In Figure 4i., the authors identified five EC populations.
 • A heatmap of top marker genes specific to each population must be provided in Supplement to better characterize each EC population.

Response: We have now included a heatmap displaying the top marker genes specific to each

Supplementary Fig. 4

of the four endothelial cell (EC) populations identified in the revised Fig. 3h (previously Figure 4i), now shown in Supplementary Fig. 4d. The previous cluster EC1 has been removed, as it does not represent an endothelial population but rather erythroid cells.

Supplementary Fig. 4d Heatmap showing the top 3 markers for each EC subtype on the left, and the markers used to define the cell populations from Fig. 3h on the right.

• The authors provided some explanations for cluster EC3, EC4 and EC5, but no information is provided for cluster EC1 and EC2. What about these two clusters?

Response: In the revised manuscript, we have further refined the characterization of the endothelial cell clusters. In the updated Fig. 3i, we now include the expression of endothelial markers, EndMT-associated markers, and mesenchymal markers, highlighting the progression of EndMT across the different clusters. As mentioned, EC1 has been removed, as it does not represent an endothelial population but rather erythroid cells. In addition, we have included Gene Ontology (GO) analysis and representative genes characterizing the EC2 cluster in Fig. 3j.

Re-clustering of ECs identified four distinct clusters (Fig. 3h, Supplementary Fig. 4c, 4d, Supplementary Table 3). Endothelial cluster 2 (EC2) exhibited high expression of endocardial markers such as *Npr3* and *Nfatc1*, along with general endothelial markers including *Pecam1* and *Kdr*, while showing minimal expression of mesenchymal markers such as *Postn* and *Tagln* (Fig. 3i). Differential expression analysis followed by GO analysis revealed that *Rnf20* deficiency in this cluster results in decreased levels of genes linked to positive regulation of cell cycle and regulation of developmental growth, including factors with key function during cardiogenesis such as *Igf1/2*, *Nrg1*, *Nrp1* and *Slit2*.

Fig. 3h UMAP analysis of EndoV, EndoC and OFT-MES populations from panel b, presenting distinct EC populations. **i** Dot plot showing the expression of different endothelial and EndMT markers. **j** Representative GO terms in genes upregulated (light red) and downregulated (light blue) in the different EC populations in **i**.

In contrast, upregulated genes were involved in glycolysis and p53-mediated signaling (Fig. 3j, Supplementary Fig. 4e). EC3 showed initial activation of EndMT-associated markers, such as *Zeb2*, which progressively increased in EC1 and was most pronounced in EC4. GO analysis of differentially expressed genes in EC3 upon *Rnf20* loss revealed an upregulation of genes associated with cell migration, while EC5 exhibited increased expression of genes involved in the TGF- β response and downregulation of cell cycle-related genes following *Rnf20* deficiency (Fig. 3i, 3j).

4.6. *There are few details concerning the bioinformatic processing of the scRNA-seq data, in term of cell quality selection (mitochondrial content, cell cycle regression), normalization and batch effect correction? It would be important for reproducibility aspect, to make available the entire code used in this study.*

Response: We thank the reviewer for highlighting the importance of transparency and reproducibility. In the revised manuscript, we have now included additional details regarding the bioinformatic processing of the scRNA-seq data. Moreover, we have made the full analysis code available via Github: https://github.com/jcorderJC12/03RNF20_CHD_NatCom, allowing for full reproducibility of the results presented in this study.

5.1. *The authors performed in Figure 5 a similar analysis as Figure 3., but here focused on Rnf20-loss in EC specifically. Regarding the number of Rnf20iEC-KO and control hearts*

showing arrhythmic behavior, a contingency table should be build and a statistical test should be applied to test the arrhythmic behavior to genotype association.

5.2. As for Figure 3e., to be able to conclude on abnormal contraction amplitude in Figure 5d., a specific quantification of peaks amplitude on different biological replicates should be performed for both *Rnf20*^{IEC-KO} and control traces, followed by statistical testing.

Response to 5.1 and 5.2: We have now provided a contingency table showing the incidence of arrhythmic behavior in *Rnf20*^{IEC-KO} and control hearts (Supplementary Fig. 5e). To evaluate whether the difference in the frequency of arrhythmia between wild-type and knockout embryos is statistically significant, we used Fisher's exact test. We have also included quantification of peak amplitudes and frequency across multiple biological replicates for both *Rnf20*^{IEC-KO} and control hearts. The results, including statistical analysis, have been added to the revised manuscript alongside the representative MYOCYTER traces (Fig. 4c-e). To quantify cardiac arrhythmia, we utilized Poincaré plot analysis based on RR intervals extracted from image sequences processed with MYOCYTER (Fig. 4d, Supplementary Table 5).

Supplementary Fig. 5

e

	Arrhythmic	Normal Rhythm	Total
Control (WT)	1	6	7
Rnf20 ^{IEC-KO}	7	2	9
Total	8	8	16
Fisher's exact test			p = 0.011

Supplementary Fig. 5e Contingency table showing the incidence of arrhythmic behavior in control (*Cdh5*^{CreERT2}^{neg}-*Rnf20*^{fl/fl}) and *Rnf20*^{IEC-KO} (*Cdh5*^{CreERT2}^{pos}-*Rnf20*^{fl/fl}) hearts. Statistical significance was assessed using Fisher's exact test (p = 0.0011).

Fig. 4

Fig. 4: Endothelial RNF20 is essential for heart development and function. **a** Schematic representation of the experimental setup. **b** Relative *Rnf20* mRNA expression in ECs isolated from E11.5 control (*Cdh5*^{CreERT2}^{neg}-*Rnf20*^{fl/fl}) and *Rnf20*^{IEC-KO} (*Cdh5*^{CreERT2}^{pos}-*Rnf20*^{fl/fl}) embryos. **c** Analysis of image sequences recorded on control and *Rnf20*^{IEC-KO} hearts using MYOCYTER. **d** Poincaré plot derived from image sequences recorded on E11.5 control and *Rnf20*^{IEC-KO} hearts. **e** Frequency and amplitude extracted from image sequences recorded on E11.5 control and *Rnf20*^{IEC-KO} hearts using MYOCYTER.

5.3. For the explanation of Figures 5k. and 5m., the authors said that the “ablation of *Rnf20* in ECs at P1 resulted in lethality, disorganized ECs, abnormal CM alignment and sarcomeric organization”. Together with representative pictures, it will be more robust to provide quantifications of EC organization, CM alignment and sarcomeric organization on multiple cells, from different biological replicates, followed by an appropriate statistical test.

Response: In the revised manuscript, we now provide quantification of EC morphology and cardiomyocyte sarcomeric organization, as follows: Consistent with the phenotype observed in embryonic development, ablation of *Rnf20* in ECs at P1 resulted in lethality after P7, a shift from the typical elongated and flattened EC morphology to a disorganized, morphologically disordered phenotype (Fig. 4l, Supplementary Fig. 5h, 5i), and abnormal sarcomeric organization (Fig. 4m, Supplementary Fig. 5j).

Supplementary Fig. 5

Fig. h, i Representative image of ECs morphology in P7 Control and *RNF20*^{IECKO} hearts (**h**) and quantification of normal versus abnormal EC morphology (arrows) normalized to the total EC count per field (**i**) (n=3 hearts for each group). **j** Quantification of CM showing organized versus disorganized sarcomeres in P7 Control and *RNF20*^{IECKO} hearts normalized to the total number of CM per field (n=3 hearts for each group).

6.1. To investigate deeper the impact of *Rnf20* loss in endothelial cells (*Cdh5*⁺) in mouse E11.5 embryos, the authors performed a scRNA-seq analysis. Figure 6 was constructed in the same way than Figure 4, so the comments are similar for both Figure. The schematic explanation presented in Figure 6a., is misleading as it represents only the heart, but SHF progenitors (which are located in the embryo trunk, dorsally from the heart tube) were identified by the analysis. A more detailed explanation of the dissection strategy must be provided.

Response: We apologize for not clearly articulating our findings in the previous version. As the reviewer correctly noted, the developing E11.5 heart no longer contains undifferentiated progenitor cells. Our intention was to specifically refer to SHF-derived lineages. In the revised manuscript, we have now clarified the annotation of the cell populations identified in the single-cell RNA-seq analysis of control and *Rnf20*^{IECKO} (*Cdh5*^{CreERT2}_{pos}-*Rnf20*^{fl/fl}) E11.5 hearts to reflect this more accurately, as presented below in response to point 6.2 and 6.3.

6.2. Related to Figure 6b., a UMAP or t-SNE representation of the control and *Rnf20*-loss samples separately must be provided in Supplement to present the cell distribution per genotype.

6.3. It's unclear which marker genes the authors used for cell annotation of Figures 6b and 6f. If they are the same as for Figure 4, it must be indicated in the manuscript. Heatmaps showing the expression of these markers per cluster must be provided in Supplement to assess the quality of the annotation.

Response to 6.2 and 6.3: We have now provided t-SNE plot separated by genotype (control and *Rnf20*^{IEC-KO} hearts) and colored according to cell type in Figure Supplementary Fig. 6a and provided heatmap showing the expression of characteristic marker genes used for cell annotation in Supplementary Fig. 6b. In addition, we now provide supplementary table listing marker gene expression across different cardiac cell populations from control and *Rnf20*^{IEC-KO} embryos (Supplementary Table 4).

Supplementary Fig. 6

Supplementary Fig. 6. Single cell analysis of control and *Rnf20*^{IEC-KO} hearts. **a** t-SNE plot separated by genotype and colored according to cell type. **b** Heatmap showing the top 3 markers for each cell type on the left, and markers used to define the cell populations from Fig. 5b on the right.

In the revised version of the manuscript, rather than clustering *Neb1*⁺/*Tnni3*⁺/*Myh6*⁺ cardiomyocytes, we have now reclustered the cardiomyocyte populations (CM1 and CM2) at a resolution of 0.8 (Fig. 3f, 3g). We provide a t-SNE plot separated by genotype (control and *Rnf20*^{IEC-KO} E11.5 hearts) and colored by cell type in Supplementary Fig. 7a, along with heatmaps displaying the expression of marker genes used for cell type annotation in Supplementary Fig. 7b. Marker gene expression across different CM populations are included in Supplementary Table 4.

Supplementary Fig. 7

Supplementary Fig. 7. Cardiomyocyte and endothelial clusters in control and *Rnf20*^{IEC-KO} hearts. **a** t-SNE plot separated by genotype and colored according to CM subtype. **b** Heatmap showing the top 3 markers for each CM subtype on the left, and markers used to define the cell populations from Fig. 3f on the right.

6.4. In Figures 6d. and 6g., the authors concluded on differences in cell type proportion between genotypes, but no statistical evidence was provided. It is unclear whether the authors processed different biological replicates per genotype, that could have allowed them to definitely conclude on cell type proportion differences. In the present version of the data analysis, no statistically robust conclusion can be claimed, as a dissection bias could be implicated in the observed differences.

Response: As mentioned for the single-cell RNA-seq analysis of *Rnf20* function in the SHF, we also used pooled hearts from five control and five *Rnf20*^{EC-KO} embryos for the endothelial-specific deletion experiments. To determine whether the frequency of specific cell populations in the *Rnf20*^{EC-KO} hearts differed significantly from the expected distribution based on wild-type embryos, we applied a binomial test. This statistical approach allows us to assess whether the observed underrepresentation is statistically significant. The details of this analysis are now included in the revised Materials and Methods section as well as in the corresponding figure legend.

Fig. 5: **b** t-SNE plot of 10x genomics scRNA-Seq datasets of E11.5 control and *Rnf20*^{EC-KO} hearts. OFT/AVCm1 and 2, outflow tract/atrioventricular canal mesenchyme 1 and 2; dOFT, dorsal outflow tract; Ery, erythrocytes; EndoC, endocardial cells; CM1 and CM2, cardiomyocytes 1 and 2; pMES, proliferative mesenchyme; EpiC1 and EpiC2, epicardial cells 1 and 2; OFTe, outflow tract epithelium; EndoV, endocardial valve cells; HSPC, hematopoietic stem and progenitor cells. **c** Percentage of cells within the different clusters colored by cell types. P-value obtained from binomial exact test. **e** UMAP re-clustering of CM1 and CM2 from panel b, colored by CM type. vCM1 and vCM2, ventricular cardiomyocytes 1 and 2; pvCM1-2, proliferative ventricular CM1-2; mvCM, mature ventricular CM; aCM1 and aCM2, atrial cardiomyocytes 1 and 2; aCM_{im}, immature atrial cardiomyocytes; AVC/OFT, atrioventricular canal/ outflow tract; OFT, outflow tract. **f** Feature plots of *Mki67* expression in the different CM populations in **e**. **g** Percentage of cells in each CM subtype, color-coded by cell type. P-values calculated using the binomial exact test.

7.1. To study cardiac contractility of rat cardiomyocytes co-cultured with HUVECs treated or not with siRNA, the authors used MYOCYTER software and presented representative traces in Figure 7j. To be able to conclude on arrhythmic events and abnormal contraction amplitude, specific analyses must be performed using Poincare plots and quantification of peaks amplitude, on different biological replicates for both control and conditions traces. These results must be added along with representative traces.

8.1. In Figure 8d., the authors presented representative contraction traces of rat cardiomyocytes co-cultured with HUVECs treated or not with siRNA. In the description of the Figure 8d., the authors conclude on the “overexpression of NPPA, LOXL2 and MIF resulted in highly arrhythmic beating behavior”. The conclusion is overstated as the related trace presented only a higher frequency but no obvious arrhythmic behavior. To be able to conclude

on arrhythmic events and abnormal contraction amplitude, specific analyses must be performed using Poincare plots and quantification of peaks amplitude, on different biological replicates for both control and conditions traces. These results must be added along with representative traces.

Response to 7.1 and 8.1: We thank the reviewer for this valuable suggestion. As requested, we have now included Poincaré plots and quantification of peak amplitudes and frequency based on independent biological replicates for all experiments addressing cardiac contractility in the co-culture system. These data are presented in Fig. 7h–j, Fig. 9a–g, Supplementary Fig. 10a–d and Supplementary Table 9, 11 of the revised manuscript.

Reviewer #3 (Remarks to the Author):

Dou et al. investigated the causes of Tetralogy of Fallot (ToF), one of the most common types of cyanotic Congenital Heart Disease (CHD), and discovered that the H2Bub ligase RNF20 was one of the down-regulated genes in patients with low peripheral oxygen saturation (<88%). Since RNF20 mutation has been associated with CHD, they decided to focus on the connection between RNF20 and heart development. Utilizing a series of cell culture, zebrafish, and conditional knockout mouse models, and combining these with sc-RNA-seq analysis, they concluded that RNF20-H2Bub mediates SHF formation by modulating optimal EndoMT during cardiac development. The discovery that Rnf20 in cardiac endothelial cells (ECs) is required for Second Heart Field (SHF) development, and its role in modulating the interaction between ECs and cardiomyocytes (CMs) through optimal TGF-beta and angiocrine signaling, is interesting. However, this paper falls short in providing mechanistic insights and could benefit from improved data interpretation, better organization, and a writing style more accessible to a broader audience.

Response: We thank the reviewer for the interest in our manuscript and for the constructive comments, which have helped us significantly improve the quality of the work.

Major comments:

1. The two key themes of this paper are TGF-beta and angiocrine signaling. However, the introduction provides an insufficient overview regarding the key knowledge gaps in the interaction between these two pathways. Additionally, they did not include two recent publications that are highly relevant to this paper: PMID: 38038666 and PMID: 37967007, particularly the study by Barish et al.

Response: We thank the reviewer for this valuable comment. In response, we have revised the introduction to place greater emphasis on the roles of TGF- β signaling and angiocrine cues, and we now highlight the emerging interplay between these pathways in the regulation of cardiac development and function (page 4, lines 21-27 and page 5, lines 7-20). We also explicitly refer to two key studies that investigated Rnf20 and Rnf20/Rnf40-mediated H2Bub1 function in cardiomyocytes during development and postnatal growth (page 6, lines 8-10).

In light of this and the feedback from the other reviewers, we have revised our focus to specifically examine the role of Rnf20 in endothelial cells, where *RNF20* expression is reduced in TOF patients (Fig. 1a), as well as in second heart field progenitors, which exhibit significantly higher *Rnf20* levels compared to first heart field progenitors (Fig. 2a). Importantly, *Rnf20* is highly expressed in the *Isl1*-derived endocardial lineages (Supplementary Fig. 3c-e), where our findings reveal its critical role in limiting EndMT and maintaining physiological angiocrine signaling, which is essential for normal heart development and function.

Fig. 1

Fig. 2

Supplementary Fig. 3

Fig. 1a Dot plot showing the expression of *RNF20* in different cell populations separated by Donor and Tetralogy of Fallot (TOF) patients from (GSE203274). P-value after Wilcox test from FindMarkers. **Fig. 2a** Dot plot of *Rnf20* expression in FHF and SHF cells of sorted *Isl1*⁺ and *NKX2.5*⁺ E7.75 embryos⁴⁴. First heart field (FHF) and second heart field (SHF). FHF population were defined as cells sorted by *Nkx2-5* with negligible expression of *Isl1*. SHF cells were sorted by *Isl1*. P=0.021 after Wilcox test from FindMarkers. **Supplementary Fig. 3:** **c** Dot plot illustrating the expression of *Isl1* and *Rnf20* across various cell types. **d** UMAP plot depicting cell clusters within the *Isl1* lineage in E8.75 embryos⁴⁹. **e** Dot plot displaying marker genes for CMs, ECs, and proliferative populations.

2. *Their efforts using mouse embryonic stem cells (mESCs) demonstrated that Rnf20 haploinsufficiency promoted cardiomyocyte (CM) differentiation but inhibited endothelial cell (EC) differentiation. However, it is clear from their own and published results that Rnf20 haploinsufficiency, which is not equivalent to loss of function, has no impact on cardiogenesis during heart development in mice. Therefore, this set of data does not align well with the current research theme.*

Response: We thank the reviewer for pointing this out. We acknowledge that our original description of the generation of the haploinsufficient mESC line and its differences from the in vivo model was insufficient. The targeting strategy employed for the mESC line differs from that used in our animal model and resulted in a ~70% reduction in Rnf20 expression compared to controls, as shown in the previous version of the manuscript in Figure S1c. However, given the revised focus of the current manuscript, we have opted to use an alternative cell culture model that more closely reflects the in vivo systems employed in this study, thereby ensuring greater consistency and relevance across our experimental approaches.

To enable a better comparison with our in vivo experiments, we depleted *Rnf20* in mesodermal precursors and assessed its impact on endothelial cells using a protocol for directed differentiation of mESCs in endothelial cells (Fig. 6f). Flow cytometric analysis for PECAM1 expression at day 7 (D7) revealed no significant difference in EC yield between control and Rnf20-deficient cells (Fig. 6g, 6h), suggesting that Rnf20 is not essential for initial EC specification. However, transcriptomic analysis of sorted PECAM1⁺ cells showed a marked upregulation of genes associated with positive regulation of cell locomotion and extracellular matrix organization, indicative of a more migratory and mesenchymal-like phenotype (Fig. 6i, Supplementary Fig. 8c, Supplementary Table 4). Conversely, genes involved in response to growth factors and receptor tyrosine kinase signaling were significantly downregulated, consistent with our in vivo observations. Comparison of differentially expressed genes in Rnf20-deficient mESC-derived ECs and HUVECs revealed a shared enrichment of genes promoting cell locomotion among the upregulated set, and genes regulating growth factor responses among the downregulated set, pointing to a conserved role of Rnf20 in endothelial cells (Fig. 6j).

By day 14 of directed EC differentiation, control ECs formed organized monolayers with robust VE-Cadherin (*Cdh5*) expression, whereas Rnf20-deficient cells displayed disrupted monolayer integrity and a substantial loss of VE-Cadherin expression (Fig. 6k). Notably, many of these cells expressed the mesenchymal marker *Snai1*, further supporting premature or aberrant induction of EndMT in the absence of Rnf20. Together, these findings indicate that while Rnf20

is dispensable for early EC specification, it is essential for maintaining endothelial identity and preventing aberrant mesenchymal transition.

Fig. 6

Fig. 6: Endothelial RNF20 inhibits EndMT. **f** Schematic representation of the directed mESC differentiation system into endothelial cells with lentiviral transduction of a control and shRnf20 construct at the mesoderm stage (mESC, mouse embryonic stem cell; MES, mesoderm; EC, endothelial cells). **g** Representative FACS analyses of Pecam1⁺ (CD31⁺) endothelial cells at day 7 of differentiation depicted in **f**. Highly CD31-positive cells were FACS sorted for RNA-seq. **h** Percentage of Pecam1⁺ ECs at day 7. **i** Representative GO terms of genes enriched among significantly upregulated (violet) and downregulated (blue) genes in RNA-seq datasets of sorted Pecam1⁺ shRnf20 ECs versus control differentiated ECs (padj < 0.05). **j** Representative GO terms enriched among shared significantly upregulated (red) and downregulated (blue) transcripts (padj < 0.05 for both datasets) upon RNF20 depletion in both HUVECs and mouse ESC differentiated ECs. **k** Immunostaining for Cdh5, Snai1 and DAPI of day 14 ECs derived from mESCs.

3. Their initial efforts using a zebrafish model provided a hint that Rnf20 might regulate Isl1-mediated Second Heart Field (SHF) development. This hypothesis was confirmed using an Isl1-Cre mouse model. They observed decreased cardiomyocyte (CM) proliferation while

endothelial cells (ECs) were not affected, indicating an altered interaction between these two cell populations due to the absence of Rnf20. However, their conclusion that 'Rnf20 potentially plays a role in both CMs and endocardial ECs for SHF development' (page 10) is not clearly substantiated by their data.

Response: We thank the reviewer for pointing this out. In the revised version of the manuscript, we have rephrased this conclusion to: Taken together, our results demonstrate that Rnf20 plays a key role in SHF development and suggest an important function in endocardial ECs in regulating EndMT and CM proliferation (page 11, lines 16-18). As already mentioned above, we now include data demonstrating that Rnf20 is highly expressed in the Isl1⁺ lineage-traced endocardial ECs, whereas its expression is low in cardiomyocytes (Supplementary Fig. 3c-e).

4. The description of the subsequent scRNA-seq analysis in the Isl1-Cre mouse heart was diffuse and rambling, yet it ultimately supports a role for Rnf20 in SHF development. However, the interpretation of how RNF20 deficiency alters transcription in endocardial endothelial cells (ECs), particularly in terms of increased gene expression in response to TGF-beta, was not clearly conveyed. Additionally, the conclusion that endothelial cells (ECs) play an important role in cardiac morphogenesis appears redundant, considering the well-established role of ECs, particularly through endothelial-to-mesenchymal transition (EndoMT), in the formation of heart valves and septa.

Response: We thank the reviewer for highlighting this important point. We have extensively revised Fig. 3 and the corresponding text in the Results section to clearly convey two key findings: (1) the increased EndMT, and (2) the reduced expression of critical angiocrine factors in endocardial endothelial cells upon Rnf20 loss.

In the revised Fig. 3, we demonstrate that the number of mesenchymal cells, including dorsal outflow tract mesenchyme (dOFTm), OFT mesenchyme (OFT-MES), and cardiac mesenchymal cells (cMES), was significantly elevated in *Isl1^{cre/+}Rnf20^{fl/fl}* hearts (Fig. 3c). Moreover, we show that Rnf20 is significantly downregulated in endocardial cells (endoC, Fig. 3d). We have also revised the text discussing the transcriptional changes in ECs in *Isl1^{cre/+}Rnf20^{fl/fl}* hearts, as follows: Re-clustering analysis of endothelial cells further identified four distinct subclusters (Fig. 3h, Supplementary Fig. 4c, d). EC cluster 2 (EC2) highly expressed endocardial/EC markers, such as *Pecam1* and *Kdr*, but showed very low expression of mesenchymal markers such as *Postn* and *Tagln* (Fig. 3i). Differential expression analysis followed by GO analysis revealed that Rnf20 deficiency in this cluster results in decreased levels of genes linked to positive regulation of cell cycle and regulation of developmental growth, including factors with key function during cardiogenesis such as *Igf1/2*,

Nrg1, *Nrp1* and *Slit2*. In contrast, upregulated genes were involved in glycolysis and p53-mediated signaling (Fig. 3j, Supplementary Fig. 4e). EC3 showed initial activation of EndMT-associated markers, such as *Zeb2*, which progressively increased in EC1 and was most pronounced in EC4. GO analysis of differentially expressed genes in EC3 upon *Rnf20* loss revealed an upregulation of genes associated with cell migration, while EC5 exhibited increased expression of genes involved in the TGF- β response and downregulation of cell cycle-related genes following *Rnf20* deficiency (Fig. 3i, 3j). Taken together, our results demonstrated that *Rnf20* plays a key role in SHF development and suggested an important function in endocardial ECs for cardiac morphogenesis and function.

Fig. 3

Fig. 3: Conditional deletion of *Rnf20* in *Isl1*-derived cells alters cardiac cellular composition. **a** Schematic representation of the heart dissection procedure used for scRNA-Seq experiments. **b** t-Stochastic neighbour embedding (t-SNE) plot of 10x genomics scRNA-Seq datasets of E10.5 control and *Isl1^{cre/+}Rnf20^{fl/fl}* hearts. CM1 and CM2, cardiomyocytes 1 and 2; EndoV, endocardial valve cells; AVCm, atrioventricular canal mesenchyme; EpiC1 and EpiC2, epicardial cells 1 and 2; EndoC, endocardial cells; dOFTm, dorsal outflow tract mesenchyme; dOFTe, dorsal outflow tract epithelium; cMES1 and cMES2, cardiac mesenchyme 1 and 2; OFT-MES, outflow tract mesenchyme; Myeloid cells. **c** Percentage of cells within the different clusters colored by cell types. P-value obtained from binomial exact test. **d** Dot plot showing *Rnf20* and *Isl1* expression in the selected cell populations from panel b, separated by genotype. P-values were calculated using the Wilcoxon test implemented in FindMarkers. **e** Dot plot showing the expression of different endothelial and EndMT markers. **f** Representative GO terms in genes upregulated (light red) and downregulated (light blue) in the different EC populations in **e**.

5. Further genetic analysis in mice led to the intriguing discovery that *Rnf20* in ECs is essential for SHF development. scRNA-seq revealed a potential mesenchymal characteristic in the EC population following *Rnf20* ablation, hinting at enhancement of endMT. The inhibitory role of *RNF20* in endMT was subsequently confirmed using two *in vitro* endMT assays.

Response: We thank the reviewer for highlighting the significance of our findings. To further support this point, we conducted additional experiments showing that loss of *Rnf20* function in endothelial cells enhances TGF- β signaling, increases Smad2/3 activation, and promotes endothelial cell migration (Fig. 6c-e). Specifically, we observed a significant increase in the number of p-SMAD2/p-SMAD3-positive cells in HUVECs following *RNF20* silencing (Fig. 6d). Notably, silencing *SMAD2* and *SMAD3* reversed the enhanced migratory capacity of *RNF20*-deficient HUVECs to control levels (Fig. 6e), indicating that activation of TGF- β signaling mediates the increased cell migration induced by *RNF20* loss of function.

Fig. 6

Fig. 6: Endothelial *RNF20* inhibits EndMT. **c** Quantification of migration rate in scratch assay in HUVECs transfected with control siRNA or siRNA against *RNF20*. Migration rate was quantified as percentage closure of the initial wound area using ImageJ. Representative images of the scratch assay are provided in Supplementary Figure 7a. **d** Immunostaining with anti-pSMAD2/3 and DAPI (**d**) and quantification of pSMAD2/3-positive cells in HUVECs transfected with control siRNA or siRNA against *RNF20*. Scale bars, 20 μ m. **e** Boyden chamber migration assay and quantification of migrated HUVECs transfected with control siRNA, siRNA against *RNF20* or *siRNF20* together with siRNAs against *SMAD2* and *SMAD3* (*siRNF20/siSMAD2/3*). Scale bars, 500 μ m.

Moreover, as detailed in our response to point 2, we used directed differentiation of embryonic stem cells into endothelial cells and found that *Rnf20* depletion at the mesodermal stage, coinciding with the onset of *Is11* expression, does not impair endothelial specification but induces aberrant EndMT (Fig. 6f-k).

6. They attempted to establish that *RNF20-H2Bub* regulates RNA polymerase II (RNA pol II) pause-release in TGF-beta genes in HUVECs using ChIP-seq analysis. However, it has already been clearly demonstrated that *BRE1A* (*RNF20*) and *H2Bub* are not associated with the release of RNA pol II into the gene body (PMID: 26279188). *RNF20-H2Bub* has been

shown to promote chromatin accessibility during cardiomyocyte maturation (PMID: 37967007) and to modulate the enhancer activity of estrogen-responsive genes. The authors should investigate these possible mechanisms in the regulation of TGF-beta genes in ECs.

Response: RNF20 plays a dual role in transcriptional control. On one hand, RNF20-mediated H2B monoubiquitination (H2Bub1) has been shown to facilitate transcriptional elongation²⁹. On the other hand, Rnf20 hinders the recruitment of TFIIS, which is required for the release of RNA polymerase II (Pol II) into active elongation at tumor-promoting genes, thereby suppressing a pro-oncogenic transcriptional program³⁰. Similarly, in a recent manuscript from our group, we demonstrated that RNF20 coordinates VEGF and Notch signaling in endothelial cells, essential for balanced vessel growth, through its dual role in regulating RNA Polymerase II activity⁴¹. This function is finely tuned, as upstream signaling pathways modulate RNF20 activity (as the reviewer noted), which in turn governs Pol II pausing and elongation^{41, 42}. We have expanded the introduction to further elaborate on the dual function of Rnf20 (page 5, lines 26-27; page 6, lines 1-6).

In the revised manuscript, we show that RNF20 restricts the recruitment of TFIIS at TGF- β pathway genes.

Fig. 10

Fig. 10: **d** Log₂ of the Pol II pausing index (PI) at genes involved in Notch, Tgf- β and Wnt signaling. **e** Genome tracks of total Pol II ChIP-Seq reads in HUVECs transfected with control siRNA (*siControl*) or siRNA against *RNF20* (*siRNF20*). **f** Ratio of the Pol II enrichment at the TSS and TTS of *TGF β 1*, *TGF β 2*, *THBS1* and *CCL2* in control and *siRNF20* HUVECs, determined by Pol II ChIP-qPCR. TSS, Transcription Start Site; TTS, Transcription Termination Site. **g** Relative mRNA expression of *TGF β 1*, *TGF β 2*, *THBS1* and *CCL2* in control and *siRNF20* HUVECs. **h** TFIIS enrichment at *TGF β 1*, *TGF β 2*, *THBS1* and *CCL2* in control and *siRNF20* HUVECs, determined by TFIIS ChIP-qPCR.

Based on our observation that TGF- β pathway genes exhibit high levels of Pol II pausing under normal conditions, and that RNF20 loss leads to a marked reduction in the Pol II pausing index, we performed Pol II ChIP-qPCR on TGF- β pathway genes in control and RNF20-knockdown HUVECs. This analysis confirmed reduced Pol II pausing upon RNF20 depletion (Fig. 10f), which correlated with a significant upregulation of these genes, as shown by qPCR (Fig. 10g). To directly assess whether RNF20 interferes with TFIIIS recruitment at TGF- β targets, we performed TFIIIS ChIP-seq in control and RNF20-knockdown HUVECs. Strikingly, TFIIIS binding at TGF- β pathway genes was significantly increased in the absence of RNF20, supporting a model in which RNF20 restrains TGF- β signaling by limiting TFIIIS recruitment and thereby modulating Pol II transcriptional elongation (Fig. 10h).

Indeed, as mentioned by the reviewer, silencing of BRE1A /RNF20 in cancer cells did not affect Pol II distribution at genes that show decreased pausing index upon PAF1 loss of function (PMID: 26279188), suggesting that PAF1 and RNF20 may regulate promoter-proximal pausing at different subset of genes.

7. Finally, their ligand-receptor analysis led to a surprising discovery: the expression of angiocrines such as Nrg1, Nrp1, Igf1, and Igf2 was significantly decreased. Furthermore, the addition of Nrg1 and the knockdown of the TGF-beta effector SMAD4 could rescue the compromised cardiomyocyte (CM) proliferation caused by Rnf20 deficiency in a co-culture model. However, the explanation of the 'NicheNet' analysis was missing, particularly concerning its ability to identify altered signaling pathways with a sensitivity surpassing that of sc-RNA-seq. The input source for the 'NicheNet' analysis is unclear, but if sc-RNA-seq data were used, a closer integration of these results with their respective sc-RNA-seq dataset would have in a smoother presentation.

Response: We apologize for the lack of clarity in the original text. We have now included an explanation of NicheNet in the Materials and Methods section, as follows: Ligand–receptor interaction analysis was performed using NicheNet (v2.0.0)⁶⁰, a computational framework that predicts intercellular communication by integrating ligand–receptor interactions with downstream signaling and transcriptional responses from single cell RNA-Seq data. In our study, a ligand was considered relevant if the corresponding angiocrine gene was highly expressed in endothelial cells (endoC), its receptor was expressed in cardiomyocytes (CMs), and the known downstream target genes of this signaling axis were differentially expressed in CMs.

We applied the `nichenet_seuratobj_cluster_de` function using the following parameters: `expression_pct = 0.10`, `lfc_cutoff = 0.25`, `filter_top_ligands = FALSE`, `top_n_ligands = 100`, and

top_n_targets = 200. This function operates on a Seurat-normalized expression matrix, identifies differentially expressed genes (DEGs) per cluster, and maps them onto the NicheNet reference database (version curated as of July 2023).

8. Furthermore, the mechanisms by which *RNF20* may regulate the expression of angiocrine signaling genes need to be addressed.

Response: We thank the reviewer for raising this point. Since *RNF20* depletion in cardiac ECs and HUVECs revealed similar effects on CM behavior, we next studied the mechanism behind the transcriptional changes in *RNF20*-deficient ECs using ChIP-sequencing for H2Bub1 and Pol II in HUVECs transfected with control siRNA and siRNA against *RNF20*⁴¹. Genome-wide H2Bub1 levels were decreased in *RNF20*-deficient HUVECs. However, the relative decrease of H2Bub1 levels was more prominent at genes downregulated in both *RNF20*-knockdown HUVECs and *Rnf20*-deficient ESC-derived ECs (Fig. 10a, 10b). These genes exhibited comparable levels of H2Bub1 enrichment at both the promoter and gene body, consistent with active transcription and efficient elongation. Among the genes showing significantly reduced H2Bub1 levels were those involved in cholesterol metabolism (e.g., *LDLR*), focal adhesion (*ZYX*), and growth factor signaling (*NRG1* and *IGF2*), consistent with their transcriptional downregulation following *Rnf20* loss in ECs (Fig. 10c). *VEGFC*, which is upregulated upon *RNF20* depletion in endothelial cells⁴¹, did not exhibit any significant changes in H2Bub1 levels.

Fig. 10

Fig. 10: **a** Genome tracks of H2Bub1 ChIP-Seq reads in HUVECs transfected with control siRNA (*siControl*) or siRNA against *RNF20* (*siRNF20*). The gray boxes highlight gene loci with lower H2Bub1 levels compared to neighboring genes. **b** Average genome-wide H2Bub1 ChIP-Seq signal as well as H2Bub1 ChIP-Seq signal at genes downregulated upon *Rnf20* LOF in both HUVECs and mESC-derived ECs. **c** H2Bub1 enrichment at *LDLR*, *ZYX*, *NRG1*, *IGF2* and *VEGFC* and *Pdk1* in HUVECs transfected with control siRNA or siRNA against *RNF20*.